# Reinforcement Learning with Action-Triggered Observations

Alexander Ryabchenko [1 2]   Wenlong Mou [1 2]

## Abstract

We introduce Action-Triggered Sporadically Traceable Markov Decision Processes (ATST-MDPs), a reinforcement learning framework for partial observability in which full state observations occur stochastically at each step, with probability determined by the chosen action. We derive Bellman equations tailored to this setting and establish the existence of an optimal policy. Exploiting the fact that sporadic observations reveal the full state, we provide an equivalent formulation in which agents commit to action-sequences between consecutive observations. Under the linear MDP assumption, we show that the value function over such action-sequences admits a linear representation in a finite-dimensional feature map, enabling standard regression-based methods. As an application, we derive ATST-LSVI-UCB, an optimistic algorithm achieving regret $\widetilde{O}(\sqrt{K d^3 (1-\gamma)^{-3}})$ for episodic learning with geometrically distributed horizons, where $K$ is the number of episodes, $d$ the feature dimension, and $\gamma$ the discount factor (episode continuation probability), matching the known rate for linear MDPs with full observability.

## 1. Introduction

Reinforcement Learning (RL) studies sequential decision-making where an agent interacts with an unknown environment. Standard formulations assume the agent observes the new environmental state after every executed action. In practice, however, state observations are often sporadic, as sensing may be costly or unreliable, and action choices may affect the frequency of observations. For example, in clinical treatment planning, the clinician needs to make tradeoffs between invasive diagnostic tests that provide accurate patient state information but carry risks, and less invasive tests that are safer but yield limited insights into the patient's condition. In financial portfolio management, traders must balance the costs of acquiring high-frequency market data against the benefits of informed decision-making.

This class of problems belongs to the general framework of Partially Observable Markov Decision Processes (POMDPs) (Aström, 1965), where the agent receives noisy partial observations generated from the underlying state. However, it is well-known that general POMDPs without additional structures are computationally and statistically intractable (Madani et al., 1999; Jin et al., 2020a). As a result, existing general-purpose POMDP methods lack specificity for scenarios where the availability of state observations depends on the agent's actions.

To address this gap, we propose a novel RL framework characterized by "action-triggered observations," where each action $a$ has an associated probability $\beta(a) \in [0, 1]$ of revealing the resulting state upon execution. A control policy must therefore simultaneously optimize actions under partial observability and strategically decide when to trigger observations to reduce uncertainty. We formalize this setting as Action-Triggered Sporadically Traceable Markov Decision Processes (ATST-MDPs), extending classical MDPs with the state observation probability function $\beta$.

The ATST-MDP framework captures a range of observation mechanisms pertaining to *active perception* (e.g., Bajcsy, 1988). In particular, the framework subsumes Action-Contingent Noiselessly Observable Markov Decision Processes (ACNO-MDPs) (Nam et al., 2021), where observations must be explicitly purchased, and *intermittent feedback* models (Hausknecht & Stone, 2015), where unreliable sensors or communication channels yield only sporadic state information. Rather than focusing on any particular observation pattern, we analyze ATST-MDPs in full generality.

**Contributions.** We develop theoretical foundations for RL with action-triggered observations. Our contributions are summarized as follows.

- We formalize ATST-MDPs and establish Bellman optimality equations on the augmented state space. We further introduce an action-sequence reformulation that circumvents the exponential complexity of augmented

[1]Department of Statistical Sciences, University of Toronto, Toronto, Canada [2]Vector Institute, Toronto, Canada. Correspondence to: Alexander Ryabchenko <alex.rbch.research@gmail.com>, Wenlong Mou <wenlong.mou@utoronto.ca>.

*Proceedings of the 43rd International Conference on Machine Learning*, Seoul, South Korea. PMLR 306, 2026. Copyright 2026 by the author(s).

states, making it compatible with existing learning methods (Sections 2–3).

- Under the linear MDP assumption, we construct an action-sequence feature map $\psi$ such that the action-sequence value-function is linear in $\psi$. We further develop data-driven estimators for such feature maps and establish a non-asymptotic sample complexity guarantee of $\widetilde{O}\left(\frac{d^3}{\varepsilon^2(1-\gamma)^2}\right)$ (Section 4).

- Leveraging the Bellman equations and linear representation, we propose Action-Triggered Sporadically Traceable Least-Squares Value Iteration with Upper Confidence Bounds (ATST-LSVI-UCB), an optimistic value iteration algorithm for episodic learning in linear ATST-MDPs. Given an $\varepsilon$-admissible feature map estimate, the algorithm achieves $\widetilde{O}(\sqrt{d^3 K(1-\gamma)^{-3}})$ regret, compared to the optimal policy under the same observation constraints. This result matches the optimal rate for standard linear MDPs (Section 5).

- We validate ATST-LSVI-UCB empirically on several simulation environments, exploring how varying observation frequencies interact with task structure in ATST-MDPs (Section 6).

**Related work.** ATST-MDPs overlap with several well-studied settings, yet none directly captures action-triggered observations. Although the absence of state feedback superficially resembles *RL with observation delays* (Katsikopoulos & Engelbrecht, 2003; Walsh et al., 2009), the delays in ATST-MDPs are endogenous, induced by the agent's actions, whereas classical delays are exogenous. This should not be confused with *RL with episodic delays* (van der Hoeven et al., 2023), where states are immediately observable but rewards are delayed by multiple episodes. Recent work on impaired observability also studies RL with delayed and missing state observations (Chen et al., 2023); ATST-MDPs differ in that observation events are triggered by the chosen actions. *Goal-conditioned RL* (Schaul et al., 2015; Andrychowicz et al., 2017) typically assumes full state observability with sparse goal-dependent reward or success signals, whereas ATST-MDPs study action-triggered sparsity in state observations. Many POMDP formulations (Pineau et al., 2003; Silver & Veness, 2010) model belief updates under partial observability; however, existing work generally does not exploit the structure induced by action-triggered observations. For additional related work, see Appendix A.

## 2. Problem Setting

**Action-Triggered Sporadically Traceable Markov Decision Processes (ATST-MDPs).** The ATST-MDP framework extends standard MDPs by introducing the action-triggered state observation mechanism. Formally, we define an ATST-MDP as a 6-tuple $(\mathcal{S}, \mathcal{A}, \mathbb{P}, r, \gamma, \beta)$, consisting of

a measurable state space $\mathcal{S}$, a finite action space $\mathcal{A}$, a transition kernel $\mathbb{P}(\cdot|s, a) \in \Delta_{\mathcal{S}}$, a deterministic reward function $r : \mathcal{S} \times \mathcal{A} \to [0, 1]$, a discount factor $\gamma \in (0, 1)$, and a state observation probability function $\beta : \mathcal{A} \to [0, 1]$.

The dynamics proceed as follows: when the agent is in state $s$ and executes action $a$, it incurs reward $r(s, a)$ and the environment transitions to a new state $s' \sim \mathbb{P}(.|s, a)$. Crucially, the state $s'$ is not necessarily observed by the agent. With probability $\beta(a)$, the action triggers an observational event we term a ***data-burst***, revealing $s'$ to the agent. Otherwise, with probability $\bar{\beta}(a) = 1 - \beta(a)$, no data-burst occurs and no state feedback is provided. Reward feedback is also linked to data-bursts: at each data-burst, we allow the agent to observe the cumulative reward incurred since the previous data-burst. While specific applications might allow for the revelation of full state-reward trajectories at data-bursts, this work addresses the general setting where only aggregated outcomes are periodically measurable.

**Notation.** We write $\varnothing$ for the empty sequence. For an arbitrary set $\mathcal{U}$ and any $n \geq 0$, let $\mathcal{U}^n$ denote the set of length-$n$ sequences over $\mathcal{U}$, so that $\mathcal{U}^0 = \{\varnothing\}$. We identify $\mathcal{U}^1$ with $\mathcal{U}$ by viewing each $u \in \mathcal{U}$ as the length-1 sequence. We write $\mathcal{U}^{\leq l} = \bigcup_{i=0}^{l} \mathcal{U}^i$ for sequences of length at most $l$, $\mathcal{U}^{<\mathbb{N}} = \bigcup_{i=0}^{\infty} \mathcal{U}^i$ for the set of all finite sequences, and $\mathcal{U}^{\mathbb{N}}$ for the set of infinite sequences over $\mathcal{U}$. We use $\oplus$ to denote concatenation (e.g., $u \oplus v = (u, v)$, $(u_1, \ldots, u_n) \oplus v = (u_1, \ldots, u_n, v)$), with $\varnothing$ as the identity so that $u \oplus \varnothing = u$ and $\mathcal{U} \times \{\varnothing\} = \mathcal{U}$. We use the shorthand $[n] = \{1, \ldots, n\}$. For vectors $x \in \mathbb{R}^d$ and matrices $M \in \mathbb{R}^{d \times d}$, $\|x\|_q$ denotes the $\ell_q$-norm, $\|M\|_q$ the induced $\ell_q$-operator norm, and $\lambda_{\min}(M)$, $\rho(M)$ the minimum eigenvalue and spectral radius of $M$, respectively. For $a \in \mathcal{A}$, we write $\beta_a = \beta(a)$ and $\bar{\beta}_a = 1 - \beta_a$.

## 3. ATST-MDPs as Decision Processes on the Augmented State Space

Since the state is revealed only at data-bursts, the agent acts under partial observability. As in POMDPs, the agent must base decisions on its observation history rather than the current state. In a POMDP, the posterior over the latent state (the belief) is a sufficient statistic for the history, so optimal decision rules can be taken to depend on the belief alone (Kaelbling et al., 1998). In our setting, the information relevant to the current state is fully captured by the last observed state together with the sequence of actions taken since that observation. Following the construction for delayed-observation MDPs (Walsh et al., 2009), we formalize this via the ***augmented state space*** $\mathcal{X} = \mathcal{S} \times \mathcal{A}^{<\mathbb{N}}$, defining the agent's augmented state as the last observed environmental state together with the (possibly empty) sequence of actions taken since.

Each augmented state $x = (s_1, a_1, \ldots, a_n) \in \mathcal{X}$ corresponds to a belief distribution $b(\cdot|x) \in \Delta_{\mathcal{S}}$: the marginal over the state obtained by starting from $s_1$ and executing the action-sequence $(a_1, \ldots, a_n)$ of length $n \geqslant 0$. For $n = 0$, $x = s_1 \in \mathcal{S}$ and $b(\cdot|x) = \delta_{s_1}$. For $n \geqslant 1$, it is obtained by marginalizing over the unobserved trajectory:

$$b(\cdot|x) = \int_{\mathcal{S}^{n-1}} \mathbb{P}(\cdot|s_n, a_n) \prod_{i=1}^{n-1} \mathbb{P}(s_{i+1}|s_i, a_i)\, ds_i. \quad (1)$$

Thus, in direct analogy to belief states in POMDPs, the augmented state $x \in \mathcal{X}$ serves as a sufficient statistic for control in ATST-MDPs: one can view the interaction as a fully observed decision process evolving on $\mathcal{X}$. Concretely, from augmented state $x$ with true state $s$, executing action $a$ transitions the environment to $s' \sim \mathbb{P}(\cdot \mid s, a)$ and updates the augmented state to either $x' = s'$ with probability $\beta(a)$ or $x' = x \oplus a$ with probability $\bar{\beta}(a)$. The induced transition kernel on $\mathcal{X}$ is $\mathbb{P}_{\mathcal{X}}(\cdot|x, a) = \beta(a) \cdot b(\cdot|x \oplus a) + \bar{\beta}(a) \cdot \delta_{x \oplus a}$, a mixture of the belief over $\mathcal{S}$ and a point mass at $x \oplus a$.

However, unlike general POMDPs, ATST-MDPs possess a special structure: the augmented state evolves like a renewal process, growing in length until a data-burst resets it to a singleton in $\mathcal{S}$. Trajectories therefore decompose into intervals between successive observations, admitting a simpler representation grounded in $\mathcal{S}$ rather than the full $\mathcal{X}$.

We analyze this process on $\mathcal{X}$ through ***augmented policies*** $\pi : \mathcal{X} \to \mathcal{A}$. The following subsections establish the Bellman equation on $\mathcal{X}$, prove existence of optimal augmented policies, and then introduce an alternative formulation that exploits the interval structure induced by data-bursts.

### 3.1. Value-Functions and Bellman Optimality

For any augmented policy $\pi : \mathcal{X} \to \mathcal{A}$, we define the action value-function $Q^\pi : \mathcal{X} \times \mathcal{A} \to [0, \frac{1}{1-\gamma}]$ as the expected cumulative discounted reward when starting from augmented state $x \in \mathcal{X}$ (with latent initial state $s_1 \sim b(\cdot|x)$), executing action $a$, and following policy $\pi$ thereafter. Formally:

$$Q^\pi(x, a) = \mathbb{E}\left[r(s_1, a) + \sum_{h=2}^{\infty} \gamma^{h-1} r(s_h, \pi(x_h))\right],$$

where $x_1 = x$, $a_1 = a$, and the expectation is taken over trajectories generated by $s_1 \sim b(.|x)$, $s_{h+1} \sim \mathbb{P}(.|s_h, a_h)$, $x_{h+1} \sim \begin{cases} s_{h+1} & \text{with probability } \beta(a_h) \\ x_h \oplus a_h & \text{otherwise} \end{cases}$.

The state value-function $V^\pi : \mathcal{X} \to [0, \frac{1}{1-\gamma}]$ is defined accordingly as the expected cumulative discounted reward when starting from augmented state $x$ and following policy $\pi$ thereafter, i.e., $V^\pi(x) = Q^\pi(x, \pi(x))$. The following theorem establishes the Bellman equation for these value functions, shows that the associated Bellman operator is a contraction, and guarantees existence of an optimal policy.

**Theorem 3.1** (Augmented Bellman Optimality). *Let* $\mathcal{M} = (\mathcal{S}, \mathcal{A}, \mathbb{P}, r, \gamma, \beta)$ *be an ATST-MDP with augmented state*

space $\mathcal{X} = \mathcal{S} \times \mathcal{A}^{<\mathbb{N}}$ *and consider the set of measurable functions* $\mathcal{V} = \{V : \mathcal{X} \to [0, \frac{1}{1-\gamma}]\}$.

- ***(Policy Evaluation)*** *For any policy* $\pi : \mathcal{X} \to \mathcal{A}$,

$$Q^\pi(x, a) = \mathbb{E}_{s \sim b(\cdot|x)}\left[r(s, a)\right]$$
$$+ \gamma\beta(a) \mathbb{E}_{s' \sim b(\cdot|x \oplus a)}\left[V^\pi(s')\right] + \gamma\bar{\beta}(a) V^\pi(x \oplus a).$$

- ***(Contraction)*** *The Bellman operator* $\mathbb{T} : \mathcal{V} \to \mathcal{V}$ *given by*

$$\mathbb{T}V(x) := \max_{a \in \mathcal{A}}\Big\{ \mathbb{E}_{s \sim b(\cdot|x)}[r(s, a)]$$
$$+ \gamma\beta(a) \mathbb{E}_{s' \sim b(\cdot|x \oplus a)}[V(s')] + \gamma\bar{\beta}(a) V(x \oplus a)\Big\},$$

*is a* $\gamma$-contraction on $(\mathcal{V}, \|.\|_\infty)$.

- ***(Optimality)*** *There exists an optimal augmented policy* $\pi^* : \mathcal{X} \to \mathcal{A}$ *achieving* $V^*(x) = \sup_\pi V^\pi(x)$ *for all* $x \in \mathcal{X}$, *where* $V^*$ *is the unique fixed point of* $\mathbb{T}$.

See Appendix B for the proof. This theorem guarantees existence of an optimal augmented policy, providing a well-defined objective for learning. The Bellman equations also serve as the foundation of our algorithms.

### 3.2. From Augmented States to action-sequences

The augmented state space $\mathcal{X} = \mathcal{S} \times \mathcal{A}^{<\mathbb{N}}$ branches over all possible action histories, yet any augmented policy $\pi$ traverses only a single branch from each observed state $s$: the sequence $a_1 = \pi(s)$, $a_2 = \pi(s; a_1)$, $a_3 = \pi(s; a_1, a_2)$, and so on, executed until the next data-burst. Since data-bursts fully reveal the state, this motivates reformulating ATST-MDPs as decision processes on $(\mathcal{S}, \mathcal{A}^{\mathbb{N}})$, where at each data-burst the agent observes $s \in \mathcal{S}$ and commits to an action-sequence $\boldsymbol{a} \in \mathcal{A}^{\mathbb{N}}$ to execute until the next observation.

Concretely, we define the *action-sequence value-function* $K^\pi : \mathcal{S} \times \mathcal{A}^{\mathbb{N}} \to [0, \frac{1}{1-\gamma}]$ as the expected discounted reward when starting from state $s$, executing $\boldsymbol{a} = (a_1, a_2, \ldots)$ until the next data-burst, and following $\pi$ thereafter. Let $T_{\text{DB}} \in \mathbb{N} \cup \{\infty\}$ denote the (random) index of the first action that triggers a data-burst. We have

$$K^\pi(s, \boldsymbol{a}) = \mathbb{E}\left[\sum_{h=1}^{T_{\text{DB}}} \gamma^{h-1} r(s_h, a_h) + \gamma^{T_{\text{DB}}} V^\pi(s_{T_{\text{DB}}+1})\right], \quad (2)$$

where the first term is the discounted reward accumulated until observation and the second is the discounted continuation value from the revealed state (0 if $T_{\text{DB}} = \infty$). Introducing the shorthand notation

$$R(s, \boldsymbol{a}) = \mathbb{E}\left[\sum_{h=1}^{T_{\text{DB}}} \gamma^{h-1} r(s_h, a_h)\right],$$
$$\mathbb{P}V(s, \boldsymbol{a}) = \mathbb{E}\left[\gamma^{T_{\text{DB}}} V(s_{T_{\text{DB}}+1})\right],$$

we arrive at the equation $K^\pi = R + \mathbb{P}V^\pi$, which separates the pre-observation discounted reward $R$ from the continuation operator $\mathbb{P}$.

On the other hand, the following result relates action-sequence value function $K^\pi$ back to $Q^\pi$ and $V^\pi$.

**Proposition 3.2.** *For any augmented policy $\pi : \mathcal{X} \to \mathcal{A}$, let $\boldsymbol{a}^\pi : \mathcal{X} \to \mathcal{A}^\mathbb{N}$ denote the induced action-sequence map, where $\boldsymbol{a}^\pi(x) = (\pi(x), \pi(x \oplus \pi(x)), \dots)$. Then, for all $s \in \mathcal{S}$ and $a \in \mathcal{A}$: $Q^\pi(s, a) = K^\pi(s, a \oplus \boldsymbol{a}^\pi(s \oplus a))$ and $V^\pi(s) = K^\pi(s, \boldsymbol{a}^\pi(s))$.*

Consequently, the value functions under two formulations are connected. Together with the Bellman equations established in the previous section, this gives fixed-point equations analogous to those used in standard value-learning methods. In conjunction with the linear MDP assumption in Section 4, this facilitates the design of model-free learning algorithms.

The action-sequence reformulation exploits the renewal structure of ATST-MDPs: data-bursts reset the agent to a known state, decomposing the problem into intervals between consecutive data-bursts. This key observation distinguishes ATST-MDPs from general POMDPs, allowing for efficient learning algorithms. Crucially, while $\mathcal{A}^\mathbb{N}$ remains infinite, this structure admits a tractable linear representation, as developed in the next section.

# 4. Linearity Enables Efficient Representation

The action-sequence formulation casts ATST-MDPs as decision processes whose actions are infinite sequences in $\mathcal{A}^\mathbb{N}$. While this formulation is conceptually clean, evaluating value-functions $K^\pi$ remains challenging without additional structure, as it involves infinitely long action-sequences.

To address this issue, in this section, we leverage the linear MDP structure, a widely-used assumption in the RL theory literature. In the linear MDP setting, the value function $K^\pi$ admits a finite-dimensional representation amenable to regression-based methods.

**Assumption 4.1** (Linear MDP, (Jin et al., 2020b))**.** There exists a feature map $\phi : \mathcal{S} \times \mathcal{A} \to \mathbb{R}^d$ such that:

$$\mathbb{P}(\cdot|s, a) = \langle \phi(s, a), \boldsymbol{\mu}(\cdot) \rangle, \quad r(s, a) = \langle \phi(s, a), \boldsymbol{\theta} \rangle,$$

where $\boldsymbol{\mu} : \mathcal{S} \to \mathbb{R}^d$ consists of $d$ finite signed measures over $\mathcal{S}$ and $\boldsymbol{\theta} \in \mathbb{R}^d$. Additionally, it holds that $\sup_{s,a} \|\phi(s, a)\|_2 \leqslant 1$, $\|\boldsymbol{\theta}\|_2 \leqslant \sqrt{d}$, and $\||\boldsymbol{\mu}|(\mathcal{S})\|_2 \leqslant \sqrt{d}$.

For learning problems, it is often assumed that the feature map $\phi$ is known and the parameters $(\boldsymbol{\mu}, \boldsymbol{\theta})$ are unknown. In recent literature, the linear MDP model has emerged as a standard testbed for RL algorithms under function approximation, while it also covers the classical tabular setting.

*Remark* 4.2 (Tabular MDPs). Assumption 4.1 subsumes finite (tabular) MDPs: taking the feature map $\phi$ as one-hot encoding of state-action pairs yields a valid linear representation (Jin et al., 2020b). Therefore, substituting $d = |\mathcal{S}||\mathcal{A}|$

into our bounds and constructions recovers the corresponding tabular guarantees.

Under the linear MDP assumption, we can construct the ***action-sequence feature map*** $\psi : \mathcal{S} \times \mathcal{A}^\mathbb{N} \to \mathbb{R}^{2d}$ such that the action-sequence value-function $K^\pi$ is linear in $\psi$ for every augmented policy $\pi$ (Theorem 4.4). We further provide data-driven methods that estimate the feature map $\psi$ efficiently, with non-asymptotic guarantees (Theorem 4.6).

## 4.1. Linearity of Action-Sequence Value Functions

In this section, we establish the linearity of action-sequence value functions by extending the linear MDP structures.

We first define the action-matrix $M_a = \int_\mathcal{S} \boldsymbol{\mu}(s)\phi(s, a)^\top ds$. This matrix captures how features evolve under action $a$: for $(s, a, a') \in \mathcal{S} \times \mathcal{A}^2$, we have

$$\mathbb{E}_{s' \sim \mathbb{P}(\cdot|s,a)}[\phi(s', a')^\top] = \phi(s, a)^\top M_{a'}.$$

Products of action-matrices thus propagate features across consecutive actions.

Based on this definition, we can extend the mapping $\phi : \mathcal{S} \times \mathcal{A} \to \mathbb{R}^d$ to augmented state $x = (s_1, a_1, \dots, a_n) \in \mathcal{X} \backslash \mathcal{S}$ by $\phi(x)^\top = \phi(s_1, a_1)^\top \prod_{i=2}^n M_{a_i}$. The following lemma provides a linear representation of the belief state.

**Lemma 4.3** (Linearity of belief)**.** *Under the above setup, for any augmented state $x \in \mathcal{X} \backslash \mathcal{S}$, we have*

$$b(\cdot|x) = \langle \phi(x), \boldsymbol{\mu}(\cdot) \rangle \quad and \quad \|\phi(x)\|_2 \leqslant 1.$$

*Moreover, $\mathbb{E}_{s \sim b(\cdot|x)}[r(s, a)] = \langle \phi(x \oplus a), \boldsymbol{\theta} \rangle$, and for every measurable function $V : \mathcal{S} \to [0, 1/(1 - \gamma)]$, it holds that $\mathbb{E}_{s' \sim b(\cdot|x \oplus a)}[V(s')] = \langle \phi(x \oplus a), \boldsymbol{v} \rangle$, where $\boldsymbol{v} = \int V(s) d\boldsymbol{\mu}(s)$ satisfies $\|\boldsymbol{v}\|_2 \leqslant \frac{\sqrt{d}}{1-\gamma}$.*

Lemma 4.3 reduces expectations of rewards and value-functions under beliefs to inner products in $\mathbb{R}^d$. To extend this to the action-sequence value function $K^\pi$ (Section 3.2), we construct a new feature map $\psi : \mathcal{S} \times \mathcal{A}^\mathbb{N} \to \mathbb{R}^{2d}$. For $s \in \mathcal{S}$, $a \in \mathcal{A}$, and $\boldsymbol{a} = (a_1, a_2, \dots) \in \mathcal{A}^\mathbb{N}$, we define its value on $(s, a \oplus \boldsymbol{a}) \in \mathcal{S} \times \mathcal{A}^\mathbb{N}$ as

$$\psi(s, a \oplus \boldsymbol{a})^\top = \tfrac{1}{2} \phi(s, a)^\top \left( \beta_a I_{1,2} + \bar\beta_a M_{1,2}(\boldsymbol{a}) \right), \quad (3)$$

where $I_{1,2} = \left[ {}_{(1-\gamma)I_d} \; {}_{\gamma I_d} \right]$, $M_{1,2}(\boldsymbol{a}) = \left[ {}_{(1-\gamma)M_1(\boldsymbol{a})} \; {}_{\gamma M_2(\boldsymbol{a})} \right]$, and matrices $M_1(\boldsymbol{a}), M_2(\boldsymbol{a}) \in \mathbb{R}^{d \times d}$ are given by:

$$M_1(\boldsymbol{a}) = I_d + \sum_{k=1}^\infty \gamma^k (\prod_{i=1}^{k-1} \bar\beta_{a_i})(\prod_{i=1}^k M_{a_i}), \quad (4a)$$

$$M_2(\boldsymbol{a}) = \sum_{k=1}^\infty \gamma^k (\prod_{i=1}^{k-1} \bar\beta_{a_i}) \beta_{a_k} (\prod_{i=1}^k M_{a_i}). \quad (4b)$$

Under this construction, the function $R$ is linear in the first $d$ coordinates of $\psi$, and $\mathbb{P}V^\pi$ is linear in the latter $d$ coordinates. Putting them together, we can establish the linearity of $K^\pi = R + \mathbb{P}V^\pi$ in $\psi$.

**Theorem 4.4** (Linearity of $K^\pi$). *For any policy $\pi : \mathcal{X} \to \mathcal{A}$, let $\boldsymbol{v}^\pi = \int_{\mathcal{S}} V^\pi(s) d\boldsymbol{\mu}(s)$ and $\boldsymbol{v}_{1,2}^\pi = 2\begin{bmatrix} \boldsymbol{\theta}/(1-\gamma) \\ \boldsymbol{v}^\pi \end{bmatrix} \in \mathbb{R}^{2d}$. Then, $\|\boldsymbol{v}_{1,2}^\pi\|_2 \leqslant \frac{4\sqrt{d}}{1-\gamma}$ and for every $s \in \mathcal{S}$ and $\boldsymbol{a} \in \mathcal{A}^{\mathbb{N}}$:*

$$K^\pi(s, \boldsymbol{a}) = \langle \boldsymbol{\psi}(s, \boldsymbol{a}), \boldsymbol{v}_{1,2}^\pi \rangle \quad and \quad \|\boldsymbol{\psi}(s, \boldsymbol{a})\|_2 \leqslant 1.$$

Although $K^\pi$ is defined on the infinite-dimensional space $\mathcal{S} \times \mathcal{A}^{\mathbb{N}}$, it is fully characterized by an inner product in $\mathbb{R}^{2d}$ between a bounded feature map and a policy-dependent vector. Thus, given access to $\boldsymbol{\psi}$, regression-based methods can be used to approximate $K^* = K^{\pi^*}$ and recover $V^*(s) = \sup_{\boldsymbol{a} \in \mathcal{A}^{\mathbb{N}}} K^*(s, \boldsymbol{a})$, as we demonstrate for episodic learning in Section 5.

### 4.2. Estimation of the Action-Sequence Feature Map

Unlike learning in fully observed linear MDP environments, for ATST-MDPs, the feature maps $(\boldsymbol{\phi}, \boldsymbol{\psi})$ are unknown in general, as they depend on transition dynamics of the underlying MDP. In this section, we study algorithms and guarantees for these estimation problems.

According to Lemma 4.3 and Theorem 4.4, the feature maps $\boldsymbol{\phi}$ and $\boldsymbol{\psi}$ are determined by the action-matrices $\{M_a\}_{a \in \mathcal{A}}$ and observation probabilities $\{\beta_a\}_{a \in \mathcal{A}}$. Given estimates $\{\widehat{M}_a, \widehat{\beta}_a\}_{a \in \mathcal{A}}$ (with $\widehat{\beta}_a = \beta_a$ if known), it is natural to define $\widehat{\boldsymbol{\psi}}$ via (3)–(4) by substituting $\widehat{M}_a$ for $M_a$ and $\widehat{\beta}_a$ for $\beta_a$. [1] This raises two questions: how does the estimation error of $\{M_a, \beta_a\}_{a \in \mathcal{A}}$ affect the feature maps, and can we estimate these quantities accurately from data?

We address both below: Theorem 4.6 quantifies how estimation error propagates to $\boldsymbol{\psi}$, and Corollary 4.9 establishes that $\widetilde{O}(d^3/\epsilon^2(1-\gamma)^2)$ exploratory transitions suffice. This is substantially easier than recovering the full transition kernel $\boldsymbol{\mu}$, which consists of $d$ latent measures with no assumed parametric form. Notably, the estimation is reward-free: once the feature map is obtained, it applies to any reward function. This contrasts with "observe before planning" approaches (Nam et al., 2021; Guo et al., 2016), which estimate full transition dynamics and rewards during exploration.

#### 4.2.1. PLUG-IN ESTIMATION FOR FEATURE MAPS

We first formalize what constitutes a good approximation of $\boldsymbol{\psi}$: a uniform approximation error bound, bounded norm, and continuity over $\mathcal{A}^{\mathbb{N}}$.

**Definition 4.5.** For $\epsilon \geqslant 0$, a function $\boldsymbol{\psi}' : \mathcal{S} \times \mathcal{A}^{\mathbb{N}} \to \mathbb{R}^{2d}$ is an $\epsilon$-*admissible approximation* of $\boldsymbol{\psi}$ if it holds that: $\sup_{s, \boldsymbol{a}} \|(\boldsymbol{\psi}' - \boldsymbol{\psi})(s, \boldsymbol{a})\|_2 \leqslant \epsilon$, $\sup_{s, \boldsymbol{a}} \|\boldsymbol{\psi}'(s, \boldsymbol{a})\|_2 \leqslant 1$, and $\boldsymbol{\psi}'(s, .)$ is continuous with respect to the product topology on $\mathcal{A}^{\mathbb{N}}$ for every $s \in \mathcal{S}$.

---

[1] The observation probabilities $\{\beta_a\}_{a \in \mathcal{A}}$ are often known in practice (see e.g. Nam et al. (2021)). In such a case, we simply let $\beta_a = \widehat{\beta}_a$ for every $a \in \mathcal{A}$.

The following result establishes that uniform convergence of $\widehat{M}_a$ and $\widehat{\beta}_a$ to their true values yields an admissible approximation with controlled error.

**Theorem 4.6.** *Suppose estimates $\{\widehat{M}_a, \widehat{\beta}_a\}_{a \in \mathcal{A}}$ satisfy $\sup_{a \in \mathcal{A}} \|\widehat{M}_a - M_a\|_2 \leqslant \varepsilon$ and $\sup_{a \in \mathcal{A}} |\widehat{\beta}_a - \beta_a| \leqslant \varepsilon_\beta$ for some $\varepsilon \in [0, \frac{1-\gamma}{2\sqrt{d}}]$ and $\varepsilon_\beta \in [0, 1]$. Then,*

$$\sup_{(s, \boldsymbol{a}) \in \mathcal{S} \times \mathcal{A}^{\mathbb{N}}} \|(\widehat{\boldsymbol{\psi}} - \boldsymbol{\psi})(s, \boldsymbol{a})\|_2 \leqslant \frac{16d}{1-\gamma}(\varepsilon + \varepsilon_\beta/\sqrt{d}).$$

*Moreover, the function $\widetilde{\boldsymbol{\psi}}(s, \boldsymbol{a}) = \frac{\widehat{\boldsymbol{\psi}}(s, \boldsymbol{a})}{1 + 16d(\varepsilon + \varepsilon_\beta/\sqrt{d})/(1-\gamma)}$ is a $\frac{32d(\varepsilon + \varepsilon_\beta/\sqrt{d})}{1-\gamma}$-admissible approximation of $\boldsymbol{\psi}$.*

See Appendix C.2 for its proof. Theorem 4.6 guarantees admissibility of feature map estimation using action-matrix and observation-probability estimators with small errors. In the next subsection, we show that such errors are achievable.

#### 4.2.2. ESTIMATION OF $\{M_a, \beta_a\}_{a \in \mathcal{A}}$

Let $\mathcal{D} \in \Delta_{\mathcal{S} \times \mathcal{A}}$ be an exploratory distribution over state–action pairs with a feature-induced second-moment matrix $\Sigma = \mathbb{E}_{(s, a) \sim \mathcal{D}}[\boldsymbol{\phi}(s, a)\boldsymbol{\phi}(s, a)^\top]$, and assume $\lambda_{\min}(\Sigma) > 0$, ensuring that $\mathcal{D}$ explores all feature directions. When $\beta$ is unknown, assume $p_{\min} = \inf_{a' \in \mathcal{A}} \mathbb{P}_{(s, a) \sim \mathcal{D}}(a = a') > 0$, so that each action is sampled with positive probability.

We draw $N$ independent samples, each consisting of $(s, a) \sim \mathcal{D}$, a next state $s' \sim \mathbb{P}(\cdot|s, a)$, an observation indicator $b \sim \text{Ber}(\beta_a)$. Given the dataset $\{s_n, a_n, s'_n, b_n\}_{n=1}^N$, we estimate action-matrices via ridge regression and observation probabilities via empirical means:

$$\widehat{M}_a = (X^\top X + I_d)^{-1} X^\top Y_a, \quad \widehat{\beta}_a = \frac{\sum_{n=1}^N b_n \mathbb{I}(a_n = a)}{\max\{\sum_{n=1}^N \mathbb{I}(a_n = a), 1\}},$$

where rows of $X, Y_a \in \mathbb{R}^{N \times d}$ are $\boldsymbol{\phi}(s_n, a_n)$ and $\boldsymbol{\phi}(s'_n, a)$ respectively. These estimators achieve an $O(N^{-\frac{1}{2}})$ rate.

**Lemma 4.7.** *There exists an absolute constant $C \geqslant 1$ such that for all $p \in (0, 1)$ and $N \geqslant \frac{4C^2 d \log(2Ad/p)}{\lambda_{\min}(\Sigma)^2}$, ridge estimators $\widehat{M}_a$ satisfy $\mathbb{P}\left(\sup_{a \in \mathcal{A}} \|\widehat{M}_a - M_a\|_2 \leqslant 4C\sqrt{\frac{d \log(2Ad/p)}{N\lambda_{\min}(\Sigma)^2}}\right) \geqslant 1 - p$.*

**Lemma 4.8.** *For all $p \in (0, 1)$ and $N \geqslant 1$, empirical means $\widehat{\beta}_a$ satisfy $\mathbb{P}\left(\sup_{a \in \mathcal{A}} |\widehat{\beta}_a - \beta_a| \leqslant \sqrt{\frac{12 \ln(3A/p)}{N p_{\min}}}\right) \geqslant 1 - p$.*

**Putting them together.** Combining these lemmas with Theorem 4.6 yields sample complexity bounds for constructing an $\epsilon$-admissible approximation of $\boldsymbol{\psi}$.

**Corollary 4.9.** *Let $\widetilde{\boldsymbol{\psi}}^{M, \beta}$ denote the normalized feature map $\widetilde{\boldsymbol{\psi}}$ from Theorem 4.6, computed using estimates $\widehat{M}_a$ and $\widehat{\beta}_a$ constructed from $N$ samples, and estimation error bounds $\varepsilon = 4C\sqrt{\frac{d \log(4Ad/p)}{N\lambda_{\min}(\Sigma)^2}}$ and $\varepsilon_\beta = \sqrt{\frac{12 \ln(6A/p)}{N p_{\min}}}$ for*

*C from Lemma 4.7. Similarly, let $\widetilde{\psi}^M$ denote the normalized feature map computed using the true probabilities $\beta_a$ and estimates $\widehat{M}_a$ from $N$ samples, with the same $\varepsilon$ and $\varepsilon_\beta = 0$.*

*There exists an absolute constant $c > 0$ such that for all $p \in (0, 1)$ and $\epsilon \in (0, 1)$, the following holds:*

1. *If $N \geqslant c \cdot \frac{d^3 \log(2Ad/p)}{\epsilon^2(1-\gamma)^2 \min\{\lambda_{\min}(\Sigma)^2, d^2 p_{\min}\}}$, then $\widetilde{\psi}^{M,\beta}$ is $\epsilon$-admissible with probability at least $1 - p$.*

2. *If $N \geqslant c \cdot \frac{d^3 \log(2Ad/p)}{\epsilon^2(1-\gamma)^2 \lambda_{\min}(\Sigma)^2}$, then $\widetilde{\psi}^M$ is $\epsilon$-admissible with probability at least $1 - p$.*

The $\widetilde{O}(\frac{d^3}{\epsilon^2(1-\gamma)^2})$ complexity is polynomial in all problem parameters and independent of $|\mathcal{S}|$, confirming that linear structure enables tractable estimation even in continuous state spaces. Knowledge of $\beta$ significantly affects dependence on the action space: when $\beta$ is known, this dependence is $O(\log|\mathcal{A}|)$, whereas unknown $\beta$ incurs $\widetilde{O}(|\mathcal{A}|)$ since $p_{\min} \leqslant 1/|\mathcal{A}|$. In the tabular case (Remark 4.2), substituting $d = |\mathcal{S}||\mathcal{A}|$ yields cubic complexity $\widetilde{O}(|\mathcal{S}|^3|\mathcal{A}|^3/\epsilon^2)$.

## 5. Episodic Learning and Regret Analysis

We now turn to the problem of episodic learning in unknown systems. Consider an agent interacting with a linear ATST-MDP over $K$ episodes (with the protocol given in Figure 1), where each episode $k$ has random length $H^k \sim \mathrm{Geom}(1 - \gamma)$, a standard reformulation of discounting in which $\gamma$ acts as a continuation probability. The agent observes states and cumulative undiscounted rewards only at data-bursts or episode termination, when the termination symbol $\perp \notin \mathcal{S}$ is returned. At the start of each episode, the agent selects a *burst-dependent policy* $\boldsymbol{\pi}^k = (\pi_u^k)_{u=1}^\infty$, where an augmented policy $\pi_u^k : \mathcal{X} \to \mathcal{A}$ governs behavior between the $(u-1)$-th and $u$-th data-bursts. This allows the policy to change across data-bursts within an episode; the linearity results of Section 4 extend directly, with $V^{\boldsymbol{\pi}}$ and $K^{\boldsymbol{\pi}}$ defined as expected total discounted rewards under this mechanism.

Performance is measured by regret against an optimal augmented policy $\pi^* : \mathcal{X} \to \mathcal{A}$, whose existence was established in Section 3:

$$\mathcal{R}_K = \sum_{k=1}^K \left( V^*(s_1^k) - V^{\boldsymbol{\pi}^k}(s_1^k) \right).$$

This is the natural benchmark: $\pi^*$ operates under the same observation constraints as the agent. Comparison to the (non-augmented) optimal policy for the underlying fully observed MDP (which corresponds to an ATST-MDP with observation probabilities $\beta \equiv 1$) is not meaningful: the performance gap can grow linearly in $K$, since that policy may exploit state information unavailable to the agent.

---

**Episodic Learning under ATST-MDP**

**For** each episode $k = 1, 2, \ldots, K$:
 The environment initializes the cumulative reward $G_0^k = 0$.
 The agent selects a burst-dependent policy $\boldsymbol{\pi}^k$.
 The adversary selects and reveals the initial state $s_1^k$.
 The agent initializes the augmented state as $x_1^k = s_1^k$.

 **For** rounds $h = 1, 2, \ldots$:
 1. The agent plays $a_h^k = \boldsymbol{\pi}^k(x_h^k)$ and incurs the (unobserved) reward $r_h^k = r(s_h^k, a_h^k)$.
    The environment updates $G_h^k = G_{h-1}^k + r_h^k$ and samples $s_{h+1}^k \sim \mathbb{P}(\cdot \mid s_h^k, a_h^k)$.
 2. With probability $1 - \gamma$ (***termination***), the environment reveals $(\perp, G_h^k)$ and ends episode $k$.
 3. With probability $\beta(a_h^k)$ (***data-burst***), the environment reveals $(s_{h+1}^k, G_h^k)$.
 4. The agent sets $x_{h+1}^k = s_{h+1}^k$ if a data-burst occurs; otherwise $x_{h+1}^k = x_h^k \oplus a_h^k$.

*Figure 1.* Execution protocol for an ATST-MDP over $K$ episodes with geometric horizons.

**Feature map access.** Since the action-sequence feature map $\psi$ (3) may be unknown, we assume the agent has access to an $\epsilon$-admissible approximation $\widehat{\psi}$ with known $\epsilon$ (Definition 4.5). By Section 4.2, such an approximation can be efficiently constructed from exploratory data.

### 5.1. Least-Squares Value Iteration for ATST-MDPs

Algorithm 1 adapts Least-Squares Value Iteration with Upper Confidence Bounds (Jin et al., 2020b) to the linear ATST-MDP setting. The algorithm takes as input an $\varepsilon$-admissible approximation $\widehat{\psi}$ of $\psi$ and an effective horizon parameter $H$ that controls both value iteration depth and the amount of history retained.

At episode $k$, the algorithm uses the *effective history* $\mathcal{H}^k = (\mathbf{s}^\tau, \boldsymbol{a}^\tau, R^\tau, \mathbf{s}_N^\tau)_{\tau=1}^{N^k}$, comprising information from the first $H$ data-bursts of each previous episode. Each tuple in $\mathcal{H}^k$ records: an observed state $\mathbf{s}^\tau \in \mathcal{S}$, the action-sequence $\boldsymbol{a}^\tau \in \mathcal{A}^\mathbb{N}$ intended from that state, the undiscounted reward $R^\tau$ accumulated until the next data-burst, and the subsequent observed state $\mathbf{s}_N^\tau \in \mathcal{S} \cup \{\perp\}$. Here $N^k = \sum_{k'=1}^{k-1} \min\{B^{k'}, H\}$, with $B^k$ the number of data-bursts in episode $k$. To ease notation, we write $\widehat{\psi}^\tau = \widehat{\psi}(\mathbf{s}^\tau, \boldsymbol{a}^\tau)$ and $\psi^\tau = \psi(\mathbf{s}^\tau, \boldsymbol{a}^\tau)$.

Each episode, ATST-LSVI-UCB operates in two phases. The *planning phase* performs backward value iteration, computing weights $\boldsymbol{w}_u^k$ that define value-functions $K_u^k : \mathcal{S} \times \mathcal{A}^\mathbb{N} \to [0, (1-\gamma)^{-1}]$. These functions aim to approximate the optimal $K^*(s, \boldsymbol{a}) = \langle \psi(s, \boldsymbol{a}), \boldsymbol{v}_{1,2}^{\pi^*} \rangle$ using the estimated feature map $\widehat{\psi}$, with a UCB bonus encouraging

**Algorithm 1** ATST-LSVI-UCB

**Input:** $\epsilon$-admissible approximation of the action-sequence feature map $\widehat{\psi} : \mathcal{S} \times \mathcal{A}^{\mathbb{N}} \to \mathbb{R}^{2d}$, discount factor $\gamma$.
**Parameters:** effective horizon $H$, regularizers $\lambda$ and $\rho$.

1: **for** episode $k = 1, \dots, K$ **do**
2:  — **Planning phase: backward value iteration** —
3:  Compile history $\mathcal{H}^k = (\mathbf{s}^\tau, \boldsymbol{a}^\tau, R^\tau, \mathbf{s}_N^\tau)_{\tau=1}^{N^k}$.
4:  Set $\Lambda^k = \lambda I + \sum_{\tau=1}^{N^k} \widehat{\psi}^\tau (\widehat{\psi}^\tau)^\top$ and initialize $K_u^k(s, \boldsymbol{a}) = \frac{1}{1-\gamma}$ for $(s, \boldsymbol{a}) \in \mathcal{S} \times \mathcal{A}^{\mathbb{N}}$ and $u \geqslant H$.
5:  **for** $u = H - 1, \dots, 1$ **do**
6:      Set $\boldsymbol{w}_u^k = (\Lambda^k)^{-1} \sum_{\tau=1}^{N^k} \widehat{\psi}^\tau \mathcal{V}_u^{\tau,k}$ with values $\mathcal{V}_u^{\tau,k} = \min\{R^\tau, H\} + \max_{\boldsymbol{a} \in \mathcal{A}^{\mathbb{N}}} K_{u+1}^k(\mathbf{s}_N^\tau, \boldsymbol{a})$.
7:      Set $K_u^k(s, \boldsymbol{a}) = \min\{\widehat{\psi}_{s,\boldsymbol{a}}^\top \boldsymbol{w}_u^k + \rho \|\widehat{\psi}_{s,\boldsymbol{a}}\|_{\Lambda_{\text{inv}}^k}, \frac{1}{1-\gamma}\}$, where $\widehat{\psi}_{s,\boldsymbol{a}} = \widehat{\psi}(s, \boldsymbol{a})$ and $\Lambda_{\text{inv}}^k = (\Lambda^k)^{-1}$.
8:  **end for**

9:  — **Execution phase: burst-to-burst rollouts** —
10:  Set $u = 1$ and receive initial state $\mathbf{s}_1^k$.
11:  **while** episode $k$ continues **do**
12:      Choose $\boldsymbol{a}_u^k \in \arg\max_{\boldsymbol{a} \in \mathcal{A}^{\mathbb{N}}} K_u^k(\mathbf{s}_u^k, \boldsymbol{a})$.
13:      Execute actions from $\boldsymbol{a}_u^k$ until either:
14:          (1) **data-burst:** receive $\mathbf{s}_{u+1}^k \in \mathcal{S}$ and $R_u^k$;
15:          (2) **termination:** receive $\mathbf{s}_{u+1}^k = \perp$ and $R_u^k$.
16:      Set $u \leftarrow u + 1$ and **break** if the episode terminated.
17:  **end while**
18: **end for**

exploration. The *execution phase* follows the greedy policy, selecting action-sequences $\boldsymbol{a}_u^k$ that maximize $K_u^k(\mathbf{s}_u^k, \cdot)$ and executing them until the next data-burst.

**Optimization over $\mathcal{A}^{\mathbb{N}}$.** Lines 7 and 12 require solving $\max_{\boldsymbol{a} \in \mathcal{A}^{\mathbb{N}}} K_u^k(s, \boldsymbol{a})$. Despite the infinite dimensionality of $\mathcal{A}^{\mathbb{N}}$, this reduces to optimizing a continuous function over $\{\widehat{\psi}(s, \boldsymbol{a}) : \boldsymbol{a} \in \mathcal{A}^{\mathbb{N}}\}$, a compact subset of $\mathbb{R}^{2d}$ when $\widehat{\psi}$ is $\varepsilon$-admissible, guaranteeing existence of a maximizer. Our analysis assumes access to an optimization oracle.

In practice, when $\gamma$ is bounded away from 1 and $\beta_{\min} = \min_{a \in \mathcal{A}} \beta(a) > 0$, distant actions have exponentially decaying influence, enabling approximation via horizon truncation: optimizing over $\mathcal{A}^L$ contributes $O((\gamma(1 - \beta_{\min}))^L)$ to approximation error. Alternatively, one may restrict optimization to a structured class of action-sequences, such as eventually periodic sequences. In Section 6, we demonstrate the viability of such restrictions empirically.

**5.2. Theoretical Guarantees**

Given a confidence parameter $p \in (0, 1)$, number of episodes $K$, and an $\epsilon$-admissible feature map $\widehat{\psi}$ with

$\epsilon \leqslant \sqrt{(1 - \gamma)/K}$, we set

$$ H = \lceil \tfrac{\log(K(1-\gamma)^{-1})}{1-\gamma} \rceil + 1, \quad \lambda = 1, \quad \rho = c \cdot dH\sqrt{\iota}, $$

where $\iota = \log(2dKH/p)$ and $c > 0$ is an absolute constant.

We have the following theoretical guarantee.

**Theorem 5.1** (Regret of Algorithm 1). *There exists an absolute constant $c \geqslant 1$ such that, under the above setup, with probability at least $1 - p$, the total regret of Algorithm 1 is*

$$ \widetilde{O}\big(\sqrt{d^3 K (1 - \gamma)^{-3} \iota^2} + d^2 (1 - \gamma)^{-2} \iota $$
$$ + \epsilon \cdot \sqrt{d^2 K^3 (1 - \gamma)^{-5} \iota}\big), $$

*where $\widetilde{O}$ omits polylogarithmic factors independent of $p$.*

The regret bound holds uniformly over all fixed observation probability functions $\beta : \mathcal{A} \to [0, 1]$. Different choices of $\beta$ may change the benchmark $V^*$, since regret is measured against the optimal augmented policy under the same observation constraints, but the stated upper bound remains valid for each such choice.

The proof is in Appendix D. For $\epsilon = O(\frac{1-\gamma}{K\sqrt{d}})$, the third term becomes lower-order and the bound reduces to $\widetilde{O}(\sqrt{d^3 K (1 - \gamma)^{-3}})$, matching the rate for fully observable linear MDPs (Jin et al., 2020b). By Corollary 4.9, the required accuracy $\epsilon \leqslant \frac{1-\gamma}{K\sqrt{d}}$ can be achieved with high probability from $\widetilde{O}(K^2 d^4 / (1 - \gamma)^4)$ exploratory samples; this approximation is reward-free, so once $\widehat{\psi}$ is constructed it applies to any reward function.

## 6. Numerical Experiments

ATST-LSVI-UCB (Algorithm 1) assumes oracle access for optimizing over the infinite action-sequence space $\mathcal{A}^{\mathbb{N}}$. In practice, one may restrict optimization to a finite candidate class $\mathcal{A}^{\text{seq}} \subset \mathcal{A}^{\mathbb{N}}$ when near-optimal policies can be expected to generate sequences from such a class. Eventually periodic sequences with bounded prefix and period lengths are a convenient choice: in many partially observable control tasks, effective behavior consists of a short corrective prefix followed by a repeating stabilization pattern, e.g., classical cartpole balancing (Barto et al., 1983). Moreover, under the linear MDP Assumption 4.1, the feature map $\psi(s, \boldsymbol{a})$ for eventually periodic $\boldsymbol{a} \in \mathcal{A}^{\mathbb{N}}$ admits a closed form via a Neumann-series argument (Lemma C.12).

We evaluate ATST-LSVI-UCB on two tabular ATST-MDP environments (Figures 2–3) with state space $\mathcal{S} = \{s_1, \dots, s_6\}$, action space $\mathcal{A} = \{0, 1\}$, discount factor $\gamma = 0.99$, and uniform observation probabilities $\beta(a) = \beta^* \in \{0.05, 0.1, 0.2, 0.5\}$. We restrict optimization to even-

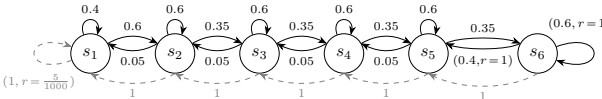

*Figure 2.* **RiverSwim.** Dashed arrows denote transitions for moving left (deterministic); solid arrows for moving right (stochastic).

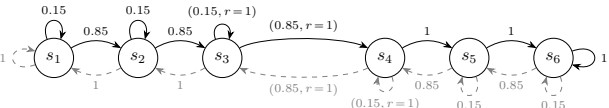

*Figure 3.* **RiverBalance.** Dashed arrows denote transitions for moving left; solid arrows for moving right. Moving away from center is deterministic; toward center is stochastic.

tually periodic sequences of the form

$$\mathcal{A}^{\mathrm{seq}} = \Big\{ \{b\}^P \oplus \big(\{1{-}b\}^{L_1} \oplus \{b\}^{L_2}\big)^\infty : b \in \{0,1\},$$
$$P \in [5], \ L_1, L_2 \in \{0,\dots,10\}, \ L_1{+}L_2 \neq 0 \Big\},$$

i.e., a prefix of $P$ repeated actions followed by an alternating periodic pattern. This yields $|\mathcal{A}^{\mathrm{seq}}| = 1012$ candidate sequences after removing equivalent representations.

**RiverSwim** (Strehl & Littman, 2008) is a standard exploration benchmark consisting of six states arranged in a chain. The agent starts in $s_1$, where a small reward is available, while $s_6$ yields a large reward. Action 1 (right) attempts to move against the current and may fail, whereas action 0 (left) moves with the current and always succeeds. The optimal policy always selects action 1, but the agent must explore extensively against the current to discover this.

**RiverBalance** is a novel variant we introduce that rewards staying near the center, in states $s_3$ and $s_4$. The current pulls the agent away from the center, so the optimal policy must repeatedly steer back and balance actions to remain there.

Figures 4–5 show the average cumulative reward over episodes for ATST-LSVI-UCB with access to the exact action-sequence feature map $\psi$ and effective horizon $H = 100$, for varying observation probabilities $\beta^* \in \{0.05, 0.1, 0.2, 0.5\}$. In **RiverSwim**, all settings converge to the optimal policy, with smaller values of $\beta^*$ converging faster: infrequent observations induce longer open-loop commitments, which encourage early exploration. In **RiverBalance**, all settings improve and then plateau, with larger $\beta^*$ attaining a higher average reward. This is consistent with the fact that sustaining high reward requires state-dependent corrections to steer back toward the rewarding center, which are more effective when observations are more frequent.

The contrasting behaviors highlight a key feature of ATST-MDPs: the role of observation frequency depends on whether near-optimal control requires state-dependent corrections. When it does not (RiverSwim), sparse observations

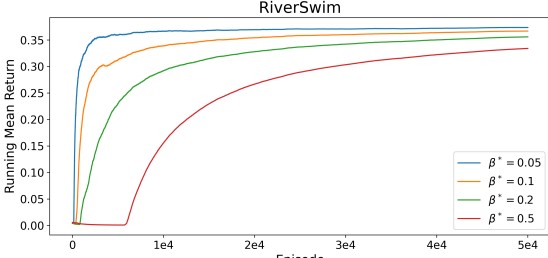

*Figure 4.* **RiverSwim.** $(1{-}\gamma)$-scaled running average of episodic reward versus episode, averaged over 5 simulations. All settings converge to the optimal policy; smaller $\beta^*$ converges faster.

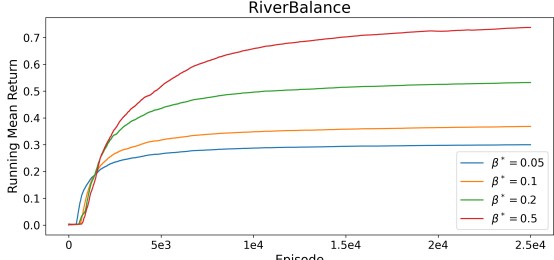

*Figure 5.* **RiverBalance.** $(1{-}\gamma)$-scaled running average of episodic reward versus episode, averaged over 5 simulations. All settings plateau; larger $\beta^*$ achieves higher reward.

can accelerate learning by encouraging longer open-loop commitments. When it does (RiverBalance), more frequent observations enable tighter closed-loop stabilization and higher reward; with sparse observations, performance degrades gracefully and the algorithm still learns a policy that is near-optimal for the corresponding observation regime.

## 7. Discussion and Future Work

This work introduces ATST-MDPs, a novel framework that captures the challenges of reinforcement learning in environments where state observability is action-triggered and sporadic. We derived Bellman optimality equations, showed a linear representation for the induced action-sequence value functions, and provided approximation guarantees for learning the action-sequence feature map from off-policy data. Building on this structure, we proposed ATST-LSVI-UCB and proved low regret for episodic learning with geometric horizons, assuming accurate feature-map estimation.

Several interesting questions remain open for future research. First, ATST-LSVI-UCB assumes access to an optimization oracle over action-sequences. Designing efficient approximation schemes, such as restricting to finite-depth action trees or developing tractable surrogate objectives, would significantly enhance practical applicability. Second, while we establish off-policy methods for estimating action-matrices and data-burst probabilities, a fully online algo-

rithm that adaptively refines these estimates during learning would provide a more robust and practical solution.

Additionally, ATST-MDPs offer a novel perspective on RL with stochastic delays (e.g., Bouteiller et al. (2021)). Classical models treat delays as *exogenous*; here they are *endogenous*, with actions shaping the distribution of observation times. A unifying view would allow *round-dependent* data-burst probabilities $\beta_t(a)$: when $\beta_t$ is action-independent, one recovers some exogenous delay models. Analyzing how different delay-generation mechanisms affect learning and regret presents a promising research direction.

Overall, our results establish a foundation for learning under action-triggered state observations, while the flexibility of our formulation opens pathways toward addressing information constraints across a wide range of sequential decision-making problems.

## Acknowledgements

This work was partially supported by NSERC grant RGPIN-2024-05092 and a Connaught New Researcher Award to WM.

## Impact Statement

This paper presents work whose goal is to advance the field of Machine Learning. There are many potential societal consequences of our work, none of which we feel must be specifically highlighted here.

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

# A. Additional Related Work

**POMDPs and planning under partial observability.** Classical work on decision making with incomplete state information is captured by POMDPs; see the survey of (Kaelbling et al., 1998) and subsequent algorithmic advances such as point-based value iteration (PBVI) (Pineau et al., 2003) and heuristic search value iteration (HSVI) (Smith & Simmons, 2004). Recent progress includes statistical and computational guarantees for learning and planning in partially observed settings (Cai et al., 2022). Much of this line is theoretical and algorithmic, with empirical validations on standard POMDP benchmarks; deep implementations typically combine belief updates with function approximation, but the core guarantees are model-based and non-neural.

**RL with delayed observations (and augmented states).** Early formulations analyze delayed MDPs and augmented-state reductions that stack the last observed state with a queue of intervening actions (Katsikopoulos & Engelbrecht, 2003; Walsh et al., 2009). More recent work examines random delays in deep RL, showing robustness and performance trade-offs under synthetic and real latency processes (Bouteiller et al., 2021), and explores imitation/learning pipelines that must handle delayed feedback (Liotet et al., 2022). This area mixes theory (augmented-state equivalence, stability) with empirical deep RL; implementations often use standard neural agents (e.g., DQN/actor-critic) evaluated under injected delays.

**Paid observations and information acquisition.** Another related line studies decision making when observations incur explicit costs. In RL, agents may choose when to acquire measurements or labels, trading reward for information (Bellinger et al., 2021; Nam et al., 2021; Wang et al., 2025). ATST-MDPs subsume ACNO-MDPs (Nam et al., 2021) as a strict special case: each base action can be replaced by two variants, one with $\beta = 0$ and one with $\beta = 1$, with the observation cost absorbed into the reward. In online learning, closely related "label-efficient" and budgeted feedback models investigate how querying constraints affect regret (Seldin et al., 2014; Audibert & Bubeck, 2010). This area blends theoretical formulations of budgeted information acquisition with empirical demonstrations, with deep implementations appearing mainly in application-driven studies.

**Intermittent observations and unreliable sensing.** A practical motif is intermittently available observations due to sensing/communication failures. Deep Recurrent Q-Learning (DRQN) (Hausknecht & Stone, 2015) tackles partial observability (flickering screen) by replacing feedforward policies with RNNs, showing empirical gains under dropped observations. Subsequent empirical studies examine control with sporadic measurements or packet loss (Klíma et al., 2018). More recent formulations introduce intermittently observable MDPs with modeling/algorithmic structure beyond ad-hoc masking (Chen & Liew, 2026). This line is largely empirical deep RL.

**Active sensing and perception.** Active perception frames sensing as a decision problem: agents select actions that improve informativeness while pursuing task reward. Active-perception POMDPs (Satsangi et al., 2018) formalize this, and recent deep RL approaches study active vision and act-then-measure protocols that interleave task actions with targeted measurements (Shang & Ryoo, 2023; Krale et al., 2023). These works are primarily empirical and use deep neural networks (vision backbones with policy/value heads), sometimes with recurrent modules for memory; theoretical analysis focuses on tractable planning surrogates and approximate belief updates rather than regret.

# B. Augmented Policies: Proofs

In this section, we prove existence of the optimal augmented policy $\pi^* : \mathcal{X} \to \mathcal{A}$. The argument follows by classic application of the Banach fixed-point theorem for the Bellman optimality operator (e.g., see (Puterman, 1994)). First, we restate and prove Theorem 3.1.

**Theorem 3.1** (Restated). *Let $\mathcal{M} = (\mathcal{S}, \mathcal{A}, \mathbb{P}, r, \gamma, \beta)$ be an ATST-MDP with augmented state space $\mathcal{X} = \mathcal{S} \times \mathcal{A}^{<\mathbb{N}}$ and consider the set of measurable functions $\mathcal{V} = \{V : \mathcal{X} \to [0, \frac{1}{1-\gamma}]\}$.*
*(i) (**Policy Evaluation**) For any policy $\pi : \mathcal{X} \to \mathcal{A}$,*

$$Q^\pi(x, a) = \mathbb{E}_{s \sim b(.|x)} \left[ r(s, a) \right] + \gamma \beta(a) \, \mathbb{E}_{s' \sim b(.|x \oplus a)} \left[ V^\pi(s') \right] + \gamma \bar{\beta}(a) \, V^\pi(x \oplus a).$$

*(ii) (**Contraction**) The Bellman operator $\mathbb{T} : \mathcal{V} \to \mathcal{V}$, defined by*

$$\mathbb{T}V(x) = \max_{a \in \mathcal{A}} \left\{ \mathbb{E}_{s \sim b(.|x)}[r(s, a)] + \gamma \beta(a) \, \mathbb{E}_{s' \sim b(.|x \oplus a)}[V(s')] + \gamma \bar{\beta}(a) \, V(x \oplus a) \right\}, \tag{5}$$

*is a $\gamma$-contraction on $(\mathcal{V}, \|.\|_\infty)$.*

*(iii) (Optimality) There exists an optimal augmented policy $\pi^* : \mathcal{X} \to \mathcal{A}$ achieving $V^*(x) = \sup_\pi V^\pi(x)$ for all $x \in \mathcal{X}$, where $V^*$ is the unique fixed point of $\mathbb{T}$.*

*Proof.* We prove each claim in turn.

**(i) Policy Evaluation.** The quantity $Q^\pi(x, a)$ is the expected return when starting from augmented state $x$, taking action $a$, and following $\pi$ thereafter. The term $\mathbb{E}_{s \sim b(.|x)}[r(s, a)]$ captures the expected immediate reward. After executing $a$, the next augmented state depends on whether a data-burst occurs: with probability $\beta(a)$, the next state $s' \sim b(\cdot \mid x \oplus a)$ is observed and the continuation value is $V^\pi(s')$; with probability $\bar\beta(a)$, no new state is observed, the augmented state transitions to $x \oplus a$, and the continuation value is $V^\pi(x \oplus a)$. Taking expectations and discounting by $\gamma$ yields the claimed expression.

**(ii) Contraction.** For any $V \in \mathcal{V}$, define $Q_V : \mathcal{X} \times \mathcal{A} \to [0, \frac{1}{1-\gamma}]$ by

$$Q_V(x, a) = \mathbb{E}_{s \sim b(.|x)}[r(s, a)] + \gamma\beta(a)\, \mathbb{E}_{s' \sim b(.|x \oplus a)}[V(s')] + \gamma\bar\beta(a)\, V(x \oplus a),$$

so that $\mathbb{T}V(x) = \max_a Q_V(x, a)$. Fix arbitrary $V, U \in \mathcal{V}$. For every $x \in \mathcal{X}$,

$$
\begin{aligned}
|\mathbb{T}V(x) - \mathbb{T}U(x)| &= |\max_a Q_V(x, a) - \max_a Q_U(x, a)| \\
&\leqslant \max_a |Q_V(x, a) - Q_U(x, a)| \\
&= \max_a \left|\gamma\beta(a)\, \mathbb{E}_{s' \sim b(.|x \oplus a)}[(V - U)(s')] + \gamma\bar\beta(a)\, (V - U)(x \oplus a)\right| \\
&\leqslant \max_a \left(\gamma\beta(a)\, \|V - U\|_\infty + \gamma\bar\beta(a)\, \|V - U\|_\infty\right) \\
&= \gamma \|V - U\|_\infty .
\end{aligned}
$$

Thus $\mathbb{T}$ is a $\gamma$-contraction on $(\mathcal{V}, \|\cdot\|_\infty)$.

**(iii) Optimality.** By part (i), the function $V^*(x) := \sup_\pi V^\pi(x)$ is a fixed point of $\mathbb{T}$. Part (ii) and the Banach fixed-point theorem imply $V^*$ is the unique such fixed point. Any policy $\pi^* : \mathcal{X} \to \mathcal{A}$ satisfying

$$\pi^*(x) \in \operatorname*{argmax}_{a \in \mathcal{A}} \left\{ \mathbb{E}_{s \sim b(.|x)}[r(s, a)] + \gamma\beta(a)\, \mathbb{E}_{s' \sim b(.|x \oplus a)}[V^*(s')] + \gamma\bar\beta(a)\, V^*(x \oplus a) \right\}$$

for all $x \in \mathcal{X}$ achieves $V^{\pi^*} = V^*$, since $\pi^*$ attains the supremum in the Bellman equation at every augmented state. The existence of such a measurable selector follows from standard arguments (Puterman, 1994).  $\square$

**Proposition 3.2** (Restated). *For any augmented policy $\pi : \mathcal{X} \to \mathcal{A}$, let $\boldsymbol{a}^\pi : \mathcal{X} \to \mathcal{A}^\mathbb{N}$ denote the induced action-sequence map, where $\boldsymbol{a}^\pi(x) = (\pi(x), \pi(x \oplus \pi(x)), \ldots)$. Then, for all $s \in \mathcal{S}$ and $a \in \mathcal{A}$: $Q^\pi(s, a) = K^\pi(s, a \oplus \boldsymbol{a}^\pi(s \oplus a))$ and $V^\pi(s) = K^\pi(s, \boldsymbol{a}^\pi(s))$.*

*Proof.* Fix $\pi : \mathcal{X} \to \mathcal{A}$. For $x \in \mathcal{X}$, the induced action-sequence map can be written in the short recursive form

$$\boldsymbol{a}^\pi(x)_1 = \pi(x), \qquad \boldsymbol{a}^\pi(x)_{h+1} = \pi(x \oplus \boldsymbol{a}^\pi(x)_{1:h}), \quad h \geqslant 1,$$

i.e., it is the sequence of actions chosen by $\pi$ along the branch obtained by appending past actions.

Start from $x_1 = s$ and take $a_1 = a$. On rounds with no data-burst, the augmented state updates as $x_{h+1} = x_h \oplus a_h$, hence $a_{h+1} = \pi(x_{h+1})$. Therefore the executed (infinite) action-sequence is

$$(a_1, a_2, \ldots) = a \oplus \boldsymbol{a}^\pi(s \oplus a).$$

By the definition (2) of $K^\pi(s, \boldsymbol{a})$ as the expected discounted reward from executing $\boldsymbol{a}$ until the first data-burst and then continuing with $\pi$, we get

$$Q^\pi(s, a) = K^\pi\big(s, a \oplus \boldsymbol{a}^\pi(s \oplus a)\big).$$

Finally, $V^\pi(s) = Q^\pi(s, \pi(s))$ and $\boldsymbol{a}^\pi(s) = \pi(s) \oplus \boldsymbol{a}^\pi(s \oplus \pi(s))$, so

$$V^\pi(s) = K^\pi\big(s, \boldsymbol{a}^\pi(s)\big). \qquad \square$$

Additionally, we provide formulas for $R$ and $\mathbb{P}V$, obtained by conditioning on $T_{\mathrm{DB}}$.

**Lemma B.1.** *For all $x \in \mathcal{X}$ and $\boldsymbol{a} \in \mathcal{A}^{\mathbb{N}}$, it holds that*

$$R(x, \boldsymbol{a}) = \sum_{h=1}^{\infty} \gamma^{h-1} \left( \prod_{i=1}^{h-1} \bar{\beta}(a_i) \right) \mathbb{E}_{s \sim b(.|\tilde{x}_h)} [r(s, a_h)],$$

$$\mathbb{P}V(x, \boldsymbol{a}) = \sum_{h=1}^{\infty} \gamma^{h} \left( \prod_{i=1}^{h-1} \bar{\beta}(a_i) \right) \beta(a_h) \, \mathbb{E}_{s' \sim b(.|\tilde{x}_{h+1})} [V(s')].$$

*where $\tilde{x}_h = x \oplus (a_i)_{i=1}^{h-1} \in \mathcal{X}$ for every $h \in \mathbb{N}$.*

*Proof.* Let $\mathbb{P}(.|\boldsymbol{a})$ denote the probability measure of $T_{\mathrm{DB}}$ over $\mathbb{N} \cup \{\infty\}$ when the agent commits to playing the sequence of actions $\boldsymbol{a} \in \mathcal{A}^{\mathbb{N}}$. Then, it holds that $\mathbb{P}(T_{\mathrm{DB}} \geqslant h \mid \boldsymbol{a}) = \prod_{i=1}^{h-1} \bar{\beta}(a_i)$ and $\mathbb{P}(T_{\mathrm{DB}} = h \mid \boldsymbol{a}) = (\prod_{i=1}^{h-1} \bar{\beta}(a_i))\beta(a_h)$.

Then, by conditioning on $T_{\mathrm{DB}}$,

$$R(x, \boldsymbol{a}) = \mathbb{E}_{s_1 \sim b(.|x)} \left[ \sum_{h=1}^{T_{\mathrm{DB}}} \gamma^{h-1} r(s_h, a_h) \,\Big|\, x_1 = x, \, (a_i)_{i=1}^{T_{\mathrm{DB}}} = \boldsymbol{a}_{1:T_{\mathrm{DB}}} \right]$$

$$= \sum_{h=1}^{\infty} \gamma^{h-1} \mathbb{E}_{s \sim b(.|x \oplus (a_1, \ldots, a_{h-1}))} [r(s, a_h)] \cdot \mathbb{P}(T_{\mathrm{DB}} \geqslant h | \boldsymbol{a})$$

$$= \sum_{h=1}^{\infty} \gamma^{h-1} \left( \prod_{i=1}^{h-1} \bar{\beta}(a_i) \right) \mathbb{E}_{s \sim b(.|\tilde{x}_h)} [r(s, a_h)],$$

$$\mathbb{P}V(x, \boldsymbol{a}) = \mathbb{E}_{s_1 \sim b(.|x)} \left[ \gamma^{T_{\mathrm{DB}}} V(s_{T_{\mathrm{DB}}+1}) \,\Big|\, x_1 = x, \, (a_i)_{i=1}^{T_{\mathrm{DB}}} = \boldsymbol{a}_{1:T_{\mathrm{DB}}} \right]$$

$$= \sum_{h=1}^{\infty} \gamma^{h} \mathbb{E}_{s' \sim b(.|x \oplus (a_1, \ldots, a_h))} [V(s')] \cdot \mathbb{P}(T_{\mathrm{DB}} = h | \boldsymbol{a})$$

$$= \sum_{h=1}^{\infty} \gamma^{h} \left( \prod_{i=1}^{h-1} \bar{\beta}(a_i) \right) \beta(a_h) \, \mathbb{E}_{s' \sim b(.|\tilde{x}_{h+1})} [V(s')]. \qquad \square$$

## C. Linear ATST-MDPs: Proofs

### C.1. Linearity of Belief and Action-Sequence Value-Function

In this subsection, we prove: Lemma 4.3 and Theorem 4.4.

**Lemma 4.3** (Restated). *For all $x \in \mathcal{X} \backslash \mathcal{S}$, $b(.|x) = \boldsymbol{\phi}(x)^{\top} \boldsymbol{\mu}(.)$ and $\|\boldsymbol{\phi}(x)\|_2 \leqslant 1$.*
*Moreover, for every map $V : \mathcal{S} \to [0, 1/(1-\gamma)]$ and $(x, a) \in \mathcal{X} \times \mathcal{A}$, it holds that*

$$\mathbb{E}_{s \sim b(.|x)} [r(s, a)] = \langle \boldsymbol{\phi}(x \oplus a), \boldsymbol{\theta} \rangle, \quad \text{and} \quad \mathbb{E}_{s' \sim b(.|x \oplus a)} [V(s')] = \langle \boldsymbol{\phi}(x \oplus a), \boldsymbol{v} \rangle,$$

*where vector $\boldsymbol{v} = \int V(s) d\boldsymbol{\mu}(s)$ satisfies $\|\boldsymbol{v}\|_2 \leqslant \frac{\sqrt{d}}{1-\gamma}$.*

*Proof.* We prove these claims separately:

1. **Linearity of belief:** Fix $x \in \mathcal{X} \backslash \mathcal{S}$ and let $x = (s_1; a_1, \ldots, a_{\Delta})$. Then, the belief $b(.|x)$ satisfies

$$b(.|x) = \int_{\mathcal{S}^{\Delta-1}} \left[ \prod_{i=2}^{\Delta} \mathbb{P}(s_i|s_{i-1}, a_{i-1}) \right] \mathbb{P}(s|s_{\Delta}, a_{\Delta}) \, ds_i$$

$$= \int_{\mathcal{S}^{\Delta-1}} \left[ \prod_{i=2}^{\Delta} \boldsymbol{\phi}(s_{i-1}, a_{i-1})^{\top} \boldsymbol{\mu}(s_i) \right] \boldsymbol{\phi}(s_{\Delta}, a_{\Delta})^{\top} \boldsymbol{\mu}(.) \, ds_i$$

$$= \boldsymbol{\phi}(s_1, a_1)^{\top} \left[ \prod_{i=2}^{\Delta} \left( \int_{\mathcal{S}} \boldsymbol{\mu}(s_i) \boldsymbol{\phi}(s_i, a_i)^{\top} ds_i \right) \right] \boldsymbol{\mu}(.)$$

$$= \langle \boldsymbol{\phi}(x), \boldsymbol{\mu}(.) \rangle.$$

2. **Norm bound:** From Assumption 4.1, $\sup_{s,a} \|\boldsymbol{\phi}(s, a)\|_2 \leqslant 1$. Consider any $x \in \mathcal{X} \backslash \mathcal{S}$ and $a \in \mathcal{A}$. Then, using linearity of belief, we can write

$$\boldsymbol{\phi}(x \oplus a)^{\top} = \boldsymbol{\phi}(x)^{\top} M_a = \int_{\mathcal{S}} \boldsymbol{\phi}(x)^{\top} \boldsymbol{\mu}(s) \boldsymbol{\phi}(s, a)^{\top} ds = \mathbb{E}_{s \sim b(.|x)} \boldsymbol{\phi}(s, a)^{\top},$$

from which the result follows by Jensen's inequality due to convexity of $l^2$-norm

$$\|\boldsymbol{\phi}(x \oplus a)\|_2 = \left\| \mathbb{E}_{s \sim b(.|x)} \boldsymbol{\phi}(s, a) \right\|_2 \leqslant \mathbb{E}_{s \sim b(.|x)} \|\boldsymbol{\phi}(s, a)\|_2 \leqslant 1.$$

3. **Linearity of expected reward and value-function:** From Assumption 4.1, $r(s, a) = \phi(s, a)^\top \boldsymbol{\theta}$. Now, for all $(x, a) \in (\mathcal{X} \backslash \mathcal{S}) \times \mathcal{A}$, we have:

$$\mathbb{E}_{s \sim b(.|x)}\left[r(s, a)\right] = \int_{\mathcal{S}} \phi(x)^\top \boldsymbol{\mu}(s) \phi(s, a)^\top \boldsymbol{\theta} \, ds = \phi(x)^\top M_a \boldsymbol{\theta} = \phi(x \oplus a)^\top \boldsymbol{\theta}.$$

Similarly, for all $x \in \mathcal{X} \backslash \mathcal{S}$, it holds that

$$\mathbb{E}_{s \sim b(.|x)}\left[V(s)\right] = \int_{\mathcal{S}} \phi(x)^\top \boldsymbol{\mu}(s) V(s) \, ds = \phi(x)^\top \boldsymbol{v},$$

where $\boldsymbol{v} = \int_{\mathcal{S}} \boldsymbol{\mu}(s) V(s) ds$ satisfies $\|\boldsymbol{v}\|_2 \leqslant \sup_s |V(s)| \cdot \||\boldsymbol{\mu}|(\mathcal{S})\|_2 \leqslant \frac{\sqrt{d}}{1-\gamma}$. $\qquad\square$

In the main text, action-sequence value functions are defined at data-bursts, and hence on $\mathcal{S} \times \mathcal{A}^{\mathbb{N}}$. For the appendix, it is convenient to use the same notation when the current information state is an arbitrary augmented state $x \in \mathcal{X}$. We extend the notation as follows. For $x \in \mathcal{X}$ and $\boldsymbol{a} = (a_1, a_2, \dots) \in \mathcal{A}^{\mathbb{N}}$, let $K^\pi(x, \boldsymbol{a})$ denote the expected discounted reward obtained by starting with latent state $s_1 \sim b(\cdot|x)$, executing $\boldsymbol{a}$ until the next data-burst, and then following $\pi$. We define $R(x, \boldsymbol{a})$ and $\mathbb{P}V(x, \boldsymbol{a})$ analogously. Finally, for $a \in \mathcal{A}$ and $\boldsymbol{a} \in \mathcal{A}^{\mathbb{N}}$, extend the feature map by

$$\boldsymbol{\psi}(x, a \oplus \boldsymbol{a})^\top = \tfrac{1}{2} \phi(x \oplus a)^\top \left(\beta_a I_{1,2} + \bar{\beta}_a M_{1,2}(\boldsymbol{a})\right).$$

This extension agrees with the main-text definition when $x = s \in \mathcal{S}$. Theorem 4.4 is recovered as the special case $x \in \mathcal{S}$ of the following result.

**Theorem C.1.** *Define* $\boldsymbol{v}_{12}^\pi = 2 \begin{bmatrix} \boldsymbol{\theta}/(1-\gamma) \\ \boldsymbol{v}^\pi \end{bmatrix} \in \mathbb{R}^{2d}$, *where* $\boldsymbol{v}^\pi = \int_{\mathcal{S}} V^\pi(s) d\boldsymbol{\mu}(s)$.
*Then, for every* $x \in \mathcal{X}$ *and sequence* $\boldsymbol{a} \in \mathcal{A}^{\mathbb{N}}$:

$$K^\pi(x, \boldsymbol{a}) = \langle \boldsymbol{\psi}(x, \boldsymbol{a}), \, \boldsymbol{v}_{12}^\pi \rangle.$$

*Moreover, it holds that* $\sup_{x, \boldsymbol{a}} \|\boldsymbol{\psi}(x, \boldsymbol{a})\|_2 \leqslant 1$ *and* $\|\boldsymbol{v}_{12}^\pi\|_2 \leqslant \frac{4\sqrt{d}}{1-\gamma}$.

*Proof.* The result follows from Theorem C.2, which proves linearity of both $R$ and $\mathbb{P}V^\pi$ in the decomposition $K^\pi = R + \mathbb{P}V^\pi$. $\qquad\square$

**Theorem C.2** (Linearity of $R$ and $\mathbb{P}V$ with respect to $\psi$)**.** *For every* $x \in \mathcal{X}$, *sequence* $\boldsymbol{a} \in \mathcal{A}^{\mathbb{N}}$, *and function* $V : \mathcal{S} \to [0, (1-\gamma)^{-1}]$, *it holds that*

$$R(x, \boldsymbol{a}) = \boldsymbol{\psi}(x, \boldsymbol{a})^\top \begin{bmatrix} 2\boldsymbol{\theta}/(1-\gamma) \\ \mathbf{0}_d \end{bmatrix} \qquad and \qquad \mathbb{P}V(x, \boldsymbol{a}) = \boldsymbol{\psi}(x, \boldsymbol{a})^\top \begin{bmatrix} \mathbf{0}_d \\ 2\boldsymbol{v} \end{bmatrix},$$

*where* $\boldsymbol{v} = \int_{\mathcal{S}} V(s) \, d\boldsymbol{\mu}(s)$ *satisfies* $\|\boldsymbol{v}\|_2 \leqslant \frac{\sqrt{d}}{1-\gamma}$. *Moreover,* $\sup_{x, \boldsymbol{a}} \|\boldsymbol{\psi}(x, \boldsymbol{a})\|_2 \leqslant 1$.

*Proof.* Using Lemmas B.1 and 4.3, we write

$$R(x, a \oplus \boldsymbol{a}) = \mathbb{E}_{s \sim b(.|x)}[r(s,a)] + \bar{\beta}(a) \sum_{k=1}^{\infty} \gamma^k \left( \prod_{i=1}^{k-1} \bar{\beta}(a_i) \right) \underset{s \sim b(.|x \oplus (a, a_1, \ldots, a_{k-1}))}{\mathbb{E}} [r(s, a_k)]$$

$$= \boldsymbol{\phi}(x \oplus a)^\top \boldsymbol{\theta} + \bar{\beta}(a) \sum_{k=1}^{\infty} \gamma^k \left( \prod_{i=1}^{k-1} \bar{\beta}(a_i) \right) \boldsymbol{\phi}(x \oplus (a, a_1, \ldots, a_k))^\top \boldsymbol{\theta}$$

$$= \boldsymbol{\phi}(x \oplus a)^\top \left( I + \bar{\beta}(a) \sum_{k=1}^{\infty} \gamma^k (\prod_{i=1}^{k-1} \bar{\beta}(a_i)) \left( \prod_{i=1}^{k} M_{a_i} \right) \right) \boldsymbol{\theta}$$

$$= \boldsymbol{\phi}(x \oplus a)^\top \left( \beta(a) I + \bar{\beta}(a) M_1(\boldsymbol{a}) \right) \boldsymbol{\theta}$$

$$= \tfrac{1}{2} \boldsymbol{\phi}(x \oplus a)^\top \left( \beta(a) \cdot (1-\gamma) I + \bar{\beta}(a) \cdot (1-\gamma) M_1(\boldsymbol{a}) \right) (2\boldsymbol{\theta}/(1-\gamma))$$

$$= \boldsymbol{\psi}(x, a \oplus \boldsymbol{a})^\top \begin{bmatrix} 2\boldsymbol{\theta}/(1-\gamma) \\ \mathbf{0}_d \end{bmatrix},$$

$$\mathbb{P}V(x, a \oplus \boldsymbol{a}) = \beta(a) \gamma \underset{s \sim b(.|x \oplus a)}{\mathbb{E}} [V(s)] + \bar{\beta}(a) \gamma \sum_{k=1}^{\infty} \gamma^k (\prod_{i=1}^{k-1} \bar{\beta}(a_i)) \beta(a_k) \underset{s \sim b(.|x \oplus (a, a_1, \ldots, a_k))}{\mathbb{E}} [V(s)]$$

$$= \beta(a) \gamma \, \boldsymbol{\phi}(x \oplus a)^\top \boldsymbol{v} + \bar{\beta}(a) \gamma \sum_{k=1}^{\infty} \gamma^k \left( \prod_{i=1}^{k-1} \bar{\beta}(a_i) \right) \beta(a_k) \boldsymbol{\phi}(x \oplus (a, a_1, \ldots, a_k))^\top \boldsymbol{v}$$

$$= \boldsymbol{\phi}(x \oplus a)^\top \left( \beta(a) \gamma I + \bar{\beta}(a) \gamma \sum_{k=1}^{\infty} \gamma^k (\prod_{i=1}^{k-1} \bar{\beta}(a_i)) \beta(a_k) \left( \prod_{i=1}^{k} M_{a_i} \right) \right) \boldsymbol{v}$$

$$= \boldsymbol{\phi}(x \oplus a)^\top \left( \beta(a) \gamma I + \bar{\beta}(a) \gamma M_2(\boldsymbol{a}) \right) \boldsymbol{v}$$

$$= \boldsymbol{\psi}(x, a \oplus \boldsymbol{a})^\top \begin{bmatrix} \mathbf{0}_d \\ 2\boldsymbol{v} \end{bmatrix}.$$

To bound the $l_2$-norm, we write

$$\|\boldsymbol{\psi}(x, a \oplus \boldsymbol{a})\|_2 \leqslant \tfrac{1-\gamma}{2} \cdot \left\| \boldsymbol{\phi}(x \oplus a) + \bar{\beta}(a) \sum_{k=1}^{\infty} \gamma^k (\prod_{i=1}^{k-1} \bar{\beta}(a_i)) \boldsymbol{\phi}(x \oplus (a, a_1, \ldots, a_k)) \right\|_2$$

$$+ \tfrac{1}{2} \left\| \beta(a) \gamma \, \boldsymbol{\phi}(x \oplus a) + \bar{\beta}(a) \gamma \sum_{k=1}^{\infty} \gamma^k (\prod_{i=1}^{k-1} \bar{\beta}(a_i)) \beta(a_k) \boldsymbol{\phi}(x \oplus (a, a_1, \ldots, a_k)) \right\|_2$$

$$\overset{(a)}{\leqslant} \left( \tfrac{1-\gamma}{2} \cdot (1 + \sum_{k=1}^{\infty} \gamma^k) + \tfrac{\gamma}{2} \cdot (\beta(a) + \bar{\beta}(a) \sum_{k=1}^{\infty} (\prod_{i=1}^{k-1} \bar{\beta}(a_i)) \beta(a_k)) \right)$$

$$\leqslant \left( \tfrac{1-\gamma}{2} \cdot \tfrac{1}{1-\gamma} + \tfrac{\gamma}{2} \cdot 1 \right) = \tfrac{1+\gamma}{2} \leqslant 1.$$

where (a) uses the fact that $\sup_{x'} \|\boldsymbol{\phi}(x')\|_2 \leqslant 1$. □

## C.2. Approximation of the Action-Sequence Feature Map: Proofs

In this subsection, we prove Theorem 4.6. A key technical tool is Lemma C.3 provided below.

**Theorem 4.6** (Restated). *Assume $\widehat{M}_a \in \mathbb{R}^{d \times d}$ and $\widehat{\beta}_a \in [0, 1]$ satisfy $\sup_{a \in \mathcal{A}} \|\widehat{M}_a - M_a\|_2 \leqslant \varepsilon$ and $\sup_{a \in \mathcal{A}} |\widehat{\beta}_a - \beta_a| \leqslant \varepsilon_\beta$ for some $\varepsilon \in [0, \tfrac{1-\gamma}{2\sqrt{d}}]$ and $\varepsilon_\beta \in [0, 1]$. Let $\widehat{\boldsymbol{\psi}} : \mathcal{S} \times \mathcal{A}^{\mathbb{N}} \to \mathbb{R}^{2d}$ be the estimated action-sequence feature map obtained from (3) by replacing action-matrices $M_a$ and data-burst probabilities $\beta_a$ with their estimates $\widehat{M}_a$, $\widehat{\beta}_a$ in computation. Then, it holds that $\sup_{s, \boldsymbol{a}} \|(\widehat{\boldsymbol{\psi}} - \boldsymbol{\psi})(s, \boldsymbol{a})\|_2 \leqslant \tfrac{16d}{1-\gamma} (\varepsilon + \varepsilon_\beta / \sqrt{d})$.*

*Moreover, function $\widetilde{\boldsymbol{\psi}}(s, \boldsymbol{a}) = \dfrac{\widehat{\boldsymbol{\psi}}(s, \boldsymbol{a})}{1 + 16d(\varepsilon + \varepsilon_\beta / \sqrt{d})/(1-\gamma)}$ is a $\tfrac{32d(\varepsilon + \varepsilon_\beta / \sqrt{d})}{1-\gamma}$-admissible estimation of $\boldsymbol{\psi}$.*

At the core of the proof is the following more general lemma, which bounds the estimation error in the feature vector $\boldsymbol{\psi}$ using that of action-matrices.

**Lemma C.3.** *Assume estimates $\widehat{M}_a$ satisfy $\sup_{a \in \mathcal{A}} \|\widehat{M}_a - M_a\|_2 \leqslant \varepsilon$ and define norm-corrected estimates $\widehat{M}_a^c = \widehat{M}_a/(1 + \varepsilon\sqrt{d})$. Also, suppose that estimates $\widehat{\beta}_a \in [0, 1]$ satisfy $\sup_{a \in \mathcal{A}} |\widehat{\beta}_a - \beta_a| \leqslant \varepsilon_\beta$. Let $\widehat{\boldsymbol{\psi}}, \widehat{\boldsymbol{\psi}}_c : \mathcal{S} \times \mathcal{A}^{\mathbb{N}} \to \mathbb{R}^{2d}$ be the estimated action-sequence feature maps obtained from $\boldsymbol{\psi}$ by replacing $M_a, \beta_a$ with their estimates $\widehat{M}_a$ (or $\widehat{M}_a^c$) and $\widehat{\beta}_a$,*

*respectively. Then, for all $s \in \mathcal{S}$ and $\boldsymbol{a} \in \mathcal{A}^{\mathbb{N}}$, it holds that*

$$\|(\widehat{\boldsymbol{\psi}}_c - \boldsymbol{\psi})(s, \boldsymbol{a})\|_2 \leqslant \frac{4d^2}{1-\gamma} \cdot (\varepsilon + \varepsilon_\beta / d^{3/2}).$$

*Moreover, if $\varepsilon < (1/\gamma - 1)/\sqrt{d}$, then it holds that*

$$\|(\widehat{\boldsymbol{\psi}} - \boldsymbol{\psi})(s, \boldsymbol{a})\|_2 \leqslant \frac{4d\,(1-\gamma)}{(1 - \gamma(1 + \varepsilon\sqrt{d}))^2} \cdot (\varepsilon + \varepsilon_\beta / \sqrt{d}).$$

Taking this lemma as given, let us prove Theorem 4.6.

*Proof of Theorem 4.6.* For $\varepsilon \in [0, \frac{1-\gamma}{2\sqrt{d}}]$, we have $\varepsilon < \frac{1/\gamma - 1}{\sqrt{d}}$. So, by the second case of Lemma C.3,

$$\sup_{s,\boldsymbol{a}} \|(\widehat{\boldsymbol{\psi}} - \boldsymbol{\psi})(s, \boldsymbol{a})\|_2 \leqslant \frac{4d\,(1-\gamma)}{(1 - \gamma(1 + \varepsilon\sqrt{d}))^2} \cdot (\varepsilon + \varepsilon_\beta / \sqrt{d}) \leqslant \frac{16d}{1-\gamma} \cdot (\varepsilon + \varepsilon_\beta / \sqrt{d}),$$

which proves the first statement. Now, we have to show that $\widetilde{\boldsymbol{\psi}}$ is $\frac{32d(\varepsilon + \varepsilon_\beta / \sqrt{d})}{1-\gamma}$-admissible estimation of $\boldsymbol{\psi}$. Let $\varepsilon_2 = \frac{16d(\varepsilon + \varepsilon_\beta / \sqrt{d})}{1-\gamma}$. Then, for every $s, \boldsymbol{a}$ write following

$$\|(\widetilde{\boldsymbol{\psi}} - \boldsymbol{\psi})(s, \boldsymbol{a})\|_2 \leqslant \frac{\|(\widehat{\boldsymbol{\psi}} - \boldsymbol{\psi})(s, \boldsymbol{a})\|_2}{1 + \varepsilon_2} + \frac{\varepsilon_2 \|\boldsymbol{\psi}(s, \boldsymbol{a})\|_2}{1 + \varepsilon_2} \leqslant 2\varepsilon_2 = \frac{32d(\varepsilon + \varepsilon_\beta / \sqrt{d})}{1-\gamma},$$

$$\|\widetilde{\boldsymbol{\psi}}(s, \boldsymbol{a})\|_2 \leqslant \frac{\|(\widehat{\boldsymbol{\psi}} - \boldsymbol{\psi})(s, \boldsymbol{a})\|_2}{1 + \varepsilon_2} + \frac{\|\boldsymbol{\psi}(s, \boldsymbol{a})\|_2}{1 + \varepsilon_2} \leqslant \frac{\varepsilon_2}{1 + \varepsilon_2} + \frac{1}{1 + \varepsilon_2} = 1.$$

So, we only have to show continuity of $\widetilde{\boldsymbol{\psi}}(s, .)$ with respect to the product topology on $\mathcal{A}^{\mathbb{N}}$ and the standard topology on $\mathbb{R}^{2d}$. This follows from the formula of $\widehat{\boldsymbol{\psi}}$, which is based on the $\gamma$-discounted summation of matrix products. Each term is bounded in operator norm as shown by Lemma C.5:

$$\gamma^n \|\textstyle\prod_{i=1}^n \widehat{M}_{a_i}\|_2 \leqslant \gamma^n \cdot \sqrt{d}(1 + \varepsilon\sqrt{d})^n \leqslant \sqrt{d} \cdot \left(\frac{1+\gamma}{2}\right)^n,$$

where exponent term $\frac{1+\gamma}{2} \in (0, 1)$ ensures convergence and therefore continuity for $\widehat{\boldsymbol{\psi}}$ and $\widetilde{\boldsymbol{\psi}}$. $\qquad\square$

### C.2.1. PROOF OF LEMMA C.3

The following lemmas are used to prove Lemma C.3.

**Lemma C.4.** *For all $n \in \mathbb{N}$ and $a_1, \ldots, a_n \in \mathcal{A}$, it holds that*

$$\|\textstyle\prod_{i=1}^n M_{a_i}\|_2 \leqslant \sqrt{d} \quad \text{and} \quad \rho(\textstyle\prod_{i=1}^n M_{a_i}) \leqslant 1.$$

*Proof.* Using the Linear MDP Assumption 4.1, we can write

$$\textstyle\prod_{i=1}^n M_{a_i} = \int_{\mathcal{S}} \boldsymbol{\mu}(s)\boldsymbol{\phi}(s, a_1)^\top \textstyle\prod_{i=2}^n M_{a_i} ds = \int_{\mathcal{S}} \boldsymbol{\mu}(s)\boldsymbol{\phi}((s; a_1, \ldots, a_n))^\top ds.$$

Then, by spectral-Frobenius inequality, it follows that

$$\begin{aligned}
\|\textstyle\prod_{i=1}^n M_{a_i}\|_2 &\leqslant \sqrt{\textstyle\sum_{i \in [d]} \left\|\int_{\mathcal{S}} \mu_i(s)\boldsymbol{\phi}((s; a_1, \ldots, a_n))^\top ds\right\|_2^2} \\
&\leqslant \sqrt{\textstyle\sum_{i \in [d]} (|\mu_i|(\mathcal{S}))^2 \cdot \sup_{x \in \mathcal{X}} \|\boldsymbol{\phi}(x)\|_2^2} \\
&= \||\boldsymbol{\mu}|(\mathcal{S})\|_2 \cdot \sup_{x \in \mathcal{X}} \|\boldsymbol{\phi}(x)\|_2 \leqslant \sqrt{d},
\end{aligned}$$

where the final inequality follows from Assumption 4.1 and Lemma 4.3.

For the spectral-radius bound, fix $B = \prod_{i=1}^{n} M_{a_i}$. Applying the operator-norm bound to the repeated sequence $(a_1, \ldots, a_n)$ repeated $m$ times gives $\|B^m\|_2 \leqslant \sqrt{d}$ for all $m \geqslant 1$. Hence, by Gelfand's formula,

$$\rho(B) = \lim_{m \to \infty} \|B^m\|_2^{1/m} \leqslant 1. \qquad \square$$

**Lemma C.5.** *Suppose that for every $a \in \mathcal{A}$, estimate $\widehat{M}_a \in \mathbb{R}^{d \times d}$ satisfies $\|M_a - \widehat{M}_a\|_2 \leqslant \varepsilon$.*

*Then, for all $n \in \mathbb{N}$ and $a_1, \ldots a_n \in \mathcal{A}$, it holds that $\| \prod_{i=1}^{n} \widehat{M}_{a_i} \|_2 \leqslant \sqrt{d}(1 + \varepsilon\sqrt{d})^n$.*

*Proof.* Let $E_a = \widehat{M}_a - M_a$ so that $\widehat{M}_a = M_a + E_a$ and $\|E_a\|_2 \leqslant \varepsilon$.

Also, let $X_a^0 = M_a$ and $X_a^1 = E_a$. Then, we can write

$$
\begin{aligned}
\| \textstyle\prod_{i=1}^{n} \widehat{M}_{a_i} \|_2 &= \| \textstyle\prod_{i=1}^{n} (M_{a_i} + E_{a_i}) \|_2 \\
&\leqslant \textstyle\sum_{\boldsymbol{b} \in \{0,1\}^n} \| \textstyle\prod_{i=1}^{n} X_{a_i}^{b_i} \|_2 \\
&\stackrel{(a)}{\leqslant} \textstyle\sum_{\boldsymbol{b} \in \{0,1\}^n} \left( \sqrt{d} \textstyle\prod_{i=1}^{n} [\mathbb{I}(b_i = 0) + \mathbb{I}(b_i = 1) \cdot \|E_{a_i}\|_2 \sqrt{d}] \right) \\
&\leqslant \sqrt{d} \cdot \textstyle\sum_{\boldsymbol{b} \in \{0,1\}^n} (\varepsilon\sqrt{d})^{\|\boldsymbol{b}\|_1} \\
&= \sqrt{d}(1 + \varepsilon\sqrt{d})^n,
\end{aligned}
$$

where (a) follows by bounding consecutive blocks of neighbouring $X_a^0$ matrices as $\|X_{a_l}^0 X_{a_{l+1}}^0 \ldots X_{a_r}^0\|_2 \leqslant \sqrt{d}$ using Lemma C.4 and pairing each such block (except maybe one) with a neighbouring matrix $X_a^1$, which has $\|X_a^1\|_2 = \|E_a\|_2 \leqslant \sqrt{d}$. $\qquad \square$

**Lemma C.6.** *Let $\varepsilon \in [0, 1)$. Suppose matrices $A, B \in \mathbb{R}^{d \times d}$ satisfy $\|A\|_2 \leqslant \sqrt{d}$ and $\|A - B\|_2 \leqslant \varepsilon$. Then, $B' = B/(1 + \varepsilon\sqrt{d})$ satisfies $\|A - B'\|_2 \leqslant 2d\varepsilon$.*

*Proof.* Let $A' = A/(1 + \varepsilon\sqrt{d})$. Using the triangle inequality, we can write

$$\|A - B'\|_2 \leqslant \|A - A'\|_2 + \|A' - B'\|_2 \leqslant \tfrac{\varepsilon\sqrt{d}}{1+\varepsilon\sqrt{d}} \cdot \|A\|_2 + \tfrac{1}{1+\varepsilon\sqrt{d}} \cdot \|A - B\|_2 \leqslant 2d\varepsilon.$$

$$\square$$

**Lemma C.7.** *Under the conditions of Lemma C.5, let $\widehat{M}_a^c = \widehat{M}_a/(1 + \varepsilon\sqrt{d})$. Then, we have*

$$\| \textstyle\prod_{i=1}^{n} \widehat{M}_{a_i} - \textstyle\prod_{i=1}^{n} M_{a_i} \|_2 \leqslant d(1 + \varepsilon\sqrt{d})^{n-1} n\varepsilon, \tag{6}$$

$$\| \textstyle\prod_{i=1}^{n} \widehat{M}_{a_i}^c - \textstyle\prod_{i=1}^{n} M_{a_i} \|_2 \leqslant 2d^2 n\varepsilon. \tag{7}$$

*Proof.* To show (6), we write

$$
\begin{aligned}
\| \textstyle\prod_{i=1}^{n} \widehat{M}_{a_i} - \textstyle\prod_{i=1}^{n} M_{a_i} \|_2 &\leqslant \textstyle\sum_{k=1}^{n} \|(\textstyle\prod_{i=1}^{k-1} \widehat{M}_{a_i})(\widehat{M}_{a_k} - M_{a_k})(\textstyle\prod_{i=k+1}^{n} M_{a_i})\|_2 \\
&\leqslant \textstyle\sum_{k=1}^{n} \| \textstyle\prod_{i=1}^{k-1} \widehat{M}_{a_i} \|_2 \|\widehat{M}_{a_k} - M_{a_k}\|_2 \| \textstyle\prod_{i=k+1}^{n} M_{a_i} \|_2 \\
&\stackrel{(a)}{\leqslant} \textstyle\sum_{k=1}^{n} \left( \sqrt{d}(1 + \sqrt{d}\varepsilon)^{k-1} \cdot \varepsilon \cdot \sqrt{d} \right) \leqslant d(1 + \varepsilon\sqrt{d})^{n-1} n\varepsilon
\end{aligned}
$$

where (a) follows from Lemmas C.4 and C.5.

Similarly, to prove (7), we write

$$
\begin{aligned}
\| \textstyle\prod_{i=1}^{n} \widehat{M}_{a_i}^c - \textstyle\prod_{i=1}^{n} M_{a_i} \|_2 &\leqslant \textstyle\sum_{k=1}^{n} \|(\textstyle\prod_{i=1}^{k-1} \widehat{M}_{a_i}^c)(\widehat{M}_{a_k}^c - M_{a_k})(\textstyle\prod_{i=k+1}^{n} M_{a_i})\|_2 \\
&\leqslant \textstyle\sum_{k=1}^{n} \| \textstyle\prod_{i=1}^{k-1} \widehat{M}_{a_i}^c \|_2 \|\widehat{M}_{a_k}^c - M_{a_k}\|_2 \| \textstyle\prod_{i=k+1}^{n} M_{a_i} \|_2 \\
&\stackrel{(b)}{\leqslant} \textstyle\sum_{k=1}^{n} (\sqrt{d} \cdot 2d\varepsilon \cdot \sqrt{d}) = 2d^2 n\varepsilon,
\end{aligned}
$$

where (b) follows from Lemmas C.4, C.5, and C.6. $\qquad \square$

**Lemma C.8.** *Let sequences $(a_i)_{i=1}^\infty$, $(b_i)_{i=1}^\infty$ with values in $[0,1]$ be such that $\sup_{i \in \mathbb{N}} |a_i - b_i| \leqslant \varepsilon$ for some $\varepsilon \in [0,1]$. Let $\bar{a}_i = 1 - a_i$ and $\bar{b}_i = 1 - b_i$ for every $i \in \mathbb{N}$. Then, it holds that*

$$\forall n \in \mathbb{N}, \quad |\textstyle\prod_{i=1}^n b_i - \prod_{i=1}^n a_i| \leqslant n\varepsilon, \tag{8}$$

$$\forall \gamma \in (0,1), \quad \textstyle\sum_{k=1}^\infty \gamma^k |(\prod_{i=1}^{k-1} \bar{b}_i) b_k - (\prod_{i=1}^{k-1} \bar{a}_i) a_k| \leqslant \frac{2\varepsilon}{1-\gamma}. \tag{9}$$

*Proof.* To prove (8) for arbitrary $n \in \mathbb{N}$, we simply write:

$$
\begin{aligned}
|\textstyle\prod_{i=1}^n b_i - \prod_{i=1}^n a_i| &\leqslant \textstyle\sum_{k=1}^n |\prod_{i=1}^{k-1} a_i \prod_{i=k}^n b_i - \prod_{i=1}^k a_i \prod_{i=k+1}^n b_i| \\
&= \textstyle\sum_{k=1}^n |b_k - a_k| \prod_{i=1}^{k-1} a_i \prod_{i=k+1}^n b_i \\
&\leqslant n\varepsilon.
\end{aligned}
$$

To prove (9) for arbitrary $\gamma \in (0,1)$, consider the finite supremum over all appropriate pairs of sequences:

$$S = \sup_{\boldsymbol{a},\boldsymbol{b} \in [0,1]^{\mathbb{N}}: \sup_i |a_i - b_i| \leqslant \varepsilon} \textstyle\sum_{k=1}^\infty \gamma^k |(\prod_{i=1}^{k-1} \bar{b}_i) b_k - (\prod_{i=1}^{k-1} \bar{a}_i) a_k| \leqslant \textstyle\sum_{k=1}^\infty \gamma^k = \frac{1}{1-\gamma},$$

with intention to show that $S \leqslant \frac{2\varepsilon}{1-\gamma}$. Then, for all $\boldsymbol{a}, \boldsymbol{b} \in [0,1]^{\mathbb{N}}$ such that $\sup_i |a_i - b_i| \leqslant \varepsilon$, we can write:

$$
\begin{aligned}
\textstyle\sum_{k=1}^\infty \gamma^k |(\prod_{i=1}^{k-1} \bar{b}_i) b_k - (\prod_{i=1}^{k-1} \bar{a}_i) a_k| &\leqslant \gamma|b_1 - a_1| + \textstyle\sum_{k=2}^\infty \gamma^k |\bar{b}_1 - \bar{a}_1| \cdot |(\prod_{i=2}^{k-1} \bar{b}_i) b_k| \\
&\quad + \textstyle\sum_{k=2}^\infty \gamma^k |\bar{a}_1| \cdot |(\prod_{i=2}^{k-1} \bar{b}_i) b_k - (\prod_{i=2}^{k-1} \bar{a}_i) a_k| \\
&\leqslant \varepsilon \cdot (1 + \textstyle\sum_{k=1}^\infty (\prod_{i=1}^{k-1} \bar{b}_{i+1}) b_{k+1}) \\
&\quad + \gamma \cdot \textstyle\sum_{k=1}^\infty \gamma^k |(\prod_{i=1}^{k-1} \bar{b}_{i+1}) b_{k+1} - (\prod_{i=1}^{k-1} \bar{a}_{i+1}) a_{k+1}| \\
&\leqslant 2\varepsilon + \gamma S.
\end{aligned}
$$

Therefore, it holds that $S \leqslant 2\varepsilon + \gamma S$ and so $S \leqslant \frac{2\varepsilon}{1-\gamma}$. □

*Proof of Lemma C.3.* Let $\widehat{\bar{\beta}}_a = 1 - \widehat{\beta}_a \in [0,1]$ to ease notation.
From Lemma C.4, it follows that matrices $M_1(\boldsymbol{a})$, $M_2(\boldsymbol{a})$ from (4) satisfy

$$\|M_1(\boldsymbol{a})\|_2 \leqslant 1 + \textstyle\sum_{k=1}^\infty \gamma^k (\prod_{i=1}^{k-1} \bar{\beta}_{a_i}) \|(\prod_{i=1}^k M_{a_i})\|_2 \leqslant \textstyle\sum_{k=0}^\infty \gamma^k \sqrt{d} \leqslant \frac{\sqrt{d}}{1-\gamma}, \tag{10a}$$

$$\|M_2(\boldsymbol{a})\|_2 \leqslant \textstyle\sum_{k=1}^\infty \gamma^k (\prod_{i=1}^{k-1} \bar{\beta}_{a_i}) \beta_{a_k} \|(\prod_{i=1}^k M_{a_i})\|_2 \leqslant \textstyle\sum_{k=1}^\infty (\prod_{i=1}^{k-1} \bar{\beta}_{a_i}) \beta_{a_k} \sqrt{d} \leqslant \sqrt{d}. \tag{10b}$$

**Part 1:** We prove the result for $\widehat{\psi}$ first. Suppose $\varepsilon \in [0, (1/\gamma - 1)/\sqrt{d})$, so that $\gamma(1 + \varepsilon\sqrt{d}) \in [0,1)$.

Let $\widehat{M}_1(\boldsymbol{a})$, $\widehat{M}_2(\boldsymbol{a})$ denote estimates for matrices $M_1(\boldsymbol{a})$, $M_2(\boldsymbol{a})$ computed using estimates $\widehat{M}_a$, $\widehat{\beta}_a$.

Note that for all $c \in [0,1)$, $\sum_{n=0}^\infty c^n n = \frac{c}{(1-c)^2}$ and $\sup_n c^n n \leqslant \frac{1}{1-c}$. Then, using Lemmas C.4, C.7, and C.8, we can

write:

$$
\begin{aligned}
\|\widehat{M}_1(\boldsymbol{a}) - M_1(\boldsymbol{a})\|_2 &\leqslant \textstyle\sum_{k=1}^{\infty} \gamma^k \|\!|(\prod_{i=1}^{k-1} \widehat{\bar{\beta}}_{a_i})(\prod_{i=1}^{k} \widehat{M}_{a_i}) - (\prod_{i=1}^{k-1} \bar{\beta}_{a_i})(\prod_{i=1}^{k} M_{a_i})\|\!|_2 \\
&\leqslant \textstyle\sum_{k=1}^{\infty} \gamma^k \|\!| \prod_{i=1}^{k} \widehat{M}_{a_i} - \prod_{i=1}^{k} M_{a_i} \|\!|_2 \\
&+ \textstyle\sum_{k=1}^{\infty} \gamma^k \left| \prod_{i=1}^{k-1} \widehat{\bar{\beta}}_{a_i} - \prod_{i=1}^{k-1} \bar{\beta}_{a_i} \right| \|\!| \prod_{i=1}^{k} M_{a_i} \|\!|_2 \\
&\leqslant \textstyle\sum_{k=1}^{\infty} \gamma^k (1 + \varepsilon\sqrt{d})^{k-1} k\,\varepsilon d + \sum_{k=1}^{\infty} \gamma^k k \varepsilon_\beta \sqrt{d} \\
&= \tfrac{\gamma(1+\varepsilon\sqrt{d})}{(1-\gamma(1+\varepsilon\sqrt{d}))^2} \cdot \tfrac{\varepsilon d}{1+\varepsilon\sqrt{d}} + \tfrac{\gamma}{(1-\gamma)^2} \cdot \varepsilon_\beta \sqrt{d} \\
&\leqslant \tfrac{d\gamma}{(1-\gamma(1+\varepsilon\sqrt{d}))^2}(\varepsilon + \varepsilon_\beta/\sqrt{d}), \\
\|\widehat{M}_2(\boldsymbol{a}) - M_2(\boldsymbol{a})\|_2 &\leqslant \textstyle\sum_{k=1}^{\infty} \gamma^k \|\!|(\prod_{i=1}^{k-1} \widehat{\bar{\beta}}_{a_i})\widehat{\beta}_{a_k}(\prod_{i=1}^{k} \widehat{M}_{a_i}) - (\prod_{i=1}^{k-1} \bar{\beta}_{a_i})\beta_{a_k}(\prod_{i=1}^{k} M_{a_i})\|\!|_2 \\
&\leqslant \textstyle\sup_{k\in\mathbb{N}} \left( \gamma^k \cdot \|\!| \prod_{i=1}^{k} \widehat{M}_{a_i} - \prod_{i=1}^{k} M_{a_i}\|\!|_2 \right) \\
&+ \textstyle\sum_{k=1}^{\infty} \gamma^k \left|(\prod_{i=1}^{k-1} \widehat{\bar{\beta}}_{a_i})\widehat{\beta}_{a_k} - (\prod_{i=1}^{k-1} \bar{\beta}_{a_i})\beta_{a_k}\right| \|\!| \prod_{i=1}^{k} M_{a_i} \|\!|_2 \\
&\leqslant \textstyle\sup_{k\in\mathbb{N}} \gamma^k (1+\varepsilon\sqrt{d})^{k-1} k\,\varepsilon d \\
&+ \textstyle\sum_{k=1}^{\infty} \gamma^k \left|(\prod_{i=1}^{k-1} \widehat{\bar{\beta}}_{a_i})\widehat{\beta}_{a_k} - (\prod_{i=1}^{k-1} \bar{\beta}_{a_i})\beta_{a_k}\right| \sqrt{d} \\
&\leqslant \tfrac{1}{1-\gamma(1+\varepsilon\sqrt{d})} \cdot \tfrac{\varepsilon d}{1+\varepsilon\sqrt{d}} + \tfrac{2}{1-\gamma} \cdot \varepsilon_\beta \sqrt{d} \\
&\leqslant \tfrac{2d}{1-\gamma(1+\varepsilon\sqrt{d})} \cdot (\varepsilon + \varepsilon_\beta/\sqrt{d}).
\end{aligned}
$$

From (3), we have that

$$
\boldsymbol{\psi}(s, a \oplus \boldsymbol{a})^\top = \tfrac{1}{2} \boldsymbol{\phi}(s \oplus a)^\top \left( \beta_a I_{12} + \bar{\beta}_a M_{12}(\boldsymbol{a}) \right),
$$

where $I_{12} = \left[\, (1-\gamma)I \;\; \gamma I \,\right] \in \mathbb{R}^{d\times 2d}$ and $M_{12}(\boldsymbol{a}) = \left[\, (1-\gamma)M_1(\boldsymbol{a}) \;\; \gamma M_2(\boldsymbol{a}) \,\right] \in \mathbb{R}^{d\times 2d}$.

Then, using the fact that $\|\boldsymbol{\phi}(s,a)\|_2 \leqslant 1$, it follows that

$$
\begin{aligned}
\|(\widehat{\boldsymbol{\psi}} - \boldsymbol{\psi})(s, a \oplus \boldsymbol{a})\|_2 &\leqslant \tfrac{1}{2} \|\!| \left[\, (1-\gamma)(\widehat{M}_1 - M_1)(\boldsymbol{a}) \;\; \gamma(\widehat{M}_2 - M_2)(\boldsymbol{a}) \,\right]^\top \|\!|_2 \\
&+ \tfrac{1}{2} |\widehat{\beta}_a - \beta_a| \cdot \|\!| \left[\, (1-\gamma)(I - M_1(\boldsymbol{a})) \;\; \gamma(I - M_2(\boldsymbol{a})) \,\right]^\top \|\!|_2 \\
&\overset{(a)}{\leqslant} \tfrac{1}{2}(1-\gamma) \|(\widehat{M}_1 - M_1)(\boldsymbol{a})\|_2 + \tfrac{1}{2}\gamma \|(\widehat{M}_2 - M_2)(\boldsymbol{a})\|_2 \\
&+ \tfrac{1}{2}\varepsilon_\beta(1-\gamma)(1 + \sqrt{d}/(1-\gamma)) + \tfrac{1}{2}\varepsilon_\beta\gamma(1+\sqrt{d}) \\
&\overset{(b)}{\leqslant} \tfrac{2\,d(1-\gamma)}{(1-\gamma(1+\varepsilon\sqrt{d}))^2} \cdot (\varepsilon + \varepsilon_\beta/\sqrt{d}) + 2\varepsilon_\beta\sqrt{d} \\
&\leqslant \tfrac{4\,d(1-\gamma)}{(1-\gamma(1+\varepsilon\sqrt{d}))^2} \cdot (\varepsilon + \varepsilon_\beta/\sqrt{d})
\end{aligned}
$$

where (a) follows from (10) and (b) from the bounds on $\|\widehat{M}_1(\boldsymbol{a}) - M_1(\boldsymbol{a})\|_2$ and $\|\widehat{M}_2(\boldsymbol{a}) - M_2(\boldsymbol{a})\|_2$ above.

**Part 2:** Here, we will prove the result for $\widehat{\boldsymbol{\psi}}_c$ using similar approach. Suppose $\varepsilon \in [0,1)$.

Let $\widehat{M}_1^c(\boldsymbol{a}), \widehat{M}_2^c(\boldsymbol{a})$ denote estimates for matrices $M_1(\boldsymbol{a}), M_2(\boldsymbol{a})$ computed using estimates $\widehat{M}_a^c, \widehat{\beta}_a$.

Using Lemmas C.4, C.7, and C.8, we write:

$$
\begin{aligned}
\|\widehat{M}_1^c(\boldsymbol{a}) - M_1(\boldsymbol{a})\|_2 &\leqslant \sum_{k=1}^{\infty} \gamma^k \|\textstyle\prod_{i=1}^k \widehat{M}_{a_i}^c - \prod_{i=1}^k M_{a_i}\|_2 \\
&\quad + \sum_{k=1}^{\infty} \gamma^k \left|\textstyle\prod_{i=1}^{k-1} \widehat{\bar{\beta}}_{a_i} - \prod_{i=1}^{k-1} \bar{\beta}_{a_i}\right| \|\textstyle\prod_{i=1}^k M_{a_i}\|_2 \\
&\leqslant \sum_{k=1}^{\infty} \gamma^k \, 2d^2 k\varepsilon + \sum_{k=1}^{\infty} \gamma^k k \varepsilon_\beta \sqrt{d} \\
&\leqslant \frac{2d\gamma}{(1-\gamma)^2} \cdot (d\varepsilon + \varepsilon_\beta/\sqrt{d}), \\
\|\widehat{M}_2^c(\boldsymbol{a}) - M_2(\boldsymbol{a})\|_2 &\leqslant \sup_{k \in \mathbb{N}} \left(\gamma^k \cdot \|\textstyle\prod_{i=1}^k \widehat{M}_{a_i}^c - \prod_{i=1}^k M_{a_i}\|_2\right) \\
&\quad + \sum_{k=1}^{\infty} \gamma^k \left|(\textstyle\prod_{i=1}^{k-1} \widehat{\bar{\beta}}_{a_i})\widehat{\beta}_{a_k} - (\prod_{i=1}^{k-1} \bar{\beta}_{a_i})\beta_{a_k}\right| \|\textstyle\prod_{i=1}^k M_{a_i}\|_2 \\
&\leqslant \sup_{k \in \mathbb{N}} \gamma^k \, 2d^2 k\varepsilon + \sum_{k=1}^{\infty} \gamma^k \left|(\textstyle\prod_{i=1}^{k-1} \widehat{\bar{\beta}}_{a_i})\widehat{\beta}_{a_k} - (\prod_{i=1}^{k-1} \bar{\beta}_{a_i})\beta_{a_k}\right| \sqrt{d} \\
&\leqslant \frac{2d^2\varepsilon}{1-\gamma} + \frac{2\varepsilon_\beta \sqrt{d}}{1-\gamma} \leqslant \frac{2d}{1-\gamma} \cdot (d\varepsilon + \varepsilon_\beta/\sqrt{d}).
\end{aligned}
$$

As in Part 1, we conclude that

$$
\begin{aligned}
\|(\widehat{\boldsymbol{\psi}}_c - \boldsymbol{\psi})(s, a \oplus \boldsymbol{a})\|_2 &\leqslant \tfrac{1}{2} \|\begin{bmatrix} (1-\gamma)(\widehat{M}_1^c - M_1)(\boldsymbol{a}) & \gamma(\widehat{M}_2^c - M_2)(\boldsymbol{a}) \end{bmatrix}^\top\|_2 \\
&\quad + \tfrac{1}{2} |\widehat{\beta}_a - \beta_a| \cdot \|\begin{bmatrix} (1-\gamma)(I - M_1(\boldsymbol{a})) & \gamma(I - M_2(\boldsymbol{a})) \end{bmatrix}^\top\|_2 \\
&\leqslant \tfrac{1}{2}(1-\gamma) \|(\widehat{M}_1^c - M_1)(\boldsymbol{a})\|_2 + \tfrac{1}{2}\gamma \|(\widehat{M}_2^c - M_2)(\boldsymbol{a})\|_2 \\
&\quad + \tfrac{1}{2}\varepsilon_\beta(1-\gamma)(1 + \sqrt{d}/(1-\gamma)) + \tfrac{1}{2}\varepsilon_\beta\gamma(1 + \sqrt{d}) \\
&\stackrel{(c)}{\leqslant} \frac{2d}{1-\gamma} \cdot (d\varepsilon + \varepsilon_\beta/\sqrt{d}) + 2\varepsilon_\beta\sqrt{d} \\
&\leqslant \frac{4d}{1-\gamma} \cdot (d\varepsilon + \varepsilon_\beta/\sqrt{d}),
\end{aligned}
$$

where (c) follows from the bounds on $\|\widehat{M}_1^c(\boldsymbol{a}) - M_1(\boldsymbol{a})\|_2$ and $\|\widehat{M}_2^c(\boldsymbol{a}) - M_2(\boldsymbol{a})\|_2$ above.

This concludes the proof of both statements. $\qquad\square$

## C.3. Off-policy Evaluation

In this subsection, we prove Lemma 4.7, which will follow from Lemma C.9, provided below. We also prove Lemma 4.8. Corollary 4.9 follows immediately from these lemmas, by setting $\varepsilon_\beta = \varepsilon\sqrt{d}$ small enough in Theorem 4.6 and picking dataset size in Lemmas 4.7 and 4.8 large enough for the resulting uniform bounds to hold with probabilities $1 - p/2$ each.

For the sake of notation, let $\boldsymbol{x}^{(n)} := \phi(s_n, a_n)$ and $\boldsymbol{y}_a^{(n)} := \phi(s_n', a)$, so that $X, Y_a \in \mathbb{R}^{N \times d}$ have rows $\boldsymbol{x}^{(n)}, \boldsymbol{y}_a^{(n)}$ respectively. Then, $\Sigma = \mathbb{E}[\boldsymbol{x}^{(1)}(\boldsymbol{x}^{(1)})^\top] = \mathbb{E}[\frac{1}{N}X^\top X]$.

Recall that we consider ridge estimators $\widehat{M}_a = (X^\top X + \lambda I_d)^{-1} X^\top Y_a$.

Observe that $\mathbb{E}[\boldsymbol{y}_a^{(n)} \mid s_n, a_n] = M_a^\top \boldsymbol{x}^{(n)}$ and $\|\boldsymbol{y}_a^{(n)}\|_2 \leqslant 1$ almost surely. Moreover, for $\boldsymbol{z}_a^{(n)} := \boldsymbol{y}_a^{(n)} - M_a^\top \boldsymbol{x}^{(n)}$, it holds that $\|\boldsymbol{z}_a^{(n)}\|_2 \leqslant 2$. In the matrix form, we consider $Z_a := Y_a - X M_a$.

**Lemma 4.7** (Restated). *There exists absolute constant $C \geqslant 1$ such that for all $p \in (0,1)$ and $N \geqslant \frac{4C^2 d \log(2Ad/p)}{\lambda_{\min}(\Sigma)^2}$, by choosing $\lambda = 1$, with probability at least $1 - p$, it holds that*

$$
\sup_{a \in \mathcal{A}} \|\widehat{M}_a^\lambda - M_a\|_2 \leqslant 4C\sqrt{\frac{d \log(2Ad/p)}{N\lambda_{\min}(\Sigma)^2}}.
$$

*Proof.* We will show that this claim holds for the same $C \geqslant 1$ as in Lemma C.9.

Fix arbitrary $p \in (0,1)$ and $N \geqslant \frac{4C^2 d \log(2Ad/p)}{\lambda_{\min}(\Sigma)^2}$. As $\lambda_{\min}(\Sigma) \leqslant \|\Sigma\|_2 \leqslant 1$, for this $N$, it holds that $\mathbb{P}(\mathcal{E}) \geqslant 1 - p$, where $\mathcal{E}$ denotes the event from Lemma C.9.

Conditioned on event $\mathcal{E}$, for every $a \in \mathcal{A}$, it holds that

$$
\begin{aligned}
\|\widehat{M}_a^\lambda - M_a\|_2 &\leqslant \|(X^\top X + \lambda I_d)^{-1} X^\top Z_a - \lambda (X^\top X + \lambda I_d)^{-1} M_a\|_2 \\
&\leqslant \|(X^\top X + \lambda I_d)^{-1}\|_2 \|X^\top Z_a\|_2 + \lambda \|(X^\top X + \lambda I_d)^{-1}\|_2 \|M_a\|_2 \\
&\leqslant \frac{\|X^\top Z_a\|_2 + \lambda \sqrt{d}}{\lambda_{\min}(X^\top X) + \lambda} \leqslant \frac{C\sqrt{N \log(2Ad/p)} + \sqrt{d}}{N\lambda_{\min}(\Sigma) - C\sqrt{Nd \log(2/p)}} \\
&\leqslant \frac{2C\sqrt{Nd \log(2Ad/p)}}{N\lambda_{\min}(\Sigma)/2} = 4C\sqrt{\frac{d \log(2Ad/p)}{N\lambda_{\min}(\Sigma)^2}}.
\end{aligned}
$$

Note that we use the fact that $\|M_a\|_2 \leqslant \sqrt{d}$ from Lemma C.4. $\qquad \square$

**Lemma C.9** (Concentration). *There exists an absolute constant $C$ such that for all $p \in (0,1)$ and $N \geqslant C^2 \cdot d \log(2Ad/p)$, event $\mathcal{E} = \mathcal{E}_X \cap (\cap_{a \in \mathcal{A}} \mathcal{E}_a)$, where*

$$
\begin{aligned}
\mathcal{E}_X : \quad & \lambda_{\min}(X^\top X) \geqslant N\lambda_{\min}(\Sigma) - C\sqrt{Nd \log(2/p)}, \\
\mathcal{E}_a : \quad & \|X^\top Z_a\|_2 \leqslant C\sqrt{N \log(2Ad/p)},
\end{aligned}
$$

*occurs with probability at least $1 - p$.*

*Proof.* It will suffice to show that there exists constant $C$ such that for every $N \geqslant C^2 \cdot d \log(2Ad/p)$, it holds that $\mathbb{P}(\mathcal{E}_X) \geqslant 1 - \frac{p}{2}$ and $\mathbb{P}(\mathcal{E}_a) \geqslant 1 - \frac{p}{2A}$ for all $a \in \mathcal{A}$.

**Part 1:** Observe that rows in matrix $X$ are independent sub-Gaussian vectors that are uniformly bounded in $l_2$-norm by 1, because $\sup_{s,a} \|\phi(s,a)\|_2 \leqslant 1$. Using Theorem C.10, fix absolute constants $C_1$ and $c_1$ so that

$$
\forall N \in \mathbb{N}, \ \forall t \geqslant 0, \ \mathbb{P}\left(\|X^\top X - N\Sigma\|_2 \leqslant N \max\{\delta, \delta^2\}\right) \geqslant 1 - 2\exp(-c_1 t^2) \quad \text{for } \delta = \frac{C_1\sqrt{d} + t}{\sqrt{N}}.
$$

Then, we claim that $\mathbb{P}(\mathcal{E}_X) \geqslant 1 - \frac{p}{2}$ if we select $C \geqslant C_1 + \sqrt{2/c_1}$.

Note that the minimal eigenvalue of $X^\top X$ can be bounded from below as follows:

$$
\lambda_{\min}(X^\top X) \geqslant \lambda_{\min}(N\Sigma) - \|X^\top X - N\Sigma\|_2.
$$

So, by setting $t = \sqrt{\log(4/p)/c_1}$, we obtain that, for all $N \geqslant C \cdot d \log(2/p)$, it holds that

$$
\begin{aligned}
\mathbb{P}(\mathcal{E}_X) &\geqslant \mathbb{P}\left(\|X^\top X - N\Sigma\|_2 \leqslant C \cdot \sqrt{Nd \log(2/p)}\right) \\
&\geqslant \mathbb{P}\left(\|X^\top X - N\Sigma\|_2 \leqslant N \cdot \frac{C_1\sqrt{d} + t}{\sqrt{N}}\right) \\
&\geqslant 1 - 2\exp(-c_1 t^2) = 1 - \frac{p}{2}.
\end{aligned}
$$

**Part 2:** We claim that $\mathbb{P}(\mathcal{E}_a) \geqslant 1 - \frac{p}{2A}$ for every action $a \in \mathcal{A}$ if we select $C \geqslant 8$.

Observe that for every action $a \in \mathcal{A}$, $Z_a^\top X = \sum_{n=1}^N S_a^{(n)}$, where matrices $S_a^{(n)} := z_a^{(n)} (x^{(n)})^\top$ are independent and satisfy the following properties:

$$
\begin{aligned}
\text{Uniformly bounded:} \quad & \|S_a^{(n)}\|_2 = \|z_a^{(n)}\|_2 \|x^{(n)}\|_2 \leqslant 2 \\
\text{Centered:} \quad & \mathbb{E}[S_a^{(n)}] = \mathbb{E}\left[\mathbb{E}[z_a^{(n)} \mid x^{(n)}](x^{(n)})^\top\right] = \mathbb{E}[\mathbf{0}(x^{(n)})^\top] = 0_{d \times d}.
\end{aligned}
$$

Moreover, it holds that

$$
\begin{aligned}
\|\mathbb{E}[S_a^{(n)} (S_a^{(n)})^\top]\|_2 &\leqslant \mathbb{E}\left[\|x^{(n)}\|_2^2 \cdot \mathbb{E}\left[\|z_a^{(n)} (z_a^{(n)})^\top\|_2 \,\Big|\, x^{(n)}\right]\right] \leqslant 4, \\
\|\mathbb{E}[(S_a^{(n)})^\top S_a^{(n)}]\|_2 &\leqslant \mathbb{E}\left[\|x^{(n)} (x^{(n)})^\top\|_2 \cdot \mathbb{E}\left[\|z_a^{(n)}\|_2^2 \,\Big|\, x^{(n)}\right]\right] \leqslant 4,
\end{aligned}
$$

which implies that the variance statistic of the sum satisfies

$$\nu(Z_a^\top X) \leqslant \sum_{n=1}^N \max\left\{ \|\mathbb{E}[S_a^{(n)}(S_a^{(n)})^\top]\|_2, \|\mathbb{E}[(S_a^{(n)})^\top S_a^{(n)}]\|_2 \right\} \leqslant 4N.$$

By Theorem C.11, we have that

$$\forall t \geqslant 0, \quad \mathbb{P}(\|X^\top Z_a\|_2 \geqslant t) \leqslant 2d \cdot \exp\left(\frac{-t^2/2}{4N+2t/3}\right) \leqslant 2d \cdot \exp\left(\frac{-t^2/8}{N+t}\right).$$

So, for $N \geqslant C^2 \cdot \log(2Ad/p)$, fixing $t = \sqrt{16N \log(4Ad/p)} \leqslant N$, yields

$$\mathbb{P}(\mathcal{E}_a) \geqslant \mathbb{P}\left(\|X^\top Z_a\|_2 \leqslant t\right) \geqslant 1 - 2d \cdot \exp\left(\frac{-t^2/8}{N+t}\right) \geqslant 1 - 2d \cdot \exp\left(\frac{-t^2}{16N}\right) = 1 - \frac{p}{2A}.$$

**Conclusion:** To sum up, the choice of the absolute constant $C = \max\{C_1 + \sqrt{2/c_1}, 8\}$ guarantees that for all $p \in (0,1)$ and $N \geqslant C^2 \cdot d\log(2Ad/p)$, it holds that $\mathbb{P}(\mathcal{E}) \geqslant 1 - p$. $\qquad\square$

**Theorem C.10** (Theorem 5.39 (5.40) from (Vershynin, 2012)). *Let $A$ be $N \times d$ matrix whose rows $A_i$ are independent sub-Gaussian vectors in $\mathbb{R}^d$ with common second moment matrix $\Sigma$. Let $K := \max_{i \in [N]} \|A_i\|_{\psi_2}$ denote the maximal sub-Gaussian norm among the rows. Then, there exist constants $c$ and $C$ that depend only on the value of $K$, such that, for every $t \geqslant 0$, the following inequality holds with probability at least $1 - 2\exp(-ct^2)$:*

$$\|\tfrac{1}{N}A^\top A - \Sigma\|_2 \leqslant \max\{\delta, \delta^2\} \quad \text{where} \quad \delta = \frac{C\sqrt{d}+t}{\sqrt{N}}.$$

**Theorem C.11** (Theorem 6.1.1 (Matrix Bernstein) from (Tropp, 2015)). *Let $S_1, \ldots, S_n$ be independent $\mathbb{R}$-valued centered random matrices with common dimensions $d_1 \times d_2$, and suppose that for some $L \geqslant 0$, it holds that $\|S_k\|_2 \leqslant L$ for every $k \in [n]$ almost surely. Consider their sum $Z := \sum_{k=1}^n S_k$ and let $\nu(Z)$ denote the variance statistic of the sum:*

$$\nu(Z) := \max\left\{ \|\mathbb{E}[ZZ^\top]\|_2, \|\mathbb{E}[Z^\top Z]\|_2 \right\}.$$

*Then, for all $t \geqslant 0$, it holds that*

$$\mathbb{P}(\|Z\|_2 \geqslant t) \leqslant (d_1 + d_2) \cdot \exp\left(\frac{-t^2/2}{\nu(Z) + Lt/3}\right).$$

**Lemma 4.8** (Restated). *For all $p \in (0,1)$, empirical mean estimators $\widehat{\beta}_a$ satisfy*

$$\mathbb{P}\left(\sup_{a \in \mathcal{A}} |\widehat{\beta}_a - \beta_a| \leqslant \sqrt{\frac{12\ln(3A/p)}{Np_{\min}}}\right) \geqslant 1 - p.$$

*Proof.* For every $a \in \mathcal{A}$, let $N_a = \sum_{n=1}^N \mathbb{I}(a_n = a)$ and $S_a = \sum_{n=1}^N b_n \mathbb{I}(a_n = a)$, so that $\widehat{\beta}_a = S_a/N_a$. Also, let $p_a = \mathbb{E}[\mathbb{I}(a_1 = a)]$, so that $p_{\min} = \inf_{a \in \mathcal{A}} p_a$.

By Multiplicative Chernoff Bound, for fixed $a \in \mathcal{A}$ and arbitrary $\varepsilon \in (0,1)$, we have

$$\mathbb{P}(N_a \leqslant \tfrac{1}{2}Np_a) \leqslant \exp(-Np_a/8),$$
$$\mathbb{P}(|S_a - N_a\beta_a| = |(N_a - S_a) - N_a\bar{\beta}_a| \geqslant \varepsilon N_a \max\{\beta_a, \bar{\beta}_a\}|N_a) \leqslant 2\exp(-\varepsilon^2 N_a \max\{\beta_a, \bar{\beta}_a\}/3),$$

which allows us to write

$$\begin{aligned}
\mathbb{P}(|\widehat{\beta}_a - \beta_a| \geqslant \varepsilon) &= \mathbb{P}(|S_a - N_a\beta_a| \geqslant \varepsilon N_a) \\
&\leqslant \mathbb{P}(|S_a - N_a\beta_a| \geqslant \varepsilon N_a | N_a > \tfrac{1}{2}Np_a) + \mathbb{P}(N_a \leqslant \tfrac{1}{2}Np_a) \\
&\leqslant 2\exp(-\varepsilon^2 Np_a \max\{\beta_a, \bar{\beta}_a\}/6) + \exp(-Np_a/8) \\
&\leqslant 3\exp(-\varepsilon^2 Np_{\min}/12).
\end{aligned}$$

Therefore, by the uniform confidence bound, for every $p \in (0,1)$, it indeed holds that

$$\mathbb{P}\left(\sup_{a \in \mathcal{A}} |\widehat{\beta}_a - \beta_a| \leqslant \sqrt{\frac{12\ln(3A/p)}{Np_{\min}}}\right) \geqslant 1 - p.$$

$\qquad\square$

## C.4. Closed Form for Eventually Periodic action-sequences

**Lemma C.12.** *Let $P \geqslant 0$, $L \geqslant 1$, and fix a sequence $\boldsymbol{a} = \boldsymbol{a}^{\mathrm{pr}} \oplus (\oplus_{t=1}^{\infty} \boldsymbol{a}^{\mathrm{per}})$ with prefix $\boldsymbol{a}^{\mathrm{pr}} = (a_1, \ldots, a_P) \in \mathcal{A}^P$ and period $\boldsymbol{a}^{\mathrm{per}} = (a_{P+1}, \ldots, a_{P+L}) \in \mathcal{A}^L$. Define $\Psi^{\mathrm{pr}} = \gamma^P \prod_{i=1}^{P} (\bar{\beta}_{a_i} M_{a_i})$ and $\Psi^{\mathrm{per}} = \gamma^L \prod_{i=P+1}^{P+L} (\bar{\beta}_{a_i} M_{a_i})$. Then:*

$$M_1(\boldsymbol{a}) = \Phi_1^{\mathrm{pr}} + \Psi^{\mathrm{pr}} (I_d - \Psi^{\mathrm{per}})^{-1} \Phi_1^{\mathrm{per}},$$
$$M_2(\boldsymbol{a}) = \Phi_2^{\mathrm{pr}} + \Psi^{\mathrm{pr}} (I_d - \Psi^{\mathrm{per}})^{-1} \Phi_2^{\mathrm{per}},$$

*where*

$$\Phi_1^{\mathrm{pr}} = I_d + \sum_{k=1}^{P} \gamma^k \left( \prod_{i=1}^{k-1} \bar{\beta}_{a_i} M_{a_i} \right) M_{a_k},$$
$$\Phi_1^{\mathrm{per}} = \sum_{j=1}^{L} \gamma^j \left( \prod_{i=P+1}^{P+j-1} \bar{\beta}_{a_i} M_{a_i} \right) M_{a_{P+j}},$$
$$\Phi_2^{\mathrm{pr}} = \sum_{k=1}^{P} \gamma^k \left( \prod_{i=1}^{k-1} \bar{\beta}_{a_i} M_{a_i} \right) \beta_{a_k} M_{a_k},$$
$$\Phi_2^{\mathrm{per}} = \sum_{j=1}^{L} \gamma^j \left( \prod_{i=P+1}^{P+j-1} \bar{\beta}_{a_i} M_{a_i} \right) \beta_{a_{P+j}} M_{a_{P+j}}.$$

*Proof.* Lemma C.4 implies that for every $n \in \mathbb{N}$ and every sequence $(a_1, \ldots, a_n) \in \mathcal{A}^n$, it holds that

$$\rho(\prod_{i=1}^{n} M_{a_i}) \leqslant 1.$$

Hence $I_d - \Psi^{\mathrm{per}}$ is invertible due to $\gamma < 1$ and $L \geqslant 1$. The result follows by expanding $(I_d - \Psi^{\mathrm{per}})^{-1}$ as a Neumann series and grouping the defining sums for $M_1(\boldsymbol{a})$ and $M_2(\boldsymbol{a})$ into prefix and repeated-period contributions. $\square$

We remark that this computation requires $O(P + L)$ matrix multiplications and one matrix inversion, yielding $O((P+L)d^3)$ complexity per sequence.

# D. Episodic Learning: Proofs

In this section, we prove Theorem 5.1. Our proof adapts the approach of Jin et al. (2020b) for ATST-MDPs with geometric horizons.

For notational convenience, let $\mathbf{s}_u^k = \perp$ for all $k \in [K]$ and $u > B^k + 1$. Let $\overline{R}^\tau = \min\{R^\tau, H\}$.

For burst-dependent policy $\boldsymbol{\pi} = (\pi_u)_{u=1}^{\infty}$ and $n \in \mathbb{N}$, let $\boldsymbol{\pi}_{(n)} = (\pi_{u+n-1})_{u=1}^{\infty}$ denote the burst-dependent policy obtained by shifting the original policy by $n - 1$ data-bursts ahead. Then, we introduce notation $K_u^{\boldsymbol{\pi}} = K^{\boldsymbol{\pi}_{(u)}}$ and $V_u^{\boldsymbol{\pi}} = V^{\boldsymbol{\pi}_{(u)}}$.

## D.1. Some Technical Lemmas

In this sections, we state some technical lemmas used in the proof of the main result. The proofs of these lemmas are deferred to later subsections.

First, we need the following lemma, which bounds the growth of the estimator's norm.

**Lemma D.1** (Bound for $\boldsymbol{w}_u^k$). *For all $(k, u) \in [K] \times [H - 1]$, $\|\boldsymbol{w}_u^k\|_2 \leqslant 4\sqrt{dkH^3/\lambda}$.*

*Proof.* For every vector $\boldsymbol{v} \in \mathbb{R}^{2d}$, we have

$$|\boldsymbol{v}^\top \boldsymbol{w}_u^k| = \left| \boldsymbol{v}^\top (\Lambda^k)^{-1} \sum_{\tau=1}^{N^k} \widehat{\boldsymbol{\psi}}^\tau [\overline{R}^\tau + \sup_{\boldsymbol{a}} K_{u+1}^k(\mathbf{s}_N^\tau, \boldsymbol{a})] \right|$$
$$\leqslant \sum_{\tau=1}^{N^k} |\boldsymbol{v}^\top (\Lambda^k)^{-1} \widehat{\boldsymbol{\psi}}^\tau| \cdot 2H$$
$$\leqslant 2H \cdot \sqrt{\left[ \sum_{\tau=1}^{N^k} \|\boldsymbol{v}\|_{(\Lambda^k)^{-1}}^2 \right] \left[ \sum_{\tau=1}^{N^k} \|\widehat{\boldsymbol{\psi}}^\tau\|_{(\Lambda^k)^{-1}}^2 \right]}$$
$$\leqslant 2H \cdot \|\boldsymbol{v}\|_2 \sqrt{kH/\lambda} \cdot \sqrt{2d},$$

where the last step follows from the fact that $N^k \leqslant kH$ and Fact D.9. $\square$

Based on this lemma, we can establish the following concentration result.

**Lemma D.2.** *Under the setting of Theorem 5.1, let $c_\rho$ be the constant parameterizing $\rho$ (i.e., $\rho = c_\rho \cdot dH\sqrt{\iota}$). There exists an absolute constant $C$, independent of $c_\rho$, such that for all fixed $p \in (0,1)$, if we let $\mathcal{E}$ denote the event that*

$$\forall (k,u) \in [K] \times [H-1]: \quad \left\| \sum_{\tau=1}^{N^k} \widehat{\psi}^\tau [V_{u+1}^k(\mathbf{s}_N^\tau) - \mathbb{P}V_{u+1}^k(\mathbf{s}^\tau, \mathbf{a}^\tau)] \right\|_{(\Lambda^k)^{-1}} \leqslant C \cdot \frac{d}{1-\gamma}\sqrt{\chi},$$

$$\forall k \in [K]: \quad \left\| \sum_{\tau=1}^{N^k} \widehat{\psi}^\tau [\overline{R}^\tau - \mathbb{E}[\overline{R}^\tau | \mathbf{s}^\tau, \mathbf{a}^\tau]] \right\|_{(\Lambda^k)^{-1}} \leqslant C \cdot Hd^{1/2}\sqrt{\iota}$$

*where $\chi = \log(2(c_\rho + 1)dKH/p)$, then $\mathbb{P}(\mathcal{E}) \geqslant 1 - p/2$*

See Section D.3 for the proof of this lemma.

To further simplify the notations, we let $\epsilon_2 = \epsilon \cdot 5\rho\sqrt{KH}$. Note that $\epsilon_2 \geqslant \epsilon\|\mathbf{w}_u^k\|_2 + \epsilon\rho$ by Lemma D.1. This constant will be used throughout the rest of the proof. Also, let $\psi_u^k = \psi(\mathbf{s}_u^k, \mathbf{a}_u^k)$ be equal to $\mathbf{0} \in \mathbb{R}^{2d}$ when $\mathbf{s}_u^k = \perp$.

We also need the following two lemmas. The first lemma provides lower bounds on the estimated action-sequence value-functions on the event that the concentration bounds hold true.

**Lemma D.3** (UCB). *Under the setting of Theorem 5.1, conditioned on event $\mathcal{E}$ from Lemma D.2,*

$$K_u^k(s, \mathbf{a}) \geqslant K^*(s, \mathbf{a}) - (H - u) \cdot \epsilon_2$$

*for all $(s, \mathbf{a}, u, k) \in \mathcal{S} \times \mathcal{A}^{\mathbb{N}} \times [H] \times [K]$.*

Additionally, we need the following lemma, which provides a recursive relation on a term arising from the error decomposition.

**Lemma D.4** (Recursive formula). *For $k \in [K]$, $u \in [H]$, we define*

- $\delta_u^k = V_u^k(\mathbf{s}_u^k) - V_u^{\pi^k}(\mathbf{s}_u^k)$,

- $\zeta_{u+1}^k = \mathbb{E}[\delta_{u+1}^k | \mathbf{s}_u^k, \mathbf{a}_u^k] - \delta_{u+1}^k$.

*Then, conditioned on the event $\mathcal{E}$, we have that for every $(k,u) \in [K] \times [H-1]$:*

$$\delta_u^k \leqslant \delta_{u+1}^k + \zeta_{u+1}^k + 2\rho\|\psi_u^k\|_{(\Lambda^k)^{-1}} + \epsilon_2.$$

See Section D.4 for the proof of Lemma D.3 and D.4.

### D.2. Proof of Theorem 5.1

Given lemmas in Section D.1, we are ready to prove Theorem 5.1. To start with, let us recall the statement of the theorem.

**Theorem 5.1** (Restated). *Suppose Algorithm 1 is executed with $\epsilon$-admissible feature map $\widehat{\psi}$ for $\epsilon \leqslant \sqrt{(1-\gamma)/K}$. There exists an absolute constant $c \geqslant 1$, such that, for all fixed $p \in (0,1)$, if we set $H = \lceil \frac{\log(K(1-\gamma)^{-1})}{1-\gamma} \rceil + 1$, $\lambda = 1$, and $\rho = c \cdot dH\sqrt{\iota}$ with $\iota = \log(2dKH/p)$, then with probability at least $1 - p$, the total regret is at most*

$$\widetilde{O}\left(\sqrt{d^3 K(1-\gamma)^{-3}\iota^2} + d^2(1-\gamma)^{-2}\iota + \epsilon \cdot \sqrt{d^2 K^3(1-\gamma)^{-5}\iota}\right).$$

*Proof.* We condition on the event $\mathcal{E}$ from Lemma D.2, which occurs with probability at least $1 - p/2$. Then, using Lemmas D.3 and D.4 and the choice of $\epsilon_2$, we can write:

$$\mathcal{R}_K = \sum_{k=1}^K \left[ V^*(\mathbf{s}_1^k) - V_1^{\pi^k}(\mathbf{s}_1^k) \right] \leqslant \sum_{k=1}^K (\delta_1^k + H\epsilon_2)$$

$$\leqslant \sum_{k=1}^K \sum_{u=1}^H \zeta_u^k + \sum_{k=1}^K \delta_H^k + 2\rho \sum_{k=1}^K \sum_{u=1}^{H-1} \|\psi_u^k\|_{(\Lambda^k)^{-1}} + 2KH\epsilon_2$$

$$\leqslant \sum_{k=1}^K \sum_{u=1}^H \zeta_u^k + \sum_{k=1}^K \delta_H^k + 2\rho \sum_{k=1}^K \sum_{u=1}^{H-1} \|\widehat{\psi}_u^k\|_{(\Lambda^k)^{-1}} + 4KH\epsilon_2.$$

- To bound the first component, we use Azuma-Hoeffding for the martingale difference sequence $\{\zeta_u^k\}_{u,k}$ (ordered chronologically with respect to rounds/episodes and including $B^k < u \leqslant H$ with $\mathbf{s}_u^k = \perp$), which satisfies $|\zeta_u^k| \leqslant \frac{2}{1-\gamma}$. For all $t \geqslant 0$, we have

$$\mathbb{P}\left(\sum_{k=1}^K \sum_{u=1}^H \zeta_u^k \leqslant t\right) \geqslant 1 - \exp\left(\frac{-t^2}{8KH(1-\gamma)^{-2}}\right).$$

Hence, with probability at least $1 - p/4$, we have that

$$\sum_{k=1}^K \sum_{u=1}^H \zeta_u^k \leqslant \sqrt{8KH(1-\gamma)^{-2}} \cdot \sqrt{\log(4/p)}.$$

- To bound the second component, observe that for each $k \in [K]$

$$\delta_H^k = V_H^k(\mathbf{s}_H^k) - V_H^{\boldsymbol{\pi}^k}(\mathbf{s}_H^k) \leqslant \frac{\mathbb{I}(\mathbf{s}_H^k \neq \perp)}{1-\gamma} - 0 \leqslant \frac{\mathbb{I}(H^k \geqslant H)}{1-\gamma},$$

and use Chernoff inequality for binary indicators $\mathbb{I}(H^k \geqslant H)$. For all $\delta \geqslant 1$, it holds that

$$\mathbb{P}\left(\sum_{k=1}^K \mathbb{I}(H^k \geqslant H) > (1+\delta)K\gamma^{H-1}\right) \leqslant \left(\frac{e^{-\delta}}{(1+\delta)^{1+\delta}}\right)^{K\gamma^{H-1}}$$

$$\leqslant \exp\left(\frac{-\delta^2 K\gamma^{H-1}}{2+\delta}\right) \leqslant \exp(-\delta K\gamma^{H-1}/3).$$

Then, by Fact D.7, with probability at least $1 - p/4$, by setting $\delta = \frac{3\log(4/p)}{K\gamma^{H-1}} \geqslant 1$, it holds that

$$\sum_{k=1}^K \delta_H^k \leqslant (1+\delta)K\gamma^{H-1}(1-\gamma)^{-1}$$

$$\leqslant (K\gamma^{H-1} + 3\log(4/p))(1-\gamma)^{-1}$$

$$\leqslant 6\log(4/p)(1-\gamma)^{-1}.$$

- To bound the third component, let $\Lambda_u^k = \Lambda^k + \sum_{u'=1}^{u-1} \widehat{\boldsymbol{\psi}}_{u'}^k(\widehat{\boldsymbol{\psi}}_{u'}^k)^\top$ for $\widehat{\boldsymbol{\psi}}_u^k = \widehat{\psi}(\mathbf{s}_u^k, \boldsymbol{a}_u^k)$. Then, write the following

$$\sum_{k=1}^K \sum_{u=1}^H \|\widehat{\boldsymbol{\psi}}_u^k\|_{(\Lambda^k)^{-1}} \leqslant \sqrt{H} \cdot \sum_{k=1}^K \sqrt{\sum_{u=1}^H \|\widehat{\boldsymbol{\psi}}_u^k\|_{(\Lambda^k)^{-1}}^2}$$

$$\overset{(a)}{\leqslant} \sqrt{H} \cdot \sum_{k=1}^K \sqrt{\sum_{u=1}^H 2\|\widehat{\boldsymbol{\psi}}_u^k\|_{(\Lambda_u^k)^{-1}}^2}$$

$$+ \sqrt{H} \cdot \sum_{k=1}^K \mathbb{I}(\det(\Lambda^{k+1}) > 2\det(\Lambda^k))\sqrt{H/\lambda}$$

$$\leqslant \sqrt{2KH} \cdot \sqrt{\sum_{k=1}^K \sum_{u=1}^H (\widehat{\boldsymbol{\psi}}_u^k)^\top (\Lambda_u^k)^{-1} \widehat{\boldsymbol{\psi}}_u^k}$$

$$+ \sqrt{H^2/\lambda} \cdot \sum_{k=1}^K \mathbb{I}(\det(\Lambda^{k+1}) > 2\det(\Lambda^k))$$

$$\overset{(b)}{\leqslant} \sqrt{2KH} \cdot \sqrt{2\log\left(\frac{\det(\Lambda^{K+1})}{\det(\Lambda^1)}\right)} + \sqrt{H^2\lambda^{-1}} \cdot \log_2\left(\frac{\det(\Lambda^{K+1})}{\det(\Lambda^1)}\right)$$

$$\overset{(c)}{\leqslant} 4\sqrt{KH} \cdot \sqrt{d\log(2KH)} + 4H \cdot d\log(2KH),$$

where (a) follows from Fact D.8, (b) from Fact D.10, and (c) from the following inequality

$$\frac{\det(\Lambda^{K+1})}{\det(\Lambda^1)} \leqslant \left(\frac{\lambda_{\max}(\Lambda^{K+1})}{\lambda_{\min}(\Lambda^1)}\right)^{2d} \leqslant \left(\frac{\lambda+KH}{\lambda}\right)^{2d} = (1+KH)^{2d} \leqslant (2KH)^{2d}.$$

In conclusion, we have that with probability at least $1 - p$:

$$\mathcal{R}_K \leqslant \sqrt{8KH(1-\gamma)^{-2}} \cdot \sqrt{\log(4/p)}$$

$$+ 6\log(4/p)(1-\gamma)^{-1}$$

$$+ 2\rho \cdot \left(4\sqrt{KH} \cdot \sqrt{d\log(2KH)} + 4H \cdot d\log(2KH)\right)$$

$$+ 4KH \cdot 5\epsilon\rho\sqrt{KH}$$

$$\leqslant c_1 \cdot \sqrt{d^3 KH^3 \iota^2} + c_2 \cdot d^2 H^2 \iota + c_3 \cdot \epsilon KH \cdot \sqrt{d^2 KH^3 \iota},$$

for some absolute constants $c_1, c_2, c_3$. $\qquad\square$

### D.3. Proof of Lemma D.2

In Theorem 5.1, we have $H = \lceil \frac{\log(K(1-\gamma)^{-1})}{1-\gamma} \rceil + 1$, $\lambda = 1$, and $\iota = \log(2dKH/p)$.

From Lemma D.1, $\|\boldsymbol{w}_u^k\|_2 \leqslant 4\sqrt{dkH^3/\lambda}$. Hence, by combining Lemmas D.12 and D.13 for function class $\mathcal{V}(4\sqrt{dkH^3/\lambda}, \rho, \lambda)$, we show that for all $\varepsilon > 0$, with probability at least $1 - p/4$: for all $(k, u) \in [K] \times [H-1]$,

$$\left\|\sum_{\tau=1}^{N^k} \widehat{\boldsymbol{\psi}}^\tau [V_{u+1}^k(\mathbf{s}_N^\tau) - \mathbb{P}V_{u+1}^k(\mathbf{s}^\tau, \boldsymbol{a}^\tau)]\right\|_{(\Lambda^k)^{-1}}^2 \leqslant \frac{4}{(1-\gamma)^2}\left[d\log\frac{kH+\lambda}{\lambda} + 2d\log\left(1 + \frac{16\sqrt{dkH^3}}{\varepsilon\sqrt{\lambda}}\right)\right.$$
$$\left. + 4d^2\log\left(1 + \frac{16\rho^2\sqrt{d}}{\varepsilon^2\lambda}\right) + \log\left(\frac{4}{p}\right)\right] + \frac{8k^2H^2\varepsilon^2}{\lambda}.$$

We set $\lambda = 1$ and $\rho = c_\rho \cdot dH\sqrt{\iota}$ and pick $\varepsilon = \frac{d}{(1-\gamma)kH}$. Then, there clearly exists absolute constant $C_1 > 0$, independent of $c_\rho$, such that

$$\left\|\sum_{\tau=1}^{N^k} \widehat{\boldsymbol{\psi}}^\tau [V_{u+1}^k(\mathbf{s}_N^\tau) - \mathbb{P}V_{u+1}^k(\mathbf{s}^\tau, \boldsymbol{a}^\tau)]\right\|_{(\Lambda^k)^{-1}}^2 \leqslant C_1 \cdot \frac{d^2}{(1-\gamma)^2}\log(2(c_\rho + 1)dKH/p).$$

For the second part, we will use the concentration of self-normalized process, where $\overline{R}^\tau | \mathbf{s}^\tau, \boldsymbol{a}^\tau \in [0, H]$ is a $H$-sub-Gaussian. By applying Theorem D.11, we can find absolute constant $C_2 > 0$ independent of $c_\rho$ such that with probability at least $1 - p/4$: for all $k \in [K]$,

$$\left\|\sum_{\tau=1}^{N^k} \widehat{\boldsymbol{\psi}}^\tau [\overline{R}^\tau - \mathbb{E}[\overline{R}^\tau | \mathbf{s}^\tau, \boldsymbol{a}^\tau]]\right\|_{(\Lambda^k)^{-1}}^2 \leqslant 4H^2\left[d\log\left(\frac{kH+\lambda}{\lambda}\right) + \log\left(\frac{4}{p}\right)\right]$$
$$\leqslant C_2 \cdot H^2 d\log(2kH/p).$$

Finally, set $C = \sqrt{\max\{C_1, C_2\}}$ to finish the proof.

### D.4. Proof of Lemmas D.3 and D.4

The proof relies on the following technical lemma.

**Lemma D.5.** *Under the setting of Theorem 5.1, there exists an absolute constant $c_\rho \geqslant 1$ such that for $\rho = c_\rho \cdot dH\sqrt{\iota}$ and arbitrary burst-dependent policy $\boldsymbol{\pi}$, on the event $\mathcal{E}$ from Lemma D.2, for all $(x, \boldsymbol{a}, k, u) \in \mathcal{X} \times \mathcal{A}^\mathbb{N} \times [K] \times [H-1]$:*

$$\langle \boldsymbol{\psi}(x, \boldsymbol{a}), \boldsymbol{w}_u^k \rangle - K_u^{\boldsymbol{\pi}}(x, \boldsymbol{a}) = \mathbb{P}(V_{u+1}^k - V_{u+1}^{\boldsymbol{\pi}})(x, \boldsymbol{a}) + \Delta_u^k(x, \boldsymbol{a}),$$

*where $\Delta_u^k(x, \boldsymbol{a})$ satisfies $|\Delta_u^k(x, \boldsymbol{a})| \leqslant \rho \|\boldsymbol{\psi}(x, \boldsymbol{a})\|_{(\Lambda^k)^{-1}}$.*

See Section D.4.1 for the proof of this lemma. Taking this lemma as given, let us now proceed with the proofs of Lemma D.3 and D.4.

*Proof of Lemma D.3.* We set $K_H^k(s, \boldsymbol{a}) = \frac{1}{1-\gamma} \geqslant K^*(s, \boldsymbol{a})$. Moreover, for all $u \in [H-1]$, we have that

$$K_u^k(s, \boldsymbol{a}) = \langle \widehat{\boldsymbol{\psi}}(s, \boldsymbol{a}), \boldsymbol{w}_u^k \rangle + \rho \|\widehat{\boldsymbol{\psi}}(s, \boldsymbol{a})\|_{(\Lambda^k)^{-1}}$$
$$\geqslant \langle \boldsymbol{\psi}(s, \boldsymbol{a}), \boldsymbol{w}_u^k \rangle + \rho \|\boldsymbol{\psi}(s, \boldsymbol{a})\|_{(\Lambda^k)^{-1}} - (\epsilon\|\boldsymbol{w}_u^k\|_2 + \rho\epsilon/\sqrt{\lambda})$$
$$\overset{(a)}{\geqslant} K^*(s, \boldsymbol{a}) + \mathbb{P}(V_{u+1}^k - V^*)(s, \boldsymbol{a}) - \epsilon_2$$
$$\geqslant K^*(s, \boldsymbol{a}) + \inf_{s', \boldsymbol{a}'}(K_{u+1}^k - K^*)(s', \boldsymbol{a}') - \epsilon_2,$$

where (a) follows from Lemmas D.5 and the choice of $\epsilon_2$.
Then, the statement follows by trivial induction over $u$ from $u = H$ to $u = 1$. $\qquad\square$

*Proof of Lemma D.4.* We can write the following by Lemma D.5 for all $s, \boldsymbol{a}$:

$$K_u^k(s, \boldsymbol{a}) - K_u^{\boldsymbol{\pi}^k}(s, \boldsymbol{a}) = \langle \widehat{\boldsymbol{\psi}}(s, \boldsymbol{a}), \boldsymbol{w}_u^k \rangle + \rho \|\widehat{\boldsymbol{\psi}}(s, \boldsymbol{a})\|_{(\Lambda^k)^{-1}} - \langle \boldsymbol{\psi}(s, \boldsymbol{a}), \boldsymbol{w}_u^{\boldsymbol{\pi}^k} \rangle$$
$$\leqslant \langle \boldsymbol{\psi}(s, \boldsymbol{a}), \boldsymbol{w}_u^k \rangle + \rho \|\boldsymbol{\psi}(s, \boldsymbol{a})\|_{(\Lambda^k)^{-1}} - \langle \boldsymbol{\psi}(s, \boldsymbol{a}), \boldsymbol{w}_u^{\boldsymbol{\pi}^k} \rangle + \epsilon_2$$
$$\leqslant \mathbb{P}(V_{u+1}^k - V_{u+1}^{\boldsymbol{\pi}^k})(s, \boldsymbol{a}) + 2\rho \|\boldsymbol{\psi}(s, \boldsymbol{a})\|_{(\Lambda^k)^{-1}} + \epsilon_2.$$

From the choice of $\boldsymbol{\pi}^k$, we have that

$$
\begin{aligned}
\delta_u^k &= K_u^k(\mathbf{s}_u^k, \boldsymbol{a}_u^k) - K_u^{\boldsymbol{\pi}^k}(\mathbf{s}_u^k, \boldsymbol{a}_u^k) \\
&\leqslant \mathbb{P}(V_{u+1}^k - V_{u+1}^{\boldsymbol{\pi}^k})(\mathbf{s}_u^k, \boldsymbol{a}_u^k) + 2\rho\|\boldsymbol{\psi}(\mathbf{s}_u^k, \boldsymbol{a}_u^k)\|_{(\Lambda^k)^{-1}} + \epsilon_2 \\
&= \delta_{u+1}^k + \zeta_{u+1}^k + 2\rho\|\boldsymbol{\psi}_u^k\|_{(\Lambda^k)^{-1}} + \epsilon_2.
\end{aligned}
$$

Note that this holds even when $\mathbf{s}_u^k = \perp$, as $0 \leqslant \epsilon_2$. $\qquad\square$

### D.4.1. PROOF OF LEMMA D.5

We first state and prove the following lemma.

**Lemma D.6** (Burst-dependent version of Theorem 4.4). *Under Assumption 4.1, for arbitrary burst-dependent policy* $\boldsymbol{\pi} = (\pi_u)_{u=1}^{\infty}$ *and* $u \in \mathbb{N}$, *it holds that: for all* $(x, \boldsymbol{a}) \in \mathcal{X} \times \mathcal{A}^{\mathbb{N}}$,

$$
K_u^{\boldsymbol{\pi}}(x, \boldsymbol{a}) = \langle \boldsymbol{\psi}(x, \boldsymbol{a}), \boldsymbol{w}_u^{\boldsymbol{\pi}} \rangle,
$$

*where* $\boldsymbol{w}_u^{\boldsymbol{\pi}} = 2 \begin{bmatrix} \boldsymbol{\theta}/(1-\gamma) \\ \int_{\mathcal{S}} V_{u+1}^{\boldsymbol{\pi}}(s) d\boldsymbol{\mu}(s) \end{bmatrix}$ *satisfies* $\|\boldsymbol{w}_u^{\boldsymbol{\pi}}\| \leqslant \frac{4\sqrt{d}}{1-\gamma}$.

*Proof.* Follows by decomposition $K_u^{\boldsymbol{\pi}} = R + \mathbb{P}V_{u+1}^{\boldsymbol{\pi}}$ and Theorem C.2. $\qquad\square$

Now we turn to the proof of Lemma D.5. As $(\boldsymbol{\psi}^\tau)^\top \boldsymbol{w}_u^{\boldsymbol{\pi}} = K_u^{\boldsymbol{\pi}}(\mathbf{s}^\tau, \boldsymbol{a}^\tau)$ by Lemma D.6, we have the following

$$
\begin{aligned}
\boldsymbol{w}_u^k - \boldsymbol{w}_u^{\boldsymbol{\pi}} &= (\Lambda^k)^{-1} \sum_{\tau=1}^{N^k} \widehat{\boldsymbol{\psi}}^\tau [\overline{R}^\tau + V_{u+1}^k(\mathbf{s}_N^\tau)] - \boldsymbol{w}_u^{\boldsymbol{\pi}} \\
&= (\Lambda^k)^{-1} \left\{ -\lambda \boldsymbol{w}_u^{\boldsymbol{\pi}} + \sum_{\tau=1}^{N^k} \widehat{\boldsymbol{\psi}}^\tau [\overline{R}^\tau + V_{u+1}^k(\mathbf{s}_N^\tau) - K_u^{\boldsymbol{\pi}}(\mathbf{s}^\tau, \boldsymbol{a}^\tau)] \right\} \\
&\quad + (\Lambda^k)^{-1} \sum_{\tau=1}^{N^k} \widehat{\boldsymbol{\psi}}^\tau (\boldsymbol{\psi}^\tau - \widehat{\boldsymbol{\psi}}^\tau)^\top \boldsymbol{w}_u^{\boldsymbol{\pi}} \\
&= \underbrace{-\lambda(\Lambda^k)^{-1}\boldsymbol{w}_u^{\boldsymbol{\pi}}}_{\boldsymbol{q}_1} + \underbrace{(\Lambda^k)^{-1} \sum_{\tau=1}^{N^k} \widehat{\boldsymbol{\psi}}^\tau [V_{u+1}^k(\mathbf{s}_N^\tau) - \mathbb{P}V_{u+1}^k(\mathbf{s}^\tau, \boldsymbol{a}^\tau)]}_{\boldsymbol{q}_2} \\
&\quad + \underbrace{(\Lambda^k)^{-1} \sum_{\tau=1}^{N^k} \widehat{\boldsymbol{\psi}}^\tau [\mathbb{P}(V_{u+1}^k - V_{u+1}^{\boldsymbol{\pi}})(\mathbf{s}^\tau, \boldsymbol{a}^\tau)]}_{\boldsymbol{q}_3} + \underbrace{(\Lambda^k)^{-1} \sum_{\tau=1}^{N^k} \widehat{\boldsymbol{\psi}}^\tau [\overline{R}^\tau - \mathbb{E}[\overline{R}^\tau | \mathbf{s}^\tau, \boldsymbol{a}^\tau]]}_{\boldsymbol{q}_4} \\
&\quad + \underbrace{(\Lambda^k)^{-1} \sum_{\tau=1}^{N^k} \widehat{\boldsymbol{\psi}}^\tau [\mathbb{E}[\overline{R}^\tau | \mathbf{s}^\tau, \boldsymbol{a}^\tau] - \mathbb{E}[R^\tau | \mathbf{s}^\tau, \boldsymbol{a}^\tau]]}_{\boldsymbol{q}_5} + \underbrace{(\Lambda^k)^{-1} \sum_{\tau=1}^{N^k} \widehat{\boldsymbol{\psi}}^\tau (\boldsymbol{\psi}^\tau - \widehat{\boldsymbol{\psi}}^\tau)^\top \boldsymbol{w}_u^{\boldsymbol{\pi}}}_{\boldsymbol{q}_6}.
\end{aligned}
$$

We bound these six components separately. Note that

$$
\begin{aligned}
|\boldsymbol{\psi}(x, \boldsymbol{a})^\top (\Lambda^k)^{-1} \sum_{\tau=1}^{N^k} \widehat{\boldsymbol{\psi}}^\tau| &\leqslant \sum_{\tau=1}^{N^k} |\boldsymbol{\psi}(x, \boldsymbol{a})^\top (\Lambda^k)^{-1} \widehat{\boldsymbol{\psi}}^\tau| \\
&\leqslant \left[ \sum_{\tau=1}^{N^k} \|\boldsymbol{\psi}(x, \boldsymbol{a})\|_{(\Lambda^k)^{-1}}^2 \right]^{1/2} \left[ \sum_{\tau=1}^{N^k} \|\widehat{\boldsymbol{\psi}}^\tau\|_{(\Lambda^k)^{-1}}^2 \right]^{1/2} \\
&\leqslant \sqrt{kH} \|\boldsymbol{\psi}(x, \boldsymbol{a})\|_{(\Lambda^k)^{-1}} \cdot \sqrt{d} \\
&= \sqrt{dkH} \cdot \|\boldsymbol{\psi}(x, \boldsymbol{a})\|_{(\Lambda^k)^{-1}}.
\end{aligned}
$$

- To bound $\boldsymbol{q}_1$, using Lemma D.6, write

$$
\begin{aligned}
|\langle \boldsymbol{\psi}(x, \boldsymbol{a}), \boldsymbol{q}_1 \rangle| &\leqslant \lambda \|\boldsymbol{w}_u^{\boldsymbol{\pi}}\|_{(\Lambda^k)^{-1}} \|\boldsymbol{\psi}(x, \boldsymbol{a})\|_{(\Lambda^k)^{-1}} \\
&\leqslant \sqrt{\lambda} \|\boldsymbol{w}_u^{\boldsymbol{\pi}}\|_2 \|\boldsymbol{\psi}(x, \boldsymbol{a})\|_{(\Lambda^k)^{-1}} \leqslant \frac{4\sqrt{d\lambda}}{1-\gamma} \cdot \|\boldsymbol{\psi}(x, \boldsymbol{a})\|_{(\Lambda^k)^{-1}}.
\end{aligned}
$$

- To bound $\boldsymbol{q}_2$ and $\boldsymbol{q}_4$, we use event $\mathcal{E}$ so that

$$
|\langle \boldsymbol{\psi}(x, \boldsymbol{a}), \boldsymbol{q}_2 + \boldsymbol{q}_4 \rangle| \leqslant C \cdot dH\sqrt{\chi} \cdot \|\boldsymbol{\psi}(x, \boldsymbol{a})\|_{(\Lambda^k)^{-1}},
$$

for some absolute constant $C > 0$ independent of $c_\rho$.

- To bound $q_3$, using Theorem C.2, observe that for some vector $v$ such that $\|v\|_2 \leqslant \frac{8\sqrt{d}}{1-\gamma}$:

$$\mathbb{P}(V_{u+1}^k - V_{u+1}^{\pi})(x, a) = \langle \psi(x, a), v \rangle.$$

Then, we can write

$$\langle \psi(x, a), q_3 \rangle = \langle \psi(x, a), v \rangle - \underbrace{\lambda \, \psi(x, a)^\top (\Lambda^k)^{-1} v}_{c_1}$$

$$+ \underbrace{\psi(x, a)^\top (\Lambda^k)^{-1} \sum_{\tau=1}^{N^k} \widehat{\psi}^\tau (\psi^\tau - \widehat{\psi}^\tau)^\top v}_{c_2},$$

where $c_1, c_2$ can be bounded as follows:

$$|c_1| \leqslant \sqrt{\lambda} \, \|v\|_2 \, \|\psi(x, a)\|_{(\Lambda^k)^{-1}} \leqslant \frac{8\sqrt{d\lambda}}{1-\gamma} \cdot \|\psi(x, a)\|_{(\Lambda^k)^{-1}}$$

$$|c_2| \leqslant |\psi(x, a)^\top (\Lambda^k)^{-1} \sum_{\tau=1}^{N^k} \widehat{\psi}^\tau| \cdot \epsilon \|v\|_2$$

$$\leqslant \sqrt{dkH} \cdot \|\psi(x, a)\|_{(\Lambda^k)^{-1}} \cdot \epsilon \cdot \frac{8\sqrt{d}}{1-\gamma} \leqslant 8\sqrt{\epsilon^2 d^2 kH(1-\gamma)^{-2}} \cdot \|\psi(x, a)\|_{(\Lambda^k)^{-1}}.$$

- To bound $q_5$, note that, as rewards are bounded to $[0, 1]$, we have

$$|\mathbb{E}[\overline{R}^\tau | \mathbf{s}^\tau, a^\tau] - \mathbb{E}[R^\tau | \mathbf{s}^\tau, a^\tau]]| \leqslant \gamma^H (1-\gamma)^{-1}.$$

By Fact D.7, for $H \geqslant \frac{\log(K(1-\gamma)^{-1})}{1-\gamma}$, $\gamma^H \leqslant \frac{1}{\sqrt{KH}}$, so we have

$$|\langle \psi(x, a), q_5 \rangle| \leqslant \frac{\gamma^H}{1-\gamma} \cdot |\psi(x, a)^\top (\Lambda^k)^{-1} \sum_{\tau=1}^{N^k} \widehat{\psi}^\tau|$$

$$\leqslant \frac{\sqrt{dkH}}{(1-\gamma)\sqrt{KH}} \cdot \|\psi(x, a)\|_{(\Lambda^k)^{-1}} \leqslant dH \cdot \|\psi(x, a)\|_{(\Lambda^k)^{-1}}$$

- To bound $q_6$, we write

$$|\langle \psi(x, a), q_6 \rangle| \leqslant \epsilon \|w_u^{\pi}\|_2 \cdot |\psi(x, a)^\top (\Lambda^k)^{-1} \sum_{\tau=1}^{N^k} \widehat{\psi}^\tau|$$

$$\leqslant \epsilon \cdot \frac{4\sqrt{d}}{1-\gamma} \cdot \sqrt{dkH} \cdot \|\psi(x, a)\|_{(\Lambda^k)^{-1}}$$

$$\leqslant 4\sqrt{\epsilon^2 d^2 kH(1-\gamma)^{-2}} \cdot \|\psi(x, a)\|_{(\Lambda^k)^{-1}}$$

To sum up, for our choice of $\lambda = 1$ and $\epsilon \leqslant \sqrt{\frac{1-\gamma}{K}}$ we have that

$$\Delta_u^k(x, a) \leqslant (25 + C) \cdot dH\sqrt{\chi} \cdot \|\psi(x, a)\|_{(\Lambda^k)^{-1}}.$$

Finally, observe that $c_\rho$ appears in $\chi$ only under the logarithm and $C$ is an absolute constant. Therefore, we can select $c_\rho$ as an absolute constant large enough such that for $\iota \geqslant \log(2)$, $c_\rho \cdot \sqrt{\iota} \geqslant (25 + C)\sqrt{\iota + \log(c_\rho + 1)}$, i.e. $\rho = c_\rho \cdot dH\sqrt{\iota} \geqslant (25 + C)dH\sqrt{\chi}$ for all $K, H, d, p$.

### D.5. Some Basic Facts

In this section, we collect some basic algebraic facts used in the proofs.

*Fact* D.7. For $n \geqslant \frac{\log(K(1-\gamma)^{-1})}{1-\gamma}$ it holds that $\gamma^n \leqslant \min\{\frac{1-\gamma}{K}, \frac{1}{n(1-\gamma)}\} \leqslant \frac{1}{\sqrt{Kn}}$.

*Proof.* As $\log(1/x) \geqslant 1 - x$ for $x > 0$, we can write

$$\gamma^n = \exp\left(-n\log(1/\gamma)\right) \leqslant \exp\left(-\log(K(1-\gamma)^{-1})\right) = \frac{1-\gamma}{K}.$$

Moreover, as $1/x \geqslant e^{-x}$ for $x > 0$, we also have

$$\gamma^n = \exp\left(-n\log(1/\gamma)\right) \leqslant \frac{1}{n\log(1/\gamma)} \leqslant \frac{1}{n(1-\gamma)}.$$

The final inequality follows trivially. $\square$

*Fact* D.8. Let $A, B \in \mathbb{R}^{d \times d}$ be positive definite matrices and $\boldsymbol{x} \in \mathbb{R}^d$. If $A \succeq B$, then

$$\|\boldsymbol{x}\|_A \leqslant \|\boldsymbol{x}\|_B \sqrt{\frac{\det(A)}{\det(B)}}.$$

*Fact* D.9. Let $(\boldsymbol{x}_n)_{n=1}^N$ be an $\mathbb{R}^D$-valued sequence and $\lambda > 0$. Then, for $\Lambda_N = \lambda I + \sum_{n=1}^N \boldsymbol{x}_n \boldsymbol{x}_n^\top$, it holds that

$$\sum_{n=1}^N \|\boldsymbol{x}_n\|_{(\Lambda_N)^{-1}}^2 \leqslant D.$$

*Proof.* Proof is exactly the same as in Lemma D.1 from Jin et al. (2020b). □

*Fact* D.10 (Abbasi-yadkori et al. (2011)). Let $(\boldsymbol{x}_n)_{n=1}^\infty$ be an $\mathbb{R}^D$-valued sequence such that $\|\boldsymbol{x}_n\|_2 \leqslant 1$ for every $n \in \mathbb{N}$. Let $\Lambda_0 \in \mathbb{R}^{D \times D}$ satisfy $\lambda_{\min}(\Lambda_0) \geqslant 1$ and define $\Lambda_N = \Lambda_0 + \sum_{n=1}^N \boldsymbol{x}_n \boldsymbol{x}_n^\top$ for every $n \in \mathbb{N}$. Then, it holds that: for all $N \in \mathbb{N}$,

$$\log\left[\frac{\det(\Lambda_N)}{\det(\Lambda_0)}\right] \leqslant \sum_{n=1}^N \|\boldsymbol{x}_n\|_{\Lambda_{n-1}^{-1}}^2 \leqslant 2\log\left[\frac{\det(\Lambda_N)}{\det(\Lambda_0)}\right].$$

### D.6. Concentration Inequalities

**Theorem D.11** (Self-Normalized Bound for Vector-Valued Martingales, Abbasi-yadkori et al. (2011)). *Let $\{\varepsilon_\tau\}_{\tau=1}^\infty$ be a $\mathbb{R}$-valued stochastic process with corresponding filtration $\{\mathcal{F}_\tau\}_{\tau=0}^\infty$, such that $\varepsilon_\tau \mid \mathcal{F}_{\tau-1}$ is zero-mean and $\sigma$-sub-Gaussian for every $\tau \geqslant 1$. Let $\{\boldsymbol{\zeta}_\tau\}_{\tau=1}^\infty$ be an $\mathbb{R}^D$-valued stochastic process where $\boldsymbol{\zeta}_\tau \in \mathcal{F}_{\tau-1}$. Let $\Lambda \in \mathbb{R}^{D \times D}$ be a positive definite matrix and define $\Lambda_N = \Lambda + \sum_{\tau=1}^N \boldsymbol{\zeta}_\tau \boldsymbol{\zeta}_\tau^\top$ for $N \geqslant 0$. Then, for all $\delta > 0$, with probability at least $1 - \delta$, it holds that*

$$\forall N \geqslant 0: \qquad \left\|\sum_{\tau=1}^N \boldsymbol{\zeta}_\tau \varepsilon_\tau\right\|_{(\Lambda_N)^{-1}}^2 \leqslant 2\sigma^2 \log\left(\frac{\det(\Lambda_N)^{1/2}\det(\Lambda)^{-1/2}}{\delta}\right).$$

**Lemma D.12.** *Let $\mathcal{V} \subset \mathbb{R}^{\mathcal{S}}$ be an arbitrary function class such that, for every $V \in \mathcal{V}$, $\sup_s |V(s)| \leqslant \frac{1}{1-\gamma}$. Let $\{s_\tau\}_{\tau=1}^\infty$ be a stochastic process on state space $\mathcal{S}$ with corresponding filtration $\{\mathcal{F}_\tau\}_{\tau=0}^\infty$. Let $\{\boldsymbol{\zeta}_\tau\}_{\tau=1}^\infty$ be an $\mathbb{R}^D$-valued stochastic process where $\boldsymbol{\zeta}_\tau \in \mathcal{F}_{\tau-1}$ and $\|\boldsymbol{\zeta}_\tau\|_2 \leqslant 1$. Let $\Lambda_N = \lambda I + \sum_{\tau=1}^N \boldsymbol{\zeta}_\tau \boldsymbol{\zeta}_\tau^\top$ for $\lambda > 0$. Then, for all $\varepsilon, \delta > 0$, with probability at least $1 - \delta$, it holds that for all $N \geqslant 0$ and $V \in \mathcal{V}$*

$$\left\|\sum_{\tau=1}^N \boldsymbol{\zeta}_\tau \{V(s_\tau) - \mathbb{E}[V(s_\tau) \mid \mathcal{F}_{\tau-1}]\}\right\|_{(\Lambda_N)^{-1}}^2 \leqslant \frac{4}{(1-\gamma)^2}\left[\frac{D}{2}\log\left(\frac{N+\lambda}{\lambda}\right) + \log\left(\frac{\mathcal{N}_\varepsilon}{\delta}\right)\right] + \frac{8N^2\varepsilon^2}{\lambda},$$

*where $\mathcal{N}_\varepsilon$ is the $\varepsilon$-covering number of $\mathcal{V}$ with respect to $dist(V, V') = \sup_s |V(s) - V'(s)|$.*

*Proof.* The result follows by applying Theorem D.11 for each element in the $\varepsilon$-covering and using the union bound for the left-hand side, as was done in the proof of Lemma D.4 from Jin et al. (2020b). □

**Lemma D.13** (Covering number bound, (Jin et al., 2020b)). *Let $\boldsymbol{\zeta} : \mathcal{S} \times \mathcal{A}^{\mathbb{N}} \to \mathbb{R}^D$ be an arbitrary state-action-sequence feature map, such that $\sup_{s,\boldsymbol{a}} \|\boldsymbol{\zeta}(s,\boldsymbol{a})\|_2 \leqslant 1$. For $L, B, \lambda > 0$, let $\mathcal{V}(L, B, \lambda)$ denote the following parametric class of mappings from $\mathcal{S}$ to $[0, \frac{1}{1-\gamma}]$:*

$$\left\{V(.) = \min\{\tfrac{1}{1-\gamma}, \sup_{\boldsymbol{a} \in \mathcal{A}^{\mathbb{N}}} \boldsymbol{\zeta}(.,\boldsymbol{a})^\top \boldsymbol{w} + \rho \|\boldsymbol{\zeta}(.,\boldsymbol{a})\|_{\Lambda^{-1}}\} : \|\boldsymbol{w}\|_2 \leqslant L, \rho \in [0, B], \Lambda \succeq \lambda I\right\}.$$

*Then, the covering number $\mathcal{N}_\varepsilon$ of $\mathcal{V}(L, B, \lambda)$ with respect to $dist(V, V') = \sup_{s \in \mathcal{S}} |V(s) - V'(s)|$ satisfies*

$$\log \mathcal{N}_\varepsilon \leqslant D \log(1 + 4L/\varepsilon) + D^2 \log\left(1 + 8D^{1/2}B^2/(\lambda\varepsilon^2)\right).$$

*Proof.* Accounting for the fact that we use a different feature map $\boldsymbol{\zeta} : \mathcal{S} \times \mathcal{A}^{\mathbb{N}} \to \mathbb{R}^D$, the proof follows similarly to Lemma D.6 from Jin et al. (2020b). □

