# OpenReview forum: "Reinforcement Learning with Action-Triggered Observations"
_ICML.cc/2026/Conference — ICML 2026 regular_

### Official Review · Reviewer_7Cpv · 2026-03-04

**Soundness:** 2
**Presentation:** 3
**Significance:** 1
**Originality:** 2
**Overall Recommendation:** 4
**Confidence:** 4

**Summary:**

This work introduces action-triggered sporadically traceable MDPs (ATST-MDPs), in which the agent's state is not systematically observed. Instead, at each step, the agent triggers its observation through an action $a$ that determines whether its state is revealed with some probability $\beta(a)$. Leveraging the specific structure of the underlying POMDP, a Bellman equation is derived on an embedded space. A linear representation with finite-dimensional features is derived, with probabilistic guarantees for the plug-in parameter estimates. A UCB-style algorithm is then proposed to optimize the linear value function, and regret bounds are provided.

**Compliance With Llm Reviewing Policy:**

Affirmed.

**Final Justification:**

The authors clarified my misunderstandings.

**Key Questions For Authors:**

- Why is finite-dimensional embedding reasonable in ATST-MDPs? Is it at all? From my understanding, even if the probability of not seeing an observation decreases geometrically, the running time until a data-burst is not formally bounded. This may also explain why the authors switched their formulation from a variable-dimensional state to an infinite-dimensional one. We are not in the standard setup where the state-space is continuous (of finite dimension, by the way), but can be fully explained by a smaller, finite number of features. Here, the dimension itself is contracted to a finite number of features, even though the original state-action variables live on a finite space.

- Why, if at all, is state-augmentation a good idea for learning ATST-MDPs? Even assuming that we can fully express the model parameters in a finite-dimensional space, we see in Thm 5.1 a quadratic dependence on $d$. Don't we gain expressiveness at the expense of higher regret? If so, is Assumption 4.1 reasonable?

**Limitations:**

The paper does not include any discussion on the limitations of this work. Several should have been addressed, as described in my previous comments.

**Strengths And Weaknesses:**

**Strengths**
- Soundness: The problem is well-described, and the text is clearly written. The theoretical analysis is conducted cleanly and rigorously.
- Presentation: The paper is well structured, and the narrative is easy to follow.
- Significance: The problem formulation and its solution provide relevant tools for related problems, such as delayed RL and other temporal perturbations of classical MDPs.
- Originality: This work reminds me of similar setups, yet it is not the same in all aspects. Perhaps the most closely related line of works is delayed observation MDPs, but there: (1) the delay is out of the agent's control (termed as *exogenous* in this work); (2) all states are eventually observed (no sparsity), with some latency (as opposed to the current agent's state in ATST-MDPs).

**Weaknesses**
- Soundness: Even though the probability of *data-burst* is controlled through $\beta$, it is still exogenous in some sense. For example, it does not depend on the agent's true state, only on its action. It is not time-dependent either, making stationary policies optimal in the augmented space at least. It is also not adversarial: the agent's actions do not directly affect the probability of a data burst, whereas we could imagine a trade-off between highly-rewarding actions and a low probability of observing the state. This leads me to my next point below.
- Significance: Although the analysis is comprehensive, its framework is quite limited, as augmentation directly solves the underlying MDP. The motivation provided in Sec. 1 is accordingly weak in my opinion, because the illustrative example is not fully covered by the considered setting. Typically, the trade-off mentioned in paragraph 1 better aligns with adversarial blindness described above.
- Presentation: Section 3.1 looks unnecessary to me: from the beginning of Sec. 3, I was surprised that the authors chose to formulate the problem in an augmented state-space with variable dimension $n+1$, instead of directly analyzing it in $\mathcal{S} \times\mathcal{A}^{<\mathbb{N}}$. I understand that Thm 3.1 relies on the former setup, but looking at the proof, it can be established directly on the latter state-space.
- Originality: The connection with goal-conditioned RL is a bit obscure to me, despite its presentation as "state-triggered feedback" versus "action-triggered feedback". The former seems more general to me, because it is harder to control. The related work section and its extended version in the appendix are too superficial. Section A lists and describes a list of related works without further diving into their connection/difference with ATST-MDPs, nor explaining what these works miss that this one brings. Several references are also missing: [1, 2, 3]. Another issue addressed in [1, 4] that is occulted here is the dimensional explosion of learning algorithms under state-augmentation.
- l.208-212: "linear MDP has emerged as a standard ..." - Please include references.

[1] Derman, E., Dalal, G., & Mannor, S. Acting in Delayed Environments with Non-Stationary Markov Policies. In International Conference on Learning Representations.

[2] Rubino, G., & Sericola, B. (1989). Sojourn times in finite Markov processes. Journal of Applied Probability, 26(4), 744-756.

[3] Ascione, G., & Cuomo, S. (2022). A sojourn-based approach to semi-Markov reinforcement learning. Journal of Scientific Computing, 92(2), 36.

[4] Walsh, Thomas J., et al. "Learning and planning in environments with delayed feedback." Autonomous Agents and Multi-Agent Systems 18.1 (2009): 83-105.

---

> ### Author Rebuttal · Authors · 2026-03-31
>
> We thank the reviewer for recognizing the paper’s clarity and technical rigor. However, several concerns raised appear to stem from overlooking the central feature of ATST-MDPs: that observability is directly affected by the chosen action. Since this point underlies several concerns raised, we clarify it first.
> ## Clarifying the setting: ATST-MDPs do capture reward-information trade-offs.
> The reviewer writes that "the agent's actions do not directly affect the probability of a data burst” — we respectfully disagree: this is precisely the **defining feature of ATST-MDPs**. Since any observation probabilities $\beta\in [0,1]^{\mathcal A}$ can be used to extend an MDP to an ATST-MDP, the framework does capture important reward-information trade-offs. For instance, highly rewarding actions may carry lower observation probability, as in our motivating examples.
>
>
>
>
> As with most existing works on partially-observed RL, we work under a stochastic setting, i.e., the state observation depends on the action stochastically, instead of through an adversarial deterministic sequence. This covers many important problems such as paid observations. We would welcome clarification of the suggested "adversarial blindness" setting so that we can discuss the connection in the revision.
>
>
>
>
> *Regarding extensions:* (a) State-action-dependent observation probabilities are an interesting direction for future work that does not follow immediately from our paper. See our detailed response to Reviewer iXp7, Q2. (b) While we consider discounted time-homogeneous MDPs, we believe our analysis can be directly extended to time-dependent models.
>
>
> ## Section 3.1 is useful: Augmented states and delayed RL.
> Augmented states are the standard tool in delayed RL (Walsh et al., 2009). Starting from the augmented state space mirrors the classical POMDP-to-belief-MDP reduction. The progression from augmented states to action sequences is deliberate: show the general structure first, establish existence of the optimal policy, then exploit the renewal property for the simpler formulation.
>
>
>
>
> ## Discussion on related work.
> We will incorporate all suggested references and expand the related work discussion in Appendix A, sharpening comparisons to delayed-observation MDPs, goal-conditioned RL, and other temporally structured models. As reviewers correctly note, standard goal-conditioned RL assumes full observability, which is fundamentally different from our setting; we will correct this in the revision. Finally, while state-triggered and action-triggered observations extend the fully-observed MDP in different directions, we emphasize that the action-triggered model already captures important phenomena—in particular, the trade-off between high-reward and high-information actions.
>
>
>
>
>
>
> ## Linear MDPs: Assumption 4.1 and finite-dimensional embedding.
> Assumption 4.1 is a standard structural assumption in RL theory (Jin et al., 2019; Yang & Wang, 2019, 2020; Zanette et al., 2020; Zhou et al., 2021) — we will add these citations at l.208–212. It subsumes finite MDPs (Remark 4.2) and it can accommodate infinite measurable state spaces with finite-dimensional structure.
> * *Q1 (finite-dimensional embedding):* As explained above, the finite-dimensional embedding in linear MDP framework is a standard structural assumption. In ATST-MDPs, this structure allows us to compress the belief state into a low-dimensional representation, making it possible to learn the optimal value function with linear function approximation.
> The running time until a data-burst is indeed not bounded, but the geometric discounting under linear dynamics allows the information about an infinite action sequence to be represented in a $2d$-dimensional vector of finite norm, avoiding the dimensional explosion of learning algorithms under state-augmentation. This is the consequence of the absolute convergence of the series in (4a)–(4b), guaranteed by $\gamma < 1$ and the bound ​in Lemma C.3.
> The linear MDP structure allows the state space itself to be general, and requires the states to admit a low-dimensional feature vector representation. It is therefore a natural model for the “standard setup” suggested by the reviewer.’
> * *Q2 (regret dependence on $d$):* We use state augmentation in our derivation to formulate the Bellman equation crucial to our analysis. However, under linear MDP assumption, the belief state admits an efficient finite-dimensional representation.
> The leading term in Theorem 5.1 is $\tilde O(d^{3/2}\sqrt{K}(1-\gamma)^{-3/2})$, which matches the rate for fully-observed linear MDPs (Jin et al., 2019). The $d^2(1-\gamma)^{-2}$ term does not grow with $K$. Thus, the penalty for this assumption is not prohibitive.
>
>
> ## Addressing limitations.
> We have discussed several limitations (in both theory and practical implementation) in the paper, and we will expand the limitations discussion in the revision following reviewers' suggestions.

---

> > ### Author Rebuttal · Reviewer_7Cpv · 2026-04-03
> >
> > The authors answered all my questions. I raise my score accordingly.

---

### Official Review · Reviewer_qt8V · 2026-03-12

**Soundness:** 4
**Presentation:** 4
**Significance:** 3
**Originality:** 3
**Overall Recommendation:** 5
**Confidence:** 4

**Summary:**

The paper studies a special case of partially observable RL where at each step the probability of observing the current state depends on the chosen action. This setting is motivated e.g. by medical cases where different tests may give different insights into the patient's state but have different associated costs/consequences. The ATST-MDP framework is formalized and shown to be equivalent to an MDP with a state space augmented by action sequences (similar to belief state in POMDPs). Then, existence of an optimal policy is shown via Bellman optimality. To enable learning in ATST-MDPs, the common linear MDP structure is assumed and it is shown that a certain formulation of value functions in the ATST-MDPs then also admit a linear representation. Based on this, a regression-type learning algorithm is proposed and shown to achieve sublinear regret (matching the rate under full observability).

**Compliance With Llm Reviewing Policy:**

Affirmed.

**Final Justification:**

My questions have been resolved in the rebuttal. My assessment remains unchanged.

**Key Questions For Authors:**

1) One limitation of ATST-MDPs (compared to POMDPs) seems to be that a state is either fully observed or not observed at all. In practice (e.g. the motivating medical example) it might also be the case that some inexact state observations are made. This seems significantly more challenging because unlike in ATST-MDPs the agent does *not* know whether the received observation is the true state or not. Do you see this as a limitation of ATST-MDPs?

2) In the experiments it is described how \beta may affect the convergence. Can this also be explained by the regret guarantee, or otherwise, why is there no dependence on (a lower/upper bound of) \beta(a) ?

**Limitations:**

yes

**Strengths And Weaknesses:**

The paper proposes a well-motivated new variant of an MDP under partial observability for which they can provide a sample-efficient learning algorithm without the need for restrictive assumptions. The results seem technically sounds and are well-presented. By starting with different formulations of the ATST-MDP, showing Bellman optimality, and then moving to learning under the linearity assumption, the paper builds up the theory in a structured way. Some limitations are discussed, such as reliance on an optimization oracle. It might be insightful to also discuss (e.g. give an example) of a setting where observations are action-dependent, but which cannot be modelled by ATST-MDPs (see also question 1).

---

> ### Author Rebuttal · Authors · 2026-03-31
>
> We thank the reviewer for the positive assessment and insightful questions. We are glad the reviewer found the setting well motivated and the theory clearly structured. We will also incorporate a clearer discussion of action-dependent observation models that fall outside ATST-MDPs.
> ## Q1. Is it a limitation that observations are either exact or absent?
> Yes — this is a deliberate modeling restriction. This exact-revelation property is precisely what makes ATST-MDPs tractable. If observations were instead noisy or partial, then the relevant sufficient statistic would no longer reduce to “last observation + actions since then” and the agent would generally need to reason about a richer posterior over latent states. In that case, the problem moves back toward a general POMDP.
>
> In the medical example, ATST-MDPs are better viewed as modeling diagnostic procedures that either yield a definitive result or remain inconclusive. This is closer to an extensive confirmatory testing process than to a single noisy test that may produce false positives or false negatives.
>
> Extending our framework to this setting is an interesting direction for future work. One natural approach is through factored MDPs, where a state $s \in \mathcal{S}$ factorizes into $s = (s^{(1)}, s^{(2)}, \ldots, s^{(n)})$ and each action may affect the observation probability of each component $s^{(j)}$ independently. This factorization structure is compatible with the linear representation used in our paper. We will discuss this in more detail in the revised manuscript.
> ## Q2. Can the effect of $\beta$ in the experiments be explained by the regret guarantee? Why is there no explicit dependence on $\beta(a)$?
> In our simulations, $\beta \in [0,1]^{\mathcal A}$ does affect convergence, but this is fully consistent with Theorem 5.1. The theorem provides a worst-case regret guarantee for each fixed $\beta$; it does not imply that all observation regimes are equally easy empirically. For an ATST-MDP $M_\beta$ extending an MDP $M$ with observation probabilities $\beta$, the benchmark (comparator) is the optimal augmented policy $\pi^\star_\beta$ induced by that same observation regime. Theorem 5.1 then establishes that $\tilde O(\sqrt K)$ regret is achievable against this benchmark, uniformly over all $\beta$.
>
> This assessment is consistent with the edge cases. When $\beta \equiv 1$, we recover the standard fully observed linear MDP setting. When $\beta \equiv 0$, the agent commits to a single open-loop action sequence throughout every episode, and the problem reduces to a linear bandit, where $\tilde O(\sqrt K)$ rates are standard. To sum up, our result establishes that the full range of intermediate regimes with different $\beta$ is learnable at the same $\tilde O(\sqrt K)$ rate.
>
> The RiverSwim experiment illustrates why different values of $\beta$ can still lead to different empirical convergence behavior. In RiverSwim, the optimal behavior is the same across $\beta$: always move right. Smaller $\beta \equiv \beta^\star$ leads to longer intervals between observations and hence longer open-loop commitments. Since the optimal strategy already requires persistent commitment, this makes the correct behavior easier to identify empirically.

---

> > ### Author Rebuttal · Reviewer_qt8V · 2026-04-01
> >
> > Thanks for the clear answers to my questions. It confirms my assessment and I maintain my score in support of the paper.

---

### Official Review · Reviewer_wGtg · 2026-03-13

**Soundness:** 3
**Presentation:** 3
**Significance:** 4
**Originality:** 4
**Overall Recommendation:** 5
**Confidence:** 2

**Summary:**

In this work, the authors proposed a new reinforcement learning (RL) setting, in which the state and the cumulative reward are only observed stochastically each round. The probability of this full state and cumulative reward observation depends on the chosen action. The author then formulate the Bellman operator under this scenario. Under linear MDP assumption, the value function can be reduced to a linear representation, and the author proposed an RL algorithm that achieves $\tilde{O}(\sqrt{Kd^{3}/(1-\gamma)^{3}})$ regret. Some experiments demonstrate that the algorithm works in practice.

**Compliance With Llm Reviewing Policy:**

Affirmed.

**Final Justification:**

My concerns are all addressed by the authors, and I will stick with my original score.

**Key Questions For Authors:**

1. In your problem formulation, why do you let the learner observe the cumulative reward instead of instantaneous reward during data-burst. According to your motivation, it makes more sense to observe instantaneous reward.
2. When the number of episode is not large enough, how does the polylogarithmic term affects the regret?
3. Does $p_{\min}$ affects the overall regret under the case where $\beta$ are unknown?

**Limitations:**

Yes, the author explicitly says that their algorithm requires access to an optimization oracle over action sequence.

**Strengths And Weaknesses:**

This is a very interesting work with substantial impact to the partially observed Markov decision process (POMDP) problems.

**Strength**
1. The authors formulate a new RL problem, which has substantial novelty and good motivation as well.
2. In this new scenario, the Bellman equation is solved.
3. The linear representation of the value function under linear MDP assumption provides significant reduction for the problem, making the difficult POMDP problem solvable.
4. The authors propose an algorithm that achieves $\tilde{O}(\sqrt{Kd^{3}/(1-\gamma)^{3}})$ regret, commensurate with that in linear MDPs with full observation.

**Weakness**
1. The main results are under linear MDP cases, which limits its generality.
2. Some acronym are introduced without explanation, making the paper a bit hard to read (e.g., ST-LSVI-UCB).
3. It would also be useful to explain what $\mathcal{O}$ is before using the notation.

---

> ### Author Rebuttal · Authors · 2026-03-31
>
> We thank the reviewer for the positive assessment and thoughtful questions. We are glad the reviewer found the setting novel and technically interesting. We will also revise the presentation to expand acronyms such as ST-LSVI-UCB on first use and introduce notation more carefully.
> ## Q1. Cumulative vs. instantaneous reward observation.
> We believe the cumulative-reward model is well motivated for the applications we have in mind. In healthcare, one may periodically observe an overall patient health score, rather than cleanly decompose the effect of each individual intervention. Similarly, in finance, one often observes cumulative gains or losses over an interval, rather than a per-step reward attribution. Thus, cumulative outcomes are a natural feedback model in settings where only aggregate effects are occasionally measurable.
>
>
> Crucially, the alternative model in which the agent observes only the instantaneous reward with probability $\beta(a)$ is not solvable in general. For example, consider a one-state system with $\mathcal A=\{a_0,a_1\}$, where $\beta(a_0)=0$, $\beta(a_1)=1$, $r(a_1)=0.5$, and $r(a_0)\in\{0,1\}$. Since the agent never observes the reward of $a_0$, it is impossible to learn whether always choosing $a_0$ or always choosing $a_1$ is optimal. We will include this detail in the revision.
> ## Q2. Polylogarithmic term when $K$ is small.
> The dependence on $K$ in Theorem 5.1 beyond the leading $\sqrt{K}$ term is only polylogarithmic, at most an $O(\log^3 K)$ factor. In practice, this does not appear problematic in our simulations; for example, with $K=10^5$ episodes we already observe fast convergence in both RiverSwim and RiverBalance.
> ## Q3. Effect of $p_{\min}$ when $\beta$ is unknown.
> The quantity $p_{\min}$ affects the overall regret in Theorem 5.1 only indirectly, through the $\epsilon$-accuracy of feature-map estimation. Its effect is at most of order $1/\sqrt{p_{\min}}$, so smaller $p_{\min}$ means poorer action coverage and hence harder estimation.
>
>
> Moreover, this dependence matters only when $p_{\min}$ is sufficiently small: if $p_{\min} \gtrsim \lambda_{\min}(\Sigma)^2/d^2$ (Corollary 4.9), then estimation of $\beta$ is no longer the bottleneck, and the feature-estimation guarantee is instead controlled by $\lambda_{\min}(\Sigma)$.

---

> > ### Author Rebuttal · Reviewer_wGtg · 2026-04-03
> >
> > Thank you all for clear response addressing my concern. I will stick with my score.

---

### Official Review · Reviewer_iXp7 · 2026-03-13

**Soundness:** 3
**Presentation:** 3
**Significance:** 3
**Originality:** 3
**Overall Recommendation:** 5
**Confidence:** 2

**Summary:**

This work studies reinforcement learning with an interesting twist: the state of the environment is not revealed to the agent at every step, but only with the action-dependent probability $\beta(a)$. This setting is motivated from practical considerations and analyzed theoretically under the linear MDP assumption. This setting brings with it several complications, most problematic of which is the need to consider infinite open-loop action sequences instead of single actions, since there is no bound on the horizon before which the state is revealed again. The main result of this paper shows that, in the linear MDP setting (in which feature $\phi(s, a)$ are known), there exist features $\psi(s, \boldsymbol{a})$ (where $\boldsymbol a$ is an infinite action sequence) under which the action value function can be expressed as a finite-dimensional inner product. This insight is used to devise an algorithm (an augmented version of LSVI-UCB) which achieves sublinear regret if the features $\psi$ are estimated to a high degree of accuracy (which the authors show is possible).

**Compliance With Llm Reviewing Policy:**

Affirmed.

**Final Justification:**

This is an interesting paper in RL theory, and the authors adequately addressed my questions. It seems that all reviewers agree that this works should be accepted at ICML.

**Key Questions For Authors:**

1. I would like to understand the impact of the action-triggered property on the speed of learning in this setting. It seems to me that the (linear) ATST-MDP is an inherently more difficult setting than the normal (linear) MDP. Could you explain how your regret bound reflects this?
2. It is a bit strange to me that $\beta$ does not depend on the state (since the dynamical effect of the action clearly depends on the state). How would the setting change if $\beta$ depended on both $s$ and $a$?
3. You mention that this setting "subsumes" the previous frameworks studied by Nam et al. What does this mean? (Isn't it simply a different setting?)
4. You write that "goal-conditioned RL (Schaul et al., 2015; Andrychowicz et al., 2018) provides observations only upon goal attainment." This is different from my understanding of goal-conditioned RL, in which the state is revealed at every step (only the reward is very sparse and depends on the goal). Can you explain this?

**Limitations:**

yes

**Strengths And Weaknesses:**

I believe that the setting studied in this paper is interesting and novel. The result showing that there is a feature $\psi$ with dimensionality $2d$ is very unexpected, and the conclusion that this setting can be tackled by LSVI-UCB with sublinear regret is very interesting. Overall, the paper is well written, and I recommend acceptance to ICML. I do have a few questions which I think should be addressed in more detail in the paper. One small nitpick is that the font sizes in Figure 4 and 5 need to be adjusted to be readable.

---

> ### Author Rebuttal · Authors · 2026-03-31
>
> We thank the reviewer for the positive assessment and these helpful questions. We will revise the paper accordingly, and we will also enlarge the font sizes in Figures 4-5.
> ## Q1. How does the action-triggered property affect the speed of learning?
> The action-triggered observations indeed make the problem harder in terms of information structure. Our paper shows that this hardness is reflected in the comparator (benchmark), not in the regret rate (speed of learning). For an ATST-MDP $M_\beta$ extending MDP $M$ with observation probabilities $\beta \in [0,1]^{\mathcal A}$, the goal is to learn the optimal $\beta$-dependent augmented policy $\pi^\star_\beta$. Thus, Theorem 5.1 shows that it is possible to achieve $\tilde O(\sqrt K)$ regret against the appropriate $\beta$-dependent benchmark, uniformly over all $\beta$.
>
> This assessment is consistent with the edge cases. When $\beta \equiv 1$, we recover the standard fully observed linear MDP setting. When $\beta \equiv 0$, the agent commits to a single open-loop action sequence throughout every episode, and the problem reduces to a linear bandit, where $\tilde O(\sqrt K)$ rates are standard. To sum up, our result establishes that the full range of intermediate regimes with different $\beta$ is learnable at the same $\tilde O(\sqrt K)$ rate.
>
> At the same time, different choices of $\beta$ can still lead to different empirical convergence behavior. This does not contradict Theorem 5.1, which gives a worst-case guarantee for each fixed $\beta$ but does not say that all $\beta$ are equally easy. In RiverSwim, the optimal behavior is the same across $\beta$: always move right. Smaller $\beta \equiv \beta^\star$ leads to longer intervals between observations and hence longer open-loop commitments. Since the optimal strategy already requires persistent commitment, this makes the correct behavior easier to identify empirically, which is what we observe in Figure 4.
> ## Q2. Why does $\beta$ depend only on the action, and what if it depended on $(s,a)$?
> This is a deliberate modeling choice. We use action-dependent observation probabilities $\beta(a)$ to capture settings where the agent controls how often state observations occur through its choice of action.
>
> If the observation probability were allowed to depend on the latent state as well, i.e., $\beta(s,a)$, then even the **absence of an observation could reveal information** about the state. For example, if $\mathcal S={s_1,s_2}$, $\mathcal A={a_1,a_2}$, and $\beta(s_i,a_j)=\mathbb{I}(i=j)$, then after taking action $a_j$, whether or not a data-burst occurs reveals the previous state. In such a model, the agent learns state information not only from received observations, but also from missing ones. As a result, the belief dynamics no longer admit the integral form in equation (1). While one can still write a Bellman equation on an augmented state space, our extension to linear MDPs no longer goes through, since the linearity of the belief representation established in Lemma 4.3 breaks down.
>
> We agree that allowing state-action-dependent observation probabilities is a natural extension, which does not follow directly from our current framework. We will clarify this point in the revision and highlight the extension to $\beta(s,a)$ as an important direction for future work.
> ## Q3. In what sense does your framework “subsume” Nam et al.?
> By “subsumes,” we mean “contains as a strict special case”. The ACNO-MDP of Nam et al. models paid observations: at each step, the agent has to choose whether to purchase a state observation at an additional cost or not. This can be represented in our framework by replacing each base action $a \in \mathcal A$ with two variants, $a^0$ and $a^1$, where $a^0$ does not reveal the state ($\beta(a^0)=0$) and $a^1$ does ($\beta(a^1)=1$), with the observation cost absorbed into the reward for $a^1$. Therefore, ACNO-MDPs correspond to a special case of ATST-MDPs in which the observation mechanism is a deterministic binary choice. We will clarify this more explicitly in the revision.
> ## Q4. Goal-conditioned RL.
> The reviewer is correct, and we thank them for catching this. Our sentence on goal-conditioned RL was imprecise. In standard goal-conditioned RL, the state is typically observed at every step, while sparsity arises through the reward or success signal associated with goal achievement. This is a **different observational model** from ours. We included this line of work only to help position our model among RL settings with sparse feedback, and we will revise the discussion accordingly to make this distinction precise. Thank you.

---

> > ### Author Rebuttal · Reviewer_iXp7 · 2026-04-03
> >
> > I thank the authors for their detailed response, and am happy to keep my score at 5.

---

### Decision · Program_Chairs · 2026-04-30

**Decision:**

Accept (regular)

**Comment:**

This paper studies reinforcement learning with partial observations, where the probability of observing the state depends on the taken actions. The key contribution is the insight that, even in this setting, the problem can be transformed into a standard MDP by augmenting the state representation with an estimated feature constructed from the linear features provided by the environment. The authors then establish theoretical guarantees for convergence. Overall, it seems that all reviewers agree that the paper addresses a novel setting and contains nontrivial technical contributions.